# Learning Multi-Index Models with Neural Networks via Mean-Field Langevin Dynamics

**Alireza Mousavi-Hosseini[1,2], Denny Wu[3,4], Murat A. Erdogdu[1,2]**
[1]University of Toronto, [2]Vector Insitute, [3]New York University, [4]Flatiron Institute
{mousavi,erdogdu}@cs.toronto,edu, dennywu@nyu.edu

## Abstract

We study the problem of learning multi-index models in high-dimensions using a two-layer neural network trained with the mean-field Langevin algorithm. Under mild distributional assumptions on the data, we characterize the *effective dimension* $d_{\text{eff}}$ that controls both sample and computational complexity by utilizing the adaptivity of neural networks to latent low-dimensional structures. When the data exhibit such a structure, $d_{\text{eff}}$ can be significantly smaller than the ambient dimension. We prove that the sample complexity grows almost linearly with $d_{\text{eff}}$, bypassing the limitations of the information and generative exponents that appeared in recent analyses of gradient-based feature learning. On the other hand, the computational complexity may inevitably grow exponentially with $d_{\text{eff}}$ in the worst-case scenario. Motivated by improving computational complexity, we take the first steps towards polynomial time convergence of the mean-field Langevin algorithm by investigating a setting where the weights are constrained to be on a compact manifold with positive Ricci curvature, such as the hypersphere. There, we study assumptions under which polynomial time convergence is achievable, whereas similar assumptions in the Euclidean setting lead to exponential time complexity.

## 1 Introduction

A key characteristic of neural networks is their adaptability to the underlying statistical model. Several works have shown that shallow neural networks trained by (variants of) gradient descent can adapt to inherent structures in the learning problem, and learn functions of low-dimensional projections with a sample complexity that depends on properties of the nonlinear link function such as the *information exponent* (Ben Arous et al., 2021) or *generative exponent* (Damian et al., 2024) for single-index models, and the *leap complexity* (Abbe et al., 2023) for multi-index models. Specifically, prior works typically established a sample complexity of $n \gtrsim d^{\Theta(s)}$ for gradient-based learning, where $s$ can be the information/leap exponent (Abbe et al., 2022; Bietti et al., 2022; Damian et al., 2023; Ba et al., 2023; Mousavi-Hosseini et al., 2023b; Bietti et al., 2023; Dandi et al., 2023) or the generative exponent (Dandi et al., 2024; Lee et al., 2024; Arnaboldi et al., 2024; Joshi et al., 2024), depending on the implementation of gradient descent. This sample complexity is also predicted by the framework of statistical query lower bounds (Damian et al., 2022; Abbe et al., 2023; Damian et al., 2024).

On the other hand, neural networks can efficiently *approximate* arbitrary multi-index models regardless of the generative/leap exponent $s$ (Barron, 1993; E et al., 2022); moreover, if the (polynomial) optimization budget is not taken into consideration, there exist computationally inefficient training algorithms that can achieve sample complexity independent of $s$ (Bach, 2017; Liu et al., 2024; Damian et al., 2024). Intuitively speaking, a function depending on $k = O_d(1)$ directions of the input data has $kd = O(d)$ parameters to be estimated, and hence the information theoretically optimal algorithm on isotropic data only requires $n \asymp d$ samples. However, thus far it has been relatively unclear whether standard first-order optimization algorithms for neural networks inherit this optimality.

A promising approach to close the sample complexity gap is to consider neural networks in the *mean-field regime* (Nitanda & Suzuki, 2017; Chizat & Bach, 2018; Mei et al., 2018; Rotskoff & Vanden-Eijnden, 2018; Sirignano & Spiliopoulos, 2020), where overparameterization is utilized to lift the gradient descent dynamics into the space of measures so that global convergence can be established. While most existing results in this regime focus on optimization instead of generalization/learnability,

recent works have shown that under restrictive data and target assumptions (such as XOR), mean-field neural networks achieve a sample complexity that does not depend on the leap complexity (Wei et al., 2019; Chizat & Bach, 2020; Telgarsky, 2023; Suzuki et al., 2023b). Among these works, Suzuki et al. (2023b) proved quantitative convergence guarantees for learning $k$-parity with $n \asymp d$ samples, despite the target function having leap index $k$. Key to this result is the convergence rate analysis of the *mean-field Langevin algorithm (MFLA)* (Hu et al., 2019; Nitanda et al., 2022; Chizat, 2022b). However, existing learnability guarantees in the mean-field regime fall short in the following aspects:

- **Learning general multi-index models.** Prior works established optimal sample complexity for mean-field neural networks under stringent assumptions on the data distribution (isotropic Gaussian, hypercube, etc.) as well as on the target function such as single-index models with specific link functions (Berthier et al., 2023; Mahankali et al., 2023), or $k$-sparse parity classification (Wei et al., 2019; Telgarsky, 2023; Suzuki et al., 2023b). *Hence, the problem of universally learning functions of low-dimensional projections with minimal data assumptions using neural networks with a standard training procedure remains largely open.*

- **Polynomial computational complexity.** To achieve optimal sample complexity, the computational complexity of the training algorithm in Telgarsky (2023); Suzuki et al. (2023b) is exponential in the ambient (input) dimension. Although such exponential dependence may be unavoidable in the most general setting, *sufficient conditions under which the mean-field algorithm can achieve statistical efficiency with polynomial compute is relatively under-explored*, with the exception of a recent work that studied the specific example of $k$-parity on anisotropic data (Nitanda et al., 2024).

## 1.1 OUR CONTRIBUTIONS

Motivated by the above discussion, in this work we address two key questions. First, we ask

*Can we train two-layer neural networks using the MFLA to learn arbitrary multi-index models with an (information theoretically) optimal sample complexity?*

We answer this in the affirmative by showing that empirical risk minimization on a standard variant of a two-layer neural network can be achieved by the MFLA. This result handles arbitrary multi-index models on subGaussian data with general covariance, hence enabling us to obtain a sample complexity with *optimal dimension dependence* up to polylogarithmic factors with standard gradient-based training. However, such a universal guarantee will inevitably suffer from an exponential computational complexity; thus, the second fundamental question we aim to answer is

*Are there conditions under which the computational complexity of the MFLA can be improved from exponential to (quasi)polynomial dimension dependence?*

We provide a positive answer in two problem settings. In the Euclidean setting, we show that the complexity of MFLA is governed by the *effective dimension* of the learning problem, instead of its ambient dimension; this implies an improved efficiency of MFLA when the data is anisotropic. In the Riemannian setting, we outline concrete conditions on the Ricci curvature of the compact manifold defining the weight space under which MFLA converges in polynomial time.

## 1.2 RELATED WORKS

**Mean-field Langevin dynamics.** The training dynamics of neural networks in the mean-field regime is described by a nonlinear partial differential equation in the space of parameter distributions (Chizat & Bach, 2018; Mei et al., 2018; Rotskoff & Vanden-Eijnden, 2018). Unlike the neural tangent kernel (NTK) description (Jacot et al., 2018; Chizat et al., 2019) that freezes the parameters around random initialization, the mean-field regime allows the parameters to travel and learn useful features, leading to improved statistical efficiency. While convergence analyses for mean-field neural networks are typically *qualitative*, in that they do not specify the speed of convergence or finite-width discrepancy, the mean-field Langevin algorithm that we study is a noticeable exception, for which the quantitative convergence rate (Hu et al., 2019; Nitanda et al., 2022; Chizat, 2022b) and uniform-in-time propagation of chaos (Chen et al., 2022; Suzuki et al., 2022; 2023a; Kook et al., 2024; Nitanda, 2024; Chewi et al., 2024) have been established.

A recent work (Takakura & Suzuki, 2024) considered a two-timescale MFLD where the second layer is optimized infinitely faster than the first layer, and provided statistical guarantees for learning

Barron spaces with a bounded activation function. The concurrent work of Wang et al. (2024) studied this two-timescale approach to MFLD in a more general setting of optimization over signed measures without considering the estimation aspect and statistical guarantees. Our formulation here bypasses the need for two-timescale dynamics while learning a similarly large class of target functions.

**Learning low-dimensional targets.** The benefit of feature learning has also been studied in a "narrow-width" setting, where parameters of the neural network align with the low-dimensional target function during gradient-based training. Examples of low-dimensional targets include single-index models (Ben Arous et al., 2021; Ba et al., 2022; Bietti et al., 2022; Mousavi-Hosseini et al., 2023a; Damian et al., 2023; Lee et al., 2024) and multi-index models (Damian et al., 2022; Abbe et al., 2022; 2023; Dandi et al., 2023; Bietti et al., 2023; Collins-Woodfin et al., 2023; Vural & Erdogdu, 2024). While the information-theoretic threshold for learning such functions is $n \gtrsim d$ (for isotropic data) (Mondelli & Montanari, 2018; Barbier et al., 2019; Damian et al., 2024), the complexity of gradient-based learning is governed by properties of the link function. For instance, in the single-index setting, prior works established a sufficient sample size of $n \gtrsim d^{\Theta(s)}$ where $s$ is the *information exponent* for one-pass SGD on the squared/correlation loss (Dudeja & Hsu, 2018; Ben Arous et al., 2021; Bietti et al., 2022; Damian et al., 2023), and the *generative exponent* (Damian et al., 2024) when the algorithm can reuse samples or access a different loss (Dandi et al., 2024; Lee et al., 2024; Arnaboldi et al., 2024; Joshi et al., 2024). This presents a gap between the information-theoretically achievable sample complexity and the performance of neural networks optimized by gradient descent.

**Notation.** We denote the Euclidean inner product with $\langle \cdot, \cdot \rangle$, the Euclidean norm for vectors and the operator norm for matrices with $\|\cdot\|$, and the Frobenius norm with $\|\cdot\|_{\mathrm{F}}$. Given a topological space $\mathcal{W}$ endowed with an underlying metric and Lebesgue measure, we use $\mathcal{P}(\mathcal{W})$, $\mathcal{P}_2(\mathcal{W})$, and $\mathcal{P}_2^{\mathrm{ac}}(\mathcal{W})$ to denote the set of (Borel) probability measures, the set of probability measures with finite second moment, and the set of absolutely continuous probability measures with finite second moment, respectively. Finally, we use $\delta_{\boldsymbol{w}_0}$ to denote the Dirac measure at $\boldsymbol{w}_0$, i.e. $\int h(\boldsymbol{w}) \mathrm{d}\delta_{\boldsymbol{w}_0}(\boldsymbol{w}) = h(\boldsymbol{w}_0)$.

## 2 PRELIMINARIES: OPTIMIZATION IN MEASURE SPACE

**Statistical model.** In this paper, we consider the regression setting where the input $\boldsymbol{x} \in \mathbb{R}^d$ is generated from some distribution and the response $y \in \mathbb{R}$ is given by the following multi-index model

$$y = g\left( \frac{\langle \boldsymbol{u}_1, \boldsymbol{x} \rangle}{\sqrt{k}}, \dots, \frac{\langle \boldsymbol{u}_k, \boldsymbol{x} \rangle}{\sqrt{k}} \right) + \xi. \tag{2.1}$$

Here, $g : \mathbb{R}^k \to \mathbb{R}$ is the unknown link function, $\xi$ is a zero-mean $\varsigma$-subGaussian noise independent from $\boldsymbol{x}$; for simplicity, we assume $\varsigma^2 \lesssim 1$. Without loss of generality, we assume that the unknown directions $\boldsymbol{u}_1, \dots, \boldsymbol{u}_k$ are orthonormal, and define $\boldsymbol{U} = (\boldsymbol{u}_1/\sqrt{k}, \dots, \boldsymbol{u}_k/\sqrt{k})^\top \in \mathbb{R}^{k \times d}$; thus, we can use the shorthand notation $y = g(\boldsymbol{U}\boldsymbol{x}) + \xi$. Throughout the paper, we consider the setting $k \ll d$, and treat $k$ as an absolute constant independent from the ambient input dimension $d$.

For a student model $\boldsymbol{x} \to \hat{y}(\boldsymbol{x}; \boldsymbol{W})$ with $\boldsymbol{W}$ denoting its model parameters, we consider loss functions of the form $\ell(\hat{y}, y) = \rho(\hat{y} - y)$ where $\rho : \mathbb{R} \to \mathbb{R}_+$ is convex. In the classical regression setting where we observe $n$ i.i.d. samples $\{(\boldsymbol{x}^{(i)}, y^{(i)})\}_{i=1}^n$ from the data distribution, the regularized population risk and the regularized empirical risk are defined respectively as

$$J_\lambda(\boldsymbol{W}) := \mathbb{E}[\ell(\hat{y}(\boldsymbol{x}; \boldsymbol{W}), y)] + \frac{\lambda}{2} R(\boldsymbol{W}) \quad \text{and} \quad \hat{J}_\lambda(\boldsymbol{W}) := \frac{1}{n} \sum_{i=1}^n \ell(\hat{y}(\boldsymbol{x}^{(i)}; \boldsymbol{W}), y^{(i)}) + \frac{\lambda}{2} R(\boldsymbol{W}),$$

where $R$ is some regularizer on the model parameters and the expectation is over the joint distribution of $(\boldsymbol{x}, y)$. In practice, we minimize the empirical risk $\hat{J}_\lambda$ as the finite sample approximation of the population risk $J_\lambda$, anticipating that both minimizers are *close* to each other.

We use a two-layer neural network coupled with $\ell_2$ regularization to learn the statistical model (2.1), where learning constitutes recovering both unknowns $\boldsymbol{U}$ and $g$. Denoting the $m$ neurons with a matrix $\boldsymbol{W} := (\boldsymbol{w}_1, \dots, \boldsymbol{w}_m)^\top$, the student model and the $\ell_2$-regularizer are given as

$$\hat{y}_m(\boldsymbol{x}; \boldsymbol{W}) := \frac{1}{m} \sum_{j=1}^m \Psi(\boldsymbol{x}; \boldsymbol{w}_j) \quad \text{and} \quad R(\boldsymbol{W}) := \frac{1}{m} \|\boldsymbol{W}\|_{\mathrm{F}}^2 = \frac{1}{m} \sum_{j=1}^m \|\boldsymbol{w}_j\|^2, \tag{2.2}$$

where $\Psi : \mathbb{R}^d \times \mathcal{W} \to \mathbb{R}$ is the activation function, and $\boldsymbol{w}_j \in \mathcal{W}$ with $\mathcal{W}$ denoting a Riemannian manifold. In this formulation, the second layer weights are all fixed to be $+1$.

To minimize an objective $J$ denoting either $J_\lambda$ or $\hat{J}_\lambda$, we will consider a discretization of the following set of SDEs, which essentially define an interacting particle system over $m$ neurons:

$$\mathrm{d}\boldsymbol{w}_j^t = -m\nabla_{\boldsymbol{w}_j}J(\boldsymbol{w}_1^t,\ldots,\boldsymbol{w}_m^t)\mathrm{d}t + \sqrt{2\beta^{-1}}\mathrm{d}\boldsymbol{B}_t^j \quad \text{for} \quad 1 \le j \le m, \tag{2.3}$$

where $\nabla_{\boldsymbol{w}}$ is the Riemannian gradient and $(\boldsymbol{B}_t^j)_{j=1}^m$ is a set of independent Brownian motions on $\mathcal{W}$. We scale the learning rate by $m$ to compensate for the fact that the gradient will be of order $m^{-1}$ with respect to each neuron. The case $\beta = \infty$ corresponds to the classical gradient flow over $J$, while the Brownian noise can help escaping from spurious local minima and saddle points.

**Optimization in measure space.** Notice that the neural network and the regularizer in (2.2) are both invariant under permutations of the weights $(\boldsymbol{w}_1, \ldots, \boldsymbol{w}_m)$; thus, an equivalent integral representation of these functions can be written using Dirac measures $\delta_{\boldsymbol{w}_j}$ centered at $\boldsymbol{w}_j$, namely

$$\hat{y}(\boldsymbol{x}; \mu_{\boldsymbol{W}}) := \int \Psi(\boldsymbol{x}; \cdot)\mathrm{d}\mu_{\boldsymbol{W}} \quad \text{and} \quad \mathcal{R}(\mu_{\boldsymbol{W}}) := \int \|\cdot\|^2 \mathrm{d}\mu_{\boldsymbol{W}} \quad \text{with} \quad \mu_{\boldsymbol{W}} = \frac{1}{m}\sum_{j=1}^m \delta_{\boldsymbol{w}_j}. \tag{2.4}$$

Here, $\mu_{\boldsymbol{W}}$ is the empirical measure supported on $m$ atoms. Indeed, $\hat{y}(\boldsymbol{x}; \mu_{\boldsymbol{W}}) = \hat{y}_m(\boldsymbol{x}; \boldsymbol{W})$ and $\mathcal{R}(\mu_{\boldsymbol{W}}) = R(\boldsymbol{W})$, and this formulation allows extension to infinite-width networks by removing the condition that measures are supported on $m$ atoms, and by expanding the feasible set of measures to $\mu \in \mathcal{P}_2(\mathcal{W})$. Thus, we rewrite the population and the empirical risks in the space of measures as

$$\mathcal{J}_\lambda(\mu_{\boldsymbol{W}}) := J_\lambda(\boldsymbol{W}) \quad \text{and} \quad \hat{\mathcal{J}}_\lambda(\mu_{\boldsymbol{W}}) := \hat{J}_\lambda(\boldsymbol{W}),$$

with domain $\mu \in \mathcal{P}_2(\mathcal{W})$. Let $\mathcal{J} : \mathcal{P}_2(\mathcal{W}) \to \mathbb{R}$ be the population risk $\mathcal{J}_\lambda$ or the empirical risk $\hat{\mathcal{J}}_\lambda$. We can equivalently state the interacting SDE system (2.3) as (see e.g. (Chizat, 2022b, Prop. 2.4))

$$\mathrm{d}\boldsymbol{w}_j^t = -\nabla_{\boldsymbol{w}}\mathcal{J}'[\mu_{\boldsymbol{W}^t}](\boldsymbol{w}_j^t)\mathrm{d}t + \sqrt{2\beta^{-1}}\mathrm{d}\boldsymbol{B}_t^j \quad \text{for} \quad 1 \le j \le m, \tag{2.5}$$

where $\mathcal{J}'[\mu] \in L^2(\mathcal{W})$ denotes the first variation of $\mathcal{J}(\mu)$ defined via

$$\int \mathcal{J}'[\mu](\boldsymbol{w})\mathrm{d}(\nu - \mu)(\boldsymbol{w}) = \lim_{\epsilon \downarrow 0} \frac{\mathcal{J}((1-\epsilon)\mu + \epsilon\nu) - \mathcal{J}(\mu)}{\epsilon}, \quad \forall \nu \in \mathcal{P}_2(\mathcal{W}), \tag{2.6}$$

which is unique up to additive constants when it exists (Santambrogio, 2015, Definition 7.12).

As $m \to \infty$, the empirical measure $\mu_{\boldsymbol{W}^t}$ weakly converges to a deterministic measure $\mu_t$ for all fixed $t$, a phenomenon known as the *propagation of chaos* (Sznitman, 1991). Furthermore, $\mu_t$ can be characterized as the law of the solution of the following SDE and non-linear Fokker-Planck equation

$$\mathrm{d}\boldsymbol{w}^t = -\nabla_{\boldsymbol{w}}\mathcal{J}'[\mu_t](\boldsymbol{w}^t)\mathrm{d}t + \sqrt{2\beta^{-1}}\mathrm{d}\boldsymbol{B}_t \quad \text{and} \quad \partial_t\mu_t = \boldsymbol{\nabla} \cdot (\mu_t\nabla\mathcal{J}'[\mu_t]) + \beta^{-1}\boldsymbol{\Delta}\mu_t, \tag{2.7}$$

where $\boldsymbol{\nabla}\cdot$ and $\boldsymbol{\Delta}$ are the Riemannian divergence and Laplacian operators, respectively. Due to the existence of mean-field interactions, (2.7) is known as the *mean-field Langevin dynamics* (MFLD).

For a pair of probability measures $\mu \ll \nu$ both in $\mathcal{P}(\mathcal{W})$, we define the relative entropy $\mathcal{H}(\mu \,|\, \nu)$ and the relative Fisher information $\mathcal{I}(\mu \,|\, \nu)$ respectively as

$$\mathcal{H}(\mu \,|\, \nu) := \int_{\mathcal{W}} \ln \frac{\mathrm{d}\mu}{\mathrm{d}\nu}\mathrm{d}\mu \quad \text{and} \quad \mathcal{I}(\mu \,|\, \nu) := \int_{\mathcal{W}} \left\|\nabla \ln \frac{\mathrm{d}\mu}{\mathrm{d}\nu}\right\|^2 \mathrm{d}\mu. \tag{2.8}$$

It is well-known at this point that $\mu_t$ in (2.7) can be interpreted as the Wasserstein gradient flow of the entropic regularized functional $\mathcal{F}_\beta(\mu) := \mathcal{J}(\mu) + \frac{1}{\beta}\mathcal{H}(\mu \,|\, \tau)$, where $\tau$ is the uniform measure on compact $\mathcal{W}$ or the Lebesgue measure on a Euclidean space (Jordan et al., 1998; Ambrosio et al., 2005; Villani, 2009). For this gradient flow to converge exponentially fast towards the minimizer $\mu_\beta^* := \arg\min_\mu \mathcal{F}_\beta(\mu)$, we require a *gradient domination* condition on $\mu_\beta^*$ in the space of probability measures, given as

$$\mathcal{H}(\mu \,|\, \mu_\beta^*) \le \frac{C_{\mathrm{LSI}}}{2}\mathcal{I}(\mu \,|\, \mu_\beta^*), \quad \forall \mu \in \mathcal{P}(\mathcal{W}), \tag{2.9}$$

which is referred to as the log-Sobolev inequality (LSI). If the measure $\mathrm{d}\nu_{\mu_t} \propto \exp(-\beta\mathcal{J}'[\mu_t])\mathrm{d}\tau$ satisfies LSI with constant $C_{\mathrm{LSI}}$ for all $t \ge 0$, $\mu_t$ enjoys the following exponential convergence

$$\mathcal{F}_\beta(\mu_t) - \mathcal{F}_\beta(\mu_\beta^*) \le e^{\frac{-2t}{\beta C_{\mathrm{LSI}}}}(\mathcal{F}_\beta(\mu_0) - \mathcal{F}_\beta(\mu_\beta^*)); \tag{2.10}$$

see e.g. (Chizat, 2022b, Theorem 3.2) and (Nitanda et al., 2022, Theorem 1).

## 3 LEARNING MULTI-INDEX MODELS IN THE EUCLIDEAN SETTING

In this section, we consider learning multi-index models in the Euclidean setting. For technical reasons, we use an approximation of ReLU denoted by $z \mapsto \phi_{\kappa,\iota}(z)$ for some $\kappa, \iota > 1$, which is given by $\phi_{\kappa,\iota}(z) = \kappa^{-1} \ln(1 + \exp(\kappa z))$ for $z \in (-\infty, \iota/2]$ and extended on $(\iota/2, \infty)$ such that $\phi_{\kappa,\iota}$ is $C^2$ smooth, $|\phi_{\kappa,\iota}| \leq \iota$, $|\phi'_{\kappa,\iota}| \leq 1$, and $|\phi''_{\kappa,\iota}| \leq \kappa$. Note that $\phi_{\kappa,\iota}$ recovers ReLU as $\kappa, \iota \to \infty$. Recall that we freeze the second-layer weights at $+1$. Consequently, non-negative activations can only learn non-negative functions. To alleviate this, we choose $\mathcal{W} = \mathbb{R}^{2d+2}$, and use the notation $\boldsymbol{w} = (\boldsymbol{\omega}_1^\top, \boldsymbol{\omega}_2^\top)^\top$ with $\boldsymbol{\omega}_1, \boldsymbol{\omega}_2 \in \mathbb{R}^{d+1}$ to denote the first and the second half of weight coordinates, and use the activation function

$$\Psi(\boldsymbol{x}; \boldsymbol{w}) \coloneqq \phi_{\kappa,\iota}(\langle \tilde{\boldsymbol{x}}, \boldsymbol{\omega}_1 \rangle) - \phi_{\kappa,\iota}(\langle \tilde{\boldsymbol{x}}, \boldsymbol{\omega}_2 \rangle), \tag{3.1}$$

where $\tilde{\boldsymbol{x}} \coloneqq (\boldsymbol{x}, \tilde{r}_x)^\top \in \mathbb{R}^{d+1}$ for a constant $\tilde{r}_x$ corresponding to a bias unit. The above can also be seen as a 2-layer neural network with activation $\phi_{\kappa,\iota}$ and second-layer weights frozen at $\pm 1$.

We use the neural network and the regularizer in (2.2) with weights $\boldsymbol{W} \coloneqq (\boldsymbol{w}_1, \ldots, \boldsymbol{w}_m)$, and minimize the resulting empirical risk $\hat{J}_\lambda(\boldsymbol{W})$ via the *mean-field Langevin algorithm* (MFLA), which is a simple time discretization of (2.3) with the stepsize $\eta$ and the number of iterations $l > 0$,

$$\boldsymbol{w}_j^{l+1} = \boldsymbol{w}_j^l - m\eta \nabla_{\boldsymbol{w}_j} \hat{J}_\lambda(\boldsymbol{W}) + \sqrt{2\eta\beta^{-1}} \boldsymbol{\xi}_j^l, \quad 1 \leq j \leq m, \tag{3.2}$$

where $\boldsymbol{\xi}_j^l$ are independent standard Gaussian random vectors. When the stepsize is sufficiently small, MFLA approximately tracks the system of continuous-time SDEs (2.3) as well as their equivalent formulation in the measure space (2.5). If, in addition, the network width $m$ is sufficiently large, propagation of chaos will kick in and the dynamics will be an approximation to MFLD (2.7), ultimately minimizing the corresponding entropic regularized objective $\mathcal{F}_{\beta,\lambda}(\mu) \coloneqq \hat{\mathcal{J}}_\lambda(\mu) + \frac{1}{\beta}\mathcal{H}(\mu \,|\, \tau)$.

We make the following assumption on the input distribution.

**Assumption 1.** *The input $\boldsymbol{x}$ has zero mean and covariance $\boldsymbol{\Sigma}$. Further, $\|\boldsymbol{x}\|$ and $\|\boldsymbol{U}\boldsymbol{x}\|$ are subGaussian with respective norms $\sigma_n\|\boldsymbol{\Sigma}^{1/2}\|_{\mathrm{F}}$ and $\sigma_u\|\boldsymbol{\Sigma}^{1/2}\boldsymbol{U}^\top\|_{\mathrm{F}}$ for some absolute constants $\sigma_n, \sigma_u$.*

One example for the above assumption is the Gaussian case $\boldsymbol{x} \sim \mathcal{N}(0, \boldsymbol{\Sigma})$ where $\|\boldsymbol{A}\boldsymbol{x}\|$ is subGaussian with norm $\|\boldsymbol{\Sigma}^{1/2}\boldsymbol{A}\|$ for any matrix $\boldsymbol{A}$. In settings we consider, we can replace the operator norm with the Frobenius norm to obtain a weaker assumption, since $\|\boldsymbol{A}\boldsymbol{x}\|$ is roughly concentrated near its mean, scaling with $\|\boldsymbol{\Sigma}^{1/2}\boldsymbol{A}^\top\|_{\mathrm{F}}$. Assumption 1 can cover much broader settings than Gaussianity, e.g. it is satisfied when $\boldsymbol{x} = \boldsymbol{\Sigma}^{1/2}\boldsymbol{z}$ and $\boldsymbol{z}$ has mean-zero i.i.d. subGaussian coordinates (Vershynin, 2018, Theorem 6.3.2.). Without loss of generality, we will consider a scaling where $\|\boldsymbol{\Sigma}\| \lesssim 1$.

A key quantity in our analysis is the *effective dimension* which governs the algorithmic guarantees.

**Definition 1** (Effective dimension). *Define $d_{\mathrm{eff}} \coloneqq c_x^2/r_x^2$ where $c_x \coloneqq \mathrm{tr}(\boldsymbol{\Sigma})^{1/2}$, $r_x \coloneqq \|\boldsymbol{\Sigma}^{1/2}\boldsymbol{U}^\top\|_{\mathrm{F}}$.*

The effective dimension $d_{\mathrm{eff}}$ can be significantly smaller than the ambient dimension $d$, leading to particularly favorable results in the following when $d_{\mathrm{eff}} = \mathrm{polylog}(d)$. This concept has numerous applications from learning theory to statistical estimation; see e.g. Vershynin (2018); Wainwright (2019); Ghorbani et al. (2020); Ba et al. (2023). In covariance estimation, for example, the effective dimension is typically defined as $\mathrm{tr}(\boldsymbol{\Sigma})/\|\boldsymbol{\Sigma}\|$ (e.g. (Wainwright, 2019, Example 6.4)), which is equivalent to $d_{\mathrm{eff}}$ in Definition 1 provided that $\boldsymbol{U}$ lives in the top eigenspace of $\boldsymbol{\Sigma}$. However, in general, $d_{\mathrm{eff}}$ might be larger than $\mathrm{tr}(\boldsymbol{\Sigma})/\|\boldsymbol{\Sigma}\|$, which is expected as one can imagine a supervised learning setup where the variations of $\boldsymbol{x}$ provide very little information about target directions $\boldsymbol{U}$, making estimation more difficult. We make the following assumption on the link function in (2.1).

**Assumption 2.** *The link function is locally Lipschitz: $|g(\boldsymbol{z}_1) - g(\boldsymbol{z}_2)| \leq L\|\boldsymbol{z}_1 - \boldsymbol{z}_2\|$ for $\boldsymbol{z}_1, \boldsymbol{z}_2 \in \mathbb{R}^k$ satisfying $\|\boldsymbol{z}_1\| \vee \|\boldsymbol{z}_2\| \leq \tilde{r}_x \coloneqq r_x(1 + \sigma_u\sqrt{2(q+1)\ln(n)})$ for some $q > 0$ and $L = \mathcal{O}(1/r_x)$. We also assume $\mathbb{E}[y^2] \lesssim 1$.*

Note that the above Lipschitz condition is only local, thus allowing polynomially growing link functions $g$. We scale the Lipschitz constant with $1/r_x$ to make sure $y$ has a variance of order $\Theta(1)$.

Recall from Section 2 that in order to prove convergence of the MFLD (and its time/particle discretization MFLA), it is sufficient for the Gibbs potential $\nu_{\mu_{\boldsymbol{W}_t}} \propto \exp(-\beta\hat{\mathcal{J}}'_\lambda[\mu_{\boldsymbol{W}_t}])$ to satisfy LSI

uniformly along its trajectory. Here, it is straightforward to derive the first variation as

$$\hat{\mathcal{J}}'_\lambda[\mu](\boldsymbol{w}) = \hat{\mathcal{J}}'_0[\mu](\boldsymbol{w}) + \frac{\lambda}{2}\|\boldsymbol{w}\|^2 \quad \text{with} \quad \hat{\mathcal{J}}'_0[\mu](\boldsymbol{w}) = \frac{1}{n}\sum_{i=1}^n \rho'(\hat{y}(\boldsymbol{x}^{(i)};\mu) - y^{(i)})\Psi(\boldsymbol{x}^{(i)};\boldsymbol{w}). \quad (3.3)$$

The following assumption introduces the uniform LSI constant for the trajectory of MFLA.

**Assumption 3.** *Let $\boldsymbol{W}^l = (\boldsymbol{w}_1^l, \ldots, \boldsymbol{w}_m^l)$ denote the trajectory of MFLA. We assume the measure $\nu_{\mu_{\boldsymbol{W}^l}} \propto \exp(-\beta\hat{\mathcal{J}}'_\lambda[\mu_{\boldsymbol{W}^l}])$ satisfies the LSI (2.9) with constant $C_{\mathrm{LSI}}$ for all $l \geq 0$, and $C_{\mathrm{LSI}} \geq \beta$.*

The above condition is stated to simplify the exposition and will be verified in our results by using the boundedness of $\phi_{\kappa,\iota}$; $\hat{\mathcal{J}}'_\lambda[\mu_{\boldsymbol{W}^l}]$ can be considered as a bounded perturbation of a strongly convex potential, thus satisfies LSI by the Holley-Stroock argument (Holley & Stroock, 1986).

**Proposition 2.** *Suppose $\rho$ is $C_\rho$-Lipschitz. Then for any $\mu \in \mathcal{P}_2(\mathbb{R}^{2d+2})$, the probability measure $\nu_\mu \propto \exp(-\beta\hat{\mathcal{J}}'_\lambda[\mu])$ with $\hat{\mathcal{J}}'_\lambda$ given by (3.3) satisfies the LSI (2.9) with constant*

$$C_{\mathrm{LSI}} \leq \frac{1}{\beta\lambda}\exp(4C_\rho\iota\beta). \quad (3.4)$$

For the squared loss, we can replace $C_\rho$ above with $\hat{\mathcal{J}}_0(\mu_{\boldsymbol{W}})^{1/2}$. With this, as $\hat{\mathcal{J}}_0(\mu_{\boldsymbol{W}})$ is uniformly bounded along the trajectory, convergence of the infinite-width MFLD can be established. However, for the finite-width MFLA, controlling $\hat{\mathcal{J}}_0(\mu_{\boldsymbol{W}})$ is challenging as there is non-trivial probability that neurons incur a large loss, which is why we require Lipschitz $\rho$. Note that the right hand side of (3.4) is independent of $\kappa$; thus, by letting $\kappa \to \infty$, the proposition implies the same LSI constant for a bounded variant of ReLU. However, for MFLA (the time discretization of MFLD), we additionally require smoothness of the activation.

### 3.1 STATISTICAL AND COMPUTATIONAL COMPLEXITY OF MFLA

The main result of this section is stated in the following theorem.

**Theorem 3.** *Under Assumptions 1, 2 and 3, consider MFLA (3.2) with parameters $\lambda = \tilde{\lambda}r_x^2$, $\beta = \tilde{\Theta}(d_{\mathrm{eff}}/\tilde{\lambda})$, and $\eta \leq \tilde{\mathcal{O}}\big(\frac{1}{C_{\mathrm{LSI}}\kappa^2\bar{r}_x^4(d+\bar{r}_x^2/\lambda)}\big)$, where $\bar{r}_x := \|\boldsymbol{\Sigma}\| \vee \tilde{r}_x$. Suppose $\tilde{\lambda}, \kappa^{-1} = o_n(1)$, $\iota = \Theta\big(\frac{\tilde{r}_x^2}{\tilde{\lambda}r_x^2}\big)$, the loss satisfies $|\rho'| \vee \rho'' \lesssim 1$, and the algorithm is initialized with the weights sampled i.i.d. from some distribution $\boldsymbol{w}_j^0 \sim \mu_0$ with $\mathbb{E}\big[\|\boldsymbol{w}_j^0\|_2^2\big] \lesssim 1$. Then, with the number of samples $n$, the number of neurons $m$, and the number of iterations $l$ that can respectively be bounded by*

$$n = \tilde{\mathcal{O}}(d_{\mathrm{eff}}), \quad m = \tilde{\mathcal{O}}\big(\frac{C_{\mathrm{LSI}}\bar{r}_x^4\kappa^2}{\beta\lambda}\big(\frac{d}{\beta} + \frac{\bar{r}_x^2}{\lambda}\big)\big), \quad l = \tilde{\mathcal{O}}\big(\frac{C_{\mathrm{LSI}}\beta}{\eta}\big), \quad (3.5)$$

*with probability at least $1 - \mathcal{O}(n^{-q})$ for some $q > 0$, the excess risk satisfies*

$$\mathbb{E}_{\boldsymbol{W}^l}\mathbb{E}_{y,\boldsymbol{x}}[\rho(y - \hat{y}_m(\boldsymbol{x};\boldsymbol{W}^l))] - \mathbb{E}_\xi[\rho(\xi)] \leq o_n(1). \quad (3.6)$$

The above theorem demonstrates that $(i)$ the effective dimension of Definition 1 controls the sample complexity, and $(ii)$ the LSI constant of Assumption 3 controls the computational complexity. To that end, we can employ the LSI estimate of Proposition 2 to arrive at the following corollary.

**Corollary 4.** *In the setting of Theorem 3, using the LSI estimate of Proposition 2, with the number of samples, the number of neurons, and the number of iterations, respectively bounded by*

$$n = \tilde{\mathcal{O}}(d_{\mathrm{eff}}), \quad m = \tilde{\mathcal{O}}(de^{\tilde{\mathcal{O}}(d_{\mathrm{eff}})}), \quad l = \tilde{\mathcal{O}}(de^{\tilde{\mathcal{O}}(d_{\mathrm{eff}})}), \quad (3.7)$$

*MFLA can achieve the excess risk bound (3.6) with $\tilde{\lambda}^{-1}, \kappa = \mathrm{polylog}(n)$.*

We observe that the above corollary demonstrates a certain adaptivity to the effective low-dimensional structure, both in terms of *statistical* and *computational* complexity. Remarkably, this property of MFLA emerges without explicitly encoding any information about the covariance structure in the algorithm. In contrast, consider "fixed-grid" methods for optimization over the space of measures $\mathcal{P}(\mathbb{R}^{2d+2})$ (see Chizat (2022a) and references therein), in which the algorithm fixes the first-layer

| Work | Class of Targets | Sample Complexity | Input | Covariance | $d_{\text{eff}}$-adaptivity |
|------|------------------|-------------------|-------|-----------|------------------------------|
| Telgarsky (2023) | 2-parity | $d$ | hypercube | isotropic | ✗ |
| Suzuki et al. (2023b) | $k$-parity | $d$ | hypercube | isotropic | ✗ |
| Nitanda et al. (2024) | $k$-parity | $\text{tr}(\boldsymbol{\Sigma}) \sum_{i=1}^{k} \|\boldsymbol{\Sigma}^{1/2} \boldsymbol{u}_i\|^{-2}$ | parallelotope | full-rank | ✓ |
| Bach (2017) | multi-index | $d^{\frac{k+3}{2}}$ | bounded | general | ✗ |
| **Theorem 3** | multi-index | $\text{tr}(\boldsymbol{\Sigma})/\|\boldsymbol{\Sigma}^{1/2}\boldsymbol{U}^{\top}\|_{\text{F}}^2$ | subGaussian | general | ✓ |

Table 1: Learning guarantees of neural networks with exponential compute (we state the dimension dependence). Our Theorem 3 improves upon prior bounds, with a potentially significant gap depending on the problem setup.

of a two-layer network's representation and only trains the second-layer, solving a convex problem similar to the random features regression (Rahimi & Recht, 2007). However, fixed-grid methods do not show any type of adaptivity to low-dimensions, and in particular their computational complexity always scales exponentially with the ambient dimension $d$, unless information about the covariance structure is explicitly used when specifying the fixed representation.

Table 1 compares recent works in various aspects. Bach (2017) requires $d^{\frac{k+3}{2}}$ sample complexity for learning general $k$-index models, which is worse than the complexity $d_{\text{eff}}$ of Theorem 3 even in the worst case $d_{\text{eff}} = d$. The improvement in our bound is due to a refined control over $\|\boldsymbol{U}\boldsymbol{x}\|$; while Bach (2017) assumes this quantity scales with $\sqrt{d}$, it can be verified that for centered $\boldsymbol{x}$, its expectation is independent of $d$. Further, Bach (2017) does not provide a quantitative analysis of the optimization complexity, and it is not clear if their algorithm is adaptive to the covariance structure. Nitanda et al. (2024) studied learning $k$-sparse parities, a subclass of multi-index models we considered, for which it is considerably simpler to construct optimal neural networks with bounded activation. While the effective dimension (and the resulting sample complexity) of Nitanda et al. (2024) is not explicitly scale-invariant, we derive a scale-invariant translation of their bound in Appendix C, and show that it is always lower bounded by our effective dimension, especially when $\boldsymbol{\Sigma}$ is nearly rank-deficient.

**Remark.** We make the following remarks on the complexity of learning multi-index models.

- Even though the complexity in Corollary 4 scales exponentially with $d_{\text{eff}}$, in Section 3.2 we outline problem settings where $d_{\text{eff}} = \text{polylog}(d)$, under which it is possible to achieve quasipolynomial runtime for the MFLA. That said, the exponential dependence in $d_{\text{eff}}$ is unavoidable in general in LSI-based analysis (Menz & Schlichting, 2014), and is consequently present in the mean-field literature (Chizat, 2022b; Suzuki et al., 2023a;b).
- In the isotropic setting $\boldsymbol{\Sigma} = \mathbf{I}_d$, recent works have shown that certain variants of SGD can learn single-index polynomials with almost linear sample complexity (Dandi et al., 2024; Lee et al., 2024; Arnaboldi et al., 2024), which matches our sample complexity without needing exponential compute. However, these analyses crucially relied on the *polynomial* link function, which has *generative exponent* at most 2 (Damian et al., 2024) and is SQ-learnable with $n = \tilde{O}_d(d)$ samples (Mondelli & Montanari, 2018; Barbier et al., 2019; Chen & Meka, 2020). In contrast, our assumption on the link function allows for arbitrarily large generative exponent, and hence the computational lower bound in Damian et al. (2024) implies that achieving learnability in the $n \asymp d$ scaling requires exponential compute for statistical query learners.

### 3.2 UTILIZING THE EFFECTIVE DIMENSION

To better demonstrate the impact of effective dimension $d_{\text{eff}}$, we consider two covariance models.

**Spiked covariance.** We consider the spiked covariance model of Mousavi-Hosseini et al. (2023b). Namely, given a spike direction $\boldsymbol{\theta} \in \mathbb{S}^{d-1}$, suppose the covariance and the target directions satisfy

$$\boldsymbol{\Sigma} = (1+\alpha)^{-1}(\mathbf{I}_d + \alpha\boldsymbol{\theta}\boldsymbol{\theta}^{\top}), \quad \alpha \asymp d^{\gamma_2}, \quad \|\boldsymbol{U}\boldsymbol{\theta}\| \asymp d^{-\gamma_1}, \quad \gamma_2 \in [0,1], \quad \gamma_1 \in [0,1/2]. \quad (3.8)$$

Note that in high-dimensional settings, $\gamma_1 = 1/2$ corresponds to a regime where $\boldsymbol{\theta}$ is sampled uniformly over $\mathbb{S}^{d-1}$, whereas $\gamma_1 = 0$ corresponds to the case where $\boldsymbol{\theta}$ has a strong (perfect) correlation with $\boldsymbol{U}$. We only consider $\gamma_2 \leq 1$ since $\gamma_2 > 1$ corresponds to a setting where the input is effectively one-dimensional. In this setting, effective dimension depends on $\gamma_1$ and $\gamma_2$.

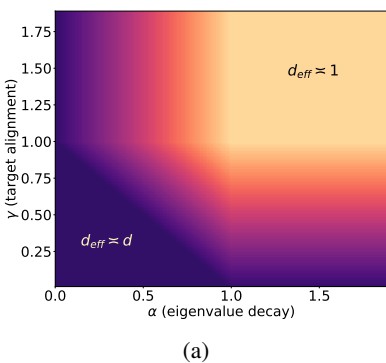
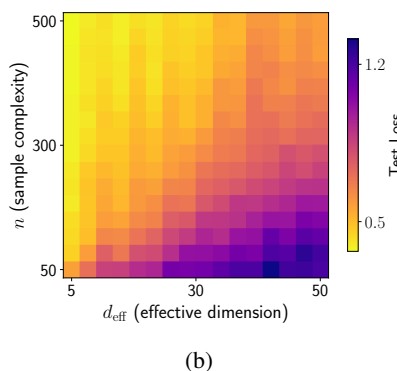

(a)                                                        (b)

Figure 1: (a) $d_{\text{eff}}$ according to Corollary 6. (b) Test loss from MFLA, details in Appendix E.

**Corollary 5.** *Under the spiked covariance model* (3.8)*, we have* $d_{\text{eff}} \asymp d^{1-\{(\gamma_2-2\gamma_1)\vee 0\}}$.

To get improvements over the isotropic effective dimension $d$, either the spike magnitude $\alpha$ or the spike-target alignment $\|U\theta\|$ needs to be sufficiently large so that $\gamma_2 > 2\gamma_1$. Recall that the effective dimension in the covariance estimation problem is $\text{tr}(\Sigma)/\|\Sigma\| \asymp d^{1-\gamma_2}$. Therefore, $d_{\text{eff}}$ in Corollary 5 only matches its unsupervised counterpart when $\gamma_1 = 0$, i.e. $\theta$ has a significant correlation with the target directions $U$. As $\gamma_2 \to 1$ and $\gamma_1 \to 0$, the effective dimension will be smaller than $\text{polylog}(d)$, leading to a computational complexity that is quasipolynomial in $d$.

**Scaling laws under power-law spectra.** Next, we consider a more general power-law decay for the eigenspectrum. Specifically, suppose $\Sigma = \sum_{i=1}^{d} \lambda_i \theta_i \theta_i^\top$ is the spectral decomposition of $\Sigma$, and

$$\frac{\lambda_i}{\lambda_1} \asymp i^{-\alpha}, \quad \frac{\|U\theta_i\|^2}{\|U\theta_1\|^2} \asymp i^{-\gamma}, \quad \text{for} \quad 1 \le i \le d, \tag{3.9}$$

for some absolute constants $\alpha, \gamma > 0$. Notice that $\sum_{i=1}^{d} \|U\theta_i\|^2 = \|U\|_{\text{F}}^2 = 1$. The following corollary characterizes $d_{\text{eff}}$ in terms of the parameters $\alpha$ and $\gamma$.

**Corollary 6.** *Under the power-law eigenspectrum for the covariance matrix* (3.9)*, we have*

$$d_{\text{eff}} \asymp \begin{cases} d^{1 \wedge (2-\alpha-\gamma)} & \alpha < 1, \gamma < 1 \\ d^{1-\alpha} & \alpha < 1, \gamma \ge 1, \\ d^{(1-\gamma) \vee 0} & \alpha \ge 1 \end{cases} \tag{3.10}$$

*where* $\asymp$ *above hides* $\text{polylog}(d)$ *dependencies.*

The scaling of $d_{\text{eff}}$ (hence the sample complexity) is illustrated in Figure 1a. We remark that the power-law assumption (3.9) is parallel to the *source condition* and *capacity condition* in the nonparametric regression literature (Cucker & Smale, 2002; Caponnetto & De Vito, 2007), where the capacity condition measures the decay of feature eigenvalues, and the source condition measures the alignment between the target and feature eigenvectors.

Also, based on (3.10), the width and number of iterations in Corollary 4 both become quasipolynomial in $d$ when $\alpha, \gamma \ge 1$. This corresponds to the setting where $\Sigma$ is approximately low-rank with most of its eigenspectrum concentrated in the first few principal components, and the corresponding eigenvectors are aligned with the row space of $U$.

## 4  POLYNOMIAL TIME CONVERGENCE IN THE RIEMANNIAN SETTING

The strong statistical learning guarantees in the previous section come at a computational price; MFLA may need $\exp(d_{\text{eff}})$ many iterations and neurons to converge. This complexity arises since in the worst case, the LSI constant that governs the convergence of MFLD will be exponential in the inverse temperature parameter $\beta$ (Menz & Schlichting, 2014). In this section, we provide

first steps towards achieving polynomial-time complexity for MFLD. In particular, we show that if we constrain the weight space to be a compact Riemannian manifold with a uniformly lower bounded Ricci curvature such as the hypersphere $\mathbb{S}^{d-1}$, we can establish a uniform LSI constant with polynomial dimension dependence, while the same set of assumptions in the Euclidean setting results in exponential dimension dependence. Notice that due to the manifold constraint on the weights, we no longer require $\ell_2$-regularization, and simply consider the objective $\mathcal{F}_\beta(\mu) = \hat{\mathcal{J}}_0(\mu) + \beta^{-1}\mathcal{H}(\mu \,|\, \tau)$.

Let $(\mathcal{W}, \mathfrak{g})$ be a $(d-1)$-dimensional compact Riemannian manifold with metric tensor $\mathfrak{g}$. We denote the Ricci curvature of $\mathcal{W}$ with $\mathrm{Ric}_{\mathfrak{g}}$. We recall the neural network $\hat{y}(\boldsymbol{x}; \mu) = \int \Psi(\boldsymbol{x}; \boldsymbol{w}) \mathrm{d}\mu(\boldsymbol{w})$ where, in this case, we choose a $C^2$-smooth activation $\Psi(\boldsymbol{x}; \cdot) : \mathcal{W} \to \mathbb{R}$ defined on the manifold. We consider the following model example to demonstrate our results.

**Example 7.** $\mathcal{W}$ is the hypersphere $\mathbb{S}^{d-1}$ equipped with its canonical metric tensor, and the activation is $\Psi(\boldsymbol{x}; \boldsymbol{w}) = \phi(\langle \boldsymbol{w}, \boldsymbol{x} \rangle)$ for some smooth $\phi : \mathbb{R} \to \mathbb{R}$. Suppose $|\phi'|, |\phi''| \lesssim 1$, and the distribution of $\boldsymbol{x}$ satisfies the conditions of Assumption 1.

The following assumption plays an important role in the analysis.

**Assumption 4.** $(\mathcal{W}, \mathfrak{g})$ *satisfies the curvature-dimension condition* $\mathrm{Ric}_{\mathfrak{g}} \succcurlyeq \varrho d\mathfrak{g}$ *for an absolute constant* $\varrho > 0$. *Further, there exists some* $\bar{\mu} \in \mathcal{P}(\mathcal{W})$ *such that* $\hat{\mathcal{J}}_0(\bar{\mu}) \le \bar{\varepsilon}$ *and* $\mathcal{H}(\bar{\mu} \,|\, \tau) \le \bar{\Delta}$ *for some constants* $\bar{\varepsilon}, \bar{\Delta}$, *where* $\tau$ *is the uniform distribution on* $\mathcal{W}$.

For the unit sphere $\mathbb{S}^{d-1}$, we have $\mathrm{Ric}_{\mathfrak{g}} \succcurlyeq (d-2)\mathfrak{g}$; thus, the curvature-dimension condition is satisfied for sufficiently large $d$. Moreover, if there exists some $\mu$ with $\hat{\mathcal{J}}_0(\mu) \le \bar{\varepsilon}$ (e.g. the minimizer of $\hat{\mathcal{J}}_0$) for which $\mathcal{H}(\mu \,|\, \tau) = \infty$, one can construct $\bar{\mu}$ such that $\hat{\mathcal{J}}_0(\bar{\mu}) \le \tilde{\mathcal{O}}(\bar{\varepsilon})$ and $\mathcal{H}(\bar{\mu} \,|\, \tau) \le \tilde{\mathcal{O}}(d)$, by smoothing $\mu$ via convolution with box kernels (see (Chizat, 2022a, Theorem 4.1) and its proof). Therefore in the worst-case, we have $\bar{\Delta} = \tilde{\Theta}(d)$. However, under a reasonable model assumption, we can verify Assumption 4 with $\bar{\Delta} = o(d)$, which is demonstrated in the below proposition.

**Proposition 8.** *Let* $y = \int \Psi(\boldsymbol{x}; \cdot) \mathrm{d}\mu^*$ *for some* $\mu^* \in \mathcal{P}(\mathcal{W})$ *such that* $\mathrm{d}\mu^* \propto e^f \mathrm{d}\tau$ *for* $f : \mathcal{W} \to \mathbb{R}$. *Then,* $\hat{\mathcal{J}}_0(\mu^*) = 0$ *and* $\mathcal{H}(\mu^* \,|\, \tau) \le \int f(\mathrm{d}\mu^* - \mathrm{d}\tau) \le \mathrm{osc}(f)$ *where* $\mathrm{osc}(f) := \sup f - \inf f$.

In the above result, the constants in Assumption 4 can be identified as $\bar{\varepsilon} = 0$ and $\bar{\Delta} = \mathrm{osc}(f)$ which is the oscillation of the log-density of $\mu^*$. Consequently, if the neurons in the teacher model are sufficiently present in all directions of the weight space, we get $\mathrm{osc}(f) = o(d)$; consider e.g. the extreme case $\mu^* = \tau$ which implies $f$ is constant. Interestingly, in the case of $k$-multi-index models, this condition implies that $k$ grows with dimension, ruling out the case $k = \mathcal{O}(1)$. We include a natural example of target functions of interest in the form of Proposition 8 in Appendix D.

For MFLD to converge to a minimizer of $\hat{\mathcal{J}}_0$, the parameter $\beta$ needs to satisfy $\beta \ge \tilde{\Omega}(\bar{\Delta})$ to ensure the entropic regularization is not the dominant term in the objective $\mathcal{F}_\beta$. In the Euclidean setting, this implies an LSI constant of order $\exp(\tilde{\mathcal{O}}(\bar{\Delta}))$, resulting in a computational complexity $\exp(\tilde{\mathcal{O}}(\bar{\Delta}))$ as shown in Theorem 3. In what follows, we demonstrate via the Bakry-Émery theory (Bakry & Émery, 1985) that in the Riemannian setting, under a uniform lower bound on the Ricci curvature, the LSI constant can be independent of $\bar{\Delta}$ and $d$ as long as we have $\bar{\Delta} = o(d)$. We include a natural example of target functions of interest in the form of Proposition 8 in Appendix D.

**Proposition 9.** *Suppose Assumption 4 holds and the loss* $\rho$ *is* $C_\rho$-*Lipschitz. Then, for all* $\mu \in \mathcal{P}(\mathcal{W})$ *and* $\beta < \varrho d/C_\rho K$, *the probability measure* $\nu_\mu \propto \exp(-\beta \hat{\mathcal{J}}_0'[\mu])$ *satisfies the LSI with constant*

$$C_{\mathrm{LSI}} \le (\varrho d - \beta C_\rho K)^{-1}, \tag{4.1}$$

*where* $K = \sup_{\|\boldsymbol{v}\|_{\mathfrak{g}} = 1} \mathbb{E}_{S_n}\left[\left|\langle \boldsymbol{v}, \nabla_{\boldsymbol{w}}^2 \Psi(\boldsymbol{x}; \boldsymbol{w}) \boldsymbol{v}\rangle\right|\right]$, *and* $\mathbb{E}_{S_n}[\cdot]$ *denotes the expectation under empirical data distribution over* $n$ *samples.*

**Remark.** In the setting of Example 7 with $n \ge \tilde{\Omega}(\mathrm{tr}(\boldsymbol{\Sigma})/\|\boldsymbol{\Sigma}\|)$, we have $K \lesssim \|\boldsymbol{\Sigma}\|$ with probability at least $1 - \mathcal{O}(n^{-q})$ for some constant $q > 0$. Consequently, the LSI constant is independent of $d$.

We can now present the following global convergence guarantee to the minimizer of $\mathcal{J}_0$ for large $d$.

**Theorem 10.** *Suppose Assumption 4 holds, and let* $K$ *be as in Proposition 9. Let* $(\mu_t)_{t \ge 0}$ *denote the law of the MFLD. For any* $\varepsilon > 0$, *let* $\beta = \bar{\Delta}/\varepsilon$ *and* $d \ge 2C_\rho K \bar{\Delta}/\varrho\varepsilon$. *Then, we have*

$$\hat{\mathcal{J}}_0(\mu_T) \lesssim \bar{\varepsilon} + \varepsilon, \quad \textit{whenever} \quad T \ge \frac{\bar{\Delta}}{\varepsilon\varrho d} \ln\left(\frac{\mathcal{F}_\beta(\mu_0)}{\varepsilon}\right). \tag{4.2}$$

*Moreover, in the setting of Example 7 and for a 1-Lipschitz loss function, if we have $d \gtrsim \bar{\Delta}/\varrho\varepsilon$ and $n \geq \Omega(\bar{\Delta}(1 + \bar{\varepsilon}/\varepsilon)/\varepsilon^2) \vee \tilde{\Omega}(\mathrm{tr}(\boldsymbol{\Sigma})/\|\boldsymbol{\Sigma}\|) \vee \tilde{\Omega}(1/\varepsilon^4)$, then*

$$\mathcal{J}_0(\mu_T) \lesssim \bar{\varepsilon} + \varepsilon, \quad whenever \quad T \geq \frac{\bar{\Delta}}{\varepsilon\varrho d} \ln\left(\frac{\mathcal{F}_\beta(\mu_0)}{\varepsilon}\right), \tag{4.3}$$

*with probability at least $1 - \mathcal{O}(n^{-q})$ over the randomness of data, for some constant $q > 0$.*

The sample complexity is controlled by the maximum of $\bar{\Delta}$ and $\mathrm{tr}(\boldsymbol{\Sigma})/\|\boldsymbol{\Sigma}\|$ up to log factors. We remark that dependence on $\varepsilon$ is not our main focus, and it may be possible to improve $1/\varepsilon^4$ with a more refined analysis. Remarkably, the time complexity improves in high dimensions, thanks to the effect of the Ricci curvature. While the above result is for the continuous-time infinite-width MFLD, the uniform-in-time propagation of chaos for MFLD strongly suggests that the cost of time/width discretizations will be polynomial, see e.g. Suzuki et al. (2023a) for the Euclidean setting, and Li & Erdogdu (2023) for the time-discretization of the Langevin diffusion on the hypersphere under LSI.

To compare the setting of this section to that of Section 3, as explored in Appendix A, we remark that the Euclidean $\ell_2$ and entropic regularizations can be combined into a single effective regularizer of the form $\beta^{-1}\mathcal{H}(\mu \,|\, \gamma)$, where $\gamma = \mathcal{N}(0, (\lambda\beta)^{-1}\mathbf{I}_{2d+2})$; therefore, in the Euclidean setting, $\gamma$ plays the role of $\tau$. Further in the proof of Lemma 20, we show that in the Euclidean setting, $\bar{\Delta} \asymp \lambda\beta/r_x^2$ and $\bar{\varepsilon} \asymp c_x/\sqrt{\lambda\beta}$; thus, to learn with any non-trivial accuracy, we have $\bar{\Delta} \asymp c_x^2/r_x^2 = d_{\mathrm{eff}}$. As discussed above, controlling the effect of entropic regularization necessitates $\beta \geq \tilde{\Omega}(\bar{\Delta})$. Unlike its Riemannian counterpart, the Euclidean LSI estimate of Proposition 2 scales with $\exp(\beta)$, ultimately resulting in a large computational gap between the two settings under the same $\bar{\Delta}$. This leaves open the question of whether $\bar{\Delta} \asymp d_{\mathrm{eff}}$ can be achievable in the Riemannian setting for $k$-multi-index models with $k = \mathcal{O}(1)$, which is an interesting direction for future exploration.

## 5 CONCLUSION

In this paper, we investigated the mean-field Langevin dynamics for learning multi-index models. We proved that the statistical and computational complexity of this problem can be characterized by an effective dimension which captures the low-dimensional structure in the input covariance, along with its correlation with the target directions. In particular, the sample complexity scales almost linearly with the effective dimension, while without additional assumptions, the computational complexity may scale exponentially with this quantity. Through this effective dimension, we showed both statistical and computational adaptivity of the MFLD to low-dimensions when training neural networks, outperforming rotationally invariant kernels and statistical query learners in terms of statistical complexity, and fixed-grid convex optimization methods in terms of computational complexity. Further, we studied conditions under which achieving a polynomial LSI in the inverse temperature, and subsequently a polynomial-in-$d$ runtime guarantee for the MFLD is possible. Specifically, we showed that under certain assumptions, which are verified for teacher models with diverse neurons, constraining the weights to a Riemannian manifold with positive Ricci curvature such as the hypersphere can lead to such polynomial dependence. In contrast, the same assumptions in the Euclidean setting result in an LSI constant scaling exponentially with the inverse temperature.

We conclude with some limitations of our work, along with future directions.

- Further assumptions are required to go beyond the current exponential computational complexity of the MFLD. We leave the study of such conditions as an important direction for future work.

- While we focused on $k = \mathcal{O}(1)$, the versatility of the MFLD analysis may allow us to let $k$ grow with dimension as in Ghorbani et al. (2019); Martin et al. (2023); Oko et al. (2024), or $g$ to exhibit a more complex hierarchical structure (Allen-Zhu & Li, 2020; Nichani et al., 2023). Learning these functions with the MFLD is an interesting direction for future research.

- Another important future direction is developing lower bounds for learning multi-index models with gradient-based methods, under more realistic assumptions (e.g., non-adversarial noise) than the statistical query setup. These lower bounds can highlight when exponential computation is inevitable for optimal sample complexity, and present rigorous information-computation tradeoffs.

ACKNOWLEDGMENTS

The authors thank Lénaïc Chizat, Mufan (Bill) Li, Fanghui Liu, and Taiji Suzuki for useful discussions. MAE was partially supported by the NSERC Grant [2019-06167], the CIFAR AI Chairs program, and the CIFAR Catalyst grant.

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

# A   PROOFS OF SECTION 3

Before presenting the layout of the proofs, we introduce a useful reformulation of the objective $\mathcal{F}_{\beta,\lambda}(\mu)$. Recall that

$$\mathcal{F}_{\beta,\lambda}(\mu) = \hat{\mathcal{J}}_0(\mu) + \frac{\lambda}{2}\mathcal{R}(\mu) + \frac{1}{\beta}\mathcal{H}(\mu).$$

Let $\gamma \propto \exp\left(\frac{-\lambda\beta}{2}\|\boldsymbol{w}\|^2\right)$ be the centered Gaussian measure on $\mathbb{R}^{2d+2}$ with variance $1/(\lambda\beta)$. Then, we can rewrite the above as

$$\mathcal{F}_{\beta,\lambda}(\mu) = \hat{\mathcal{J}}_0(\mu) + \frac{1}{\beta}\mathcal{H}(\mu\,|\,\gamma) + \frac{d}{2\beta}\ln\left(\frac{\lambda\beta}{2\pi}\right).$$

As a result, we can define

$$\tilde{\mathcal{F}}_{\beta,\lambda}(\mu) := \hat{\mathcal{J}}_0(\mu) + \frac{1}{\beta}\mathcal{H}(\mu\,|\,\gamma), \tag{A.1}$$

which is non-negative and equivalent to $\mathcal{F}_\beta$ up to an additive constant. Notice that

$$\mu_\beta^* := \arg\min_\mu \mathcal{F}_{\beta,\lambda}(\mu) = \arg\min_\mu \tilde{\mathcal{F}}_{\beta,\lambda}(\mu).$$

This reformulation, which was also used in Suzuki et al. (2023b), allows us to combine the effect of weight decay and entropic regularization into a single non-negative term $\mathcal{H}(\mu\,|\,\gamma)$. Furthermore, the simple density expression for the Gaussian measure $\gamma$ allows us to achieve useful estimates for $\mathcal{H}(\mu\,|\,\gamma)$. In particular, as we will show below, it is possible to control $\mathcal{H}(\mu_\beta^*\,|\,\gamma)$ with effective dimension rather than ambient dimension, which leads to dependence on $d_{\text{eff}}$ rather than $d$ in our bounds.

We break down the proof of Theorem 3 into three steps:

1. In Section A.2 we show that there exists a measure $\mu^* \in \mathcal{P}_2(\mathbb{R}^{2d+2})$ where $\hat{y}(\cdot;\mu^*)$ can approximate $g$ on the training set with bounds on $\mathcal{R}(\mu^*)$. This construction provides upper bounds on $\hat{\mathcal{J}}_0(\mu_\beta^*)$ and $\mathcal{H}(\mu_\beta^*\,|\,\gamma)$.

2. In Section A.3, given the bound on $\mathcal{H}(\mu_\beta^*\,|\,\gamma)$, we perform a generalization analysis via Rademacher complexity tools which leads to a bound on $\mathcal{J}_0(\mu_\beta^*)$.

3. Finally, in Section A.4, we estimate the LSI constant and constants related to smoothness/discretization along the trajectory, which imply that $\mathcal{F}_{\beta,\lambda}^m(\mu_l^m)$ converges to $\mathcal{F}_\beta(\mu_\beta^*)$, where $\mathcal{F}_{\beta,\lambda}^m$ is an adjusted objective over $\mathcal{P}(\mathbb{R}^{(2d+2)m})$ defined in (A.6). This bound implies the convergence of $\mathbb{E}_{\boldsymbol{W}\sim\mu_l^m}[J_0(\boldsymbol{W})]$ to $\mathcal{J}_0(\mu_\beta^*)$, which was bounded in the previous step.

Before laying out these steps, in Section A.1, we will introduce the required concentration results. In the following, we will use the unregularized population $\mathcal{J}_0(\mu) := \mathbb{E}[\ell(\hat{y}(\boldsymbol{x};\mu),y)]$ and empirical $\hat{\mathcal{J}}_0(\mu) = \mathbb{E}_{S_n}[\ell(\hat{y}(\boldsymbol{x};\mu),y)]$ risks, and also consider the finite-width versions $J_0(\boldsymbol{W}) := \mathcal{J}_0(\mu_{\boldsymbol{W}})$ and $\hat{J}_0(\boldsymbol{W}) := \hat{\mathcal{J}}_0(\mu_{\boldsymbol{W}})$. Additionally, we will use $\phi_\infty(z) := z \vee 0$ to denote the ReLU activation.

## A.1   CONCENTRATION BOUNDS

We begin by specifying the definition of subGaussian and subexponential random variables in our setting.

**Definition 11** (Wainwright (2019)). *A random variable $x$ is $\sigma$-subGaussian if $\mathbb{E}\left[e^{\lambda(x-\mathbb{E}[x])}\right] \leq e^{\lambda^2\sigma^2/2}$ for all $\lambda \in \mathbb{R}$, and is $(\nu,\alpha)$-subexponential if $\mathbb{E}\left[e^{\lambda(x-\mathbb{E}[x])}\right] \leq e^{\lambda^2\nu^2/2}$ for all $|\lambda| \leq 1/\alpha$. If $x$ is $\sigma$-subGaussian, then*

$$\mathbb{P}(x - \mathbb{E}[x] \geq t) \leq \exp\left(\frac{-t^2}{2\sigma^2}\right). \tag{A.2}$$

*If $x$ is $(\nu,\alpha)$-subexponential, then*

$$\mathbb{P}(x - \mathbb{E}[x] \geq t) \leq \exp\left(-\frac{1}{2}\min\left(\frac{t^2}{\nu^2}, \frac{t}{\alpha}\right)\right) \tag{A.3}$$

Moreover, for centered random variables, let $|\cdot|_{\psi_2}$ and $|\cdot|_{\psi_1}$ denote the subGaussian and subexponential norm respectively (Vershynin, 2018, Definitions 2.5.6 and 2.7.5). Then $x$ is $\sigma$-subGaussian if and only if $\sigma \asymp |x - \mathbb{E}[x]|_{\psi_2}$, and is $(\nu, \nu)$-subexponential if and only if $\nu \asymp |x - \mathbb{E}[x]|_{\psi_1}$.

Next, we bound several quantities that appear in various parts of our proofs.

**Lemma 12.** *Under Assumption 1, for any $q > 0$ and all $1 \le i \le n$, with probability at least $1 - n^{-q}$,*

$$\left\| \boldsymbol{U}\boldsymbol{x}^{(i)} \right\| \le r_x \left( 1 + \sigma_u \sqrt{2(q+1)\ln n} \right) = \tilde{r}_x. \tag{A.4}$$

**Proof.** By subGaussianity of $\|\boldsymbol{U}\boldsymbol{x}\|$ from Assumption 1 and the subGaussian tail bound, with probability at least $1 - n^{-q-1}$

$$\left\| \boldsymbol{U}\boldsymbol{x}^{(i)} \right\| \le \mathbb{E}[\|\boldsymbol{U}\boldsymbol{x}\|] + \sigma_u r_x \sqrt{2(q+1)\ln n}$$

$$\le r_x + \sigma_u r_x \sqrt{2(q+1)\ln n}.$$

The statement of lemma follows from a union bound over $1 \le i \le n$. $\qquad\square$

**Lemma 13.** *Under Assumption 1, we have $\mathbb{E}_{S_n}\left[\|\boldsymbol{x}\|^2\right] \lesssim c_x^2$ with probability at least $1 - \exp(-\Omega(n))$.*

**Proof.** By the triangle inequality,

$$\left|\|\boldsymbol{x}\|\right|_{\psi_2} \le \left|\|\boldsymbol{x}\| - \mathbb{E}[\|\boldsymbol{x}\|]\right|_{\psi_2} + \left|\mathbb{E}[\|\boldsymbol{x}\|]\right|_{\psi_2} \lesssim \sigma_n \left\|\boldsymbol{\Sigma}^{1/2}\right\|_{\mathrm{F}} + \mathrm{tr}(\boldsymbol{\Sigma})^{1/2} \lesssim \mathrm{tr}(\boldsymbol{\Sigma})^{1/2}.$$

Recall $c_x^2 := \mathrm{tr}(\boldsymbol{\Sigma})$. Furthermore, by (Vershynin, 2018, Lemma 2.7.6) we have

$$\left|\|\boldsymbol{x}\|^2\right|_{\psi_1} = \left|\|\boldsymbol{x}\|\right|_{\psi_2}^2 \lesssim c_x^2.$$

We arrive at a similar result for the centered random variable $\|\boldsymbol{x}\|^2 - \mathbb{E}[\|\boldsymbol{x}\|]^2 = \|\boldsymbol{x}\|^2 - c_x^2$. We conclude the proof by the subexponential tail inequality,

$$\mathbb{P}\left( \mathbb{E}_{S_n}\left[\|\boldsymbol{x}\|^2\right] - c_x^2 \ge t c_x^2 \right) \le \exp\left( -\min(t, t^2)\Omega(n) \right).$$

$\qquad\square$

**Lemma 14.** *Under Assumption 1, we have $\mathbb{E}_{S_n}\left[y^2\right] \lesssim 1$ with probability at least $1 - n^{-q}$.*

**Proof.** By the local Lipschitzness of $g$, on the event of Lemma 12, we have

$$|y|^2 \le 3g(0)^2 + 3\mathcal{O}(1/r_x^2)\|\boldsymbol{U}\boldsymbol{x}\|^2 + 3\xi^2.$$

By a similar argument to Lemma 13 we have

$$\left|\|\boldsymbol{U}\boldsymbol{x}\|^2\right|_{\psi_1} = \left|\|\boldsymbol{U}\boldsymbol{x}\|\right|_{\psi_2}^2 \le 2\left|\|\boldsymbol{U}\boldsymbol{x}\| - \mathbb{E}[\|\boldsymbol{U}\boldsymbol{x}\|]\right|_{\psi_2}^2 + 2\left|\mathbb{E}[\|\boldsymbol{U}\boldsymbol{x}\|]\right|^2 \lesssim (1 + \sigma_u^2)r_x^2,$$

since $\mathbb{E}\left[\|\boldsymbol{U}\boldsymbol{x}\|^2\right] = r_x^2$. As a result, by the subexponential tail bound,

$$\mathbb{E}_{S_n}\left[\|\boldsymbol{U}\boldsymbol{x}\|^2\right] - \mathbb{E}\left[\|\boldsymbol{U}\boldsymbol{x}\|^2\right] \lesssim (1 + \sigma_u^2)r_x^2 \lesssim r_x^2,$$

with probability at least $1 - \exp(-\Omega(n))$. Similarly, $\left|\xi^2\right|_{\psi_1} \le |\xi|_{\psi_2}^2 \lesssim \varsigma^2$, therefore,

$$\mathbb{E}_{S_n}\left[\xi^2\right] - \mathbb{E}\left[\xi^2\right] \lesssim \varsigma^2 \lesssim 1,$$

with probability at least $1 - \exp(-\Omega(n))$. The statement of the lemma follows by a union bound. $\quad\square$

**Lemma 15.** *Under Assumption 1, for any $q > 0$ and $n \gtrsim \frac{c_x^2}{\|\boldsymbol{\Sigma}\|}(1 + \sigma_n^2(q+1)\ln(n))\ln(dn^q)$, with probability at least $1 - \mathcal{O}(n^{-q})$ we have $\left\|\mathbb{E}_{S_n}\left[\boldsymbol{x}\boldsymbol{x}^\top\right]\right\| \lesssim \|\boldsymbol{\Sigma}\|$. Further, if $q \ge 1$, then $\mathbb{E}\left[\left\|\mathbb{E}_{S_n}\left[\boldsymbol{x}\boldsymbol{x}^\top\right]\right\|^{1/2}\right] \lesssim \|\boldsymbol{\Sigma}\|^{1/2}$.*

**Proof.** First, note that by subGaussianity of $\|x\|$, for every fixed $i$, we have with probability at least $1 - n^{-q-1}$,

$$\left\|\boldsymbol{x}^{(i)}\right\| - \mathbb{E}[\|\boldsymbol{x}\|] \leq \sigma_n \left\|\boldsymbol{\Sigma}^{1/2}\right\|_{\mathrm{F}} \sqrt{2(q+1)\ln n}.$$

Since $\mathbb{E}[\|\boldsymbol{x}\|] \leq c_x$, via a union bound, with probability at least $1 - n^{-q}$,

$$\left\|\boldsymbol{x}^{(i)}\right\| \leq c_x + \sigma_n c_x \sqrt{2(q+1)\ln n} =: \tilde{c}_x.$$

Define the clipped version of $\boldsymbol{x}$ via $\boldsymbol{x}_c = \boldsymbol{x}(1 \wedge \frac{\tilde{c}_x}{\|\boldsymbol{x}\|})$. Then, on the above event,

$$\mathbb{E}_{S_n}\!\left[\boldsymbol{x}\boldsymbol{x}^\top\right] = \mathbb{E}_{S_n}\!\left[\boldsymbol{x}_c\boldsymbol{x}_c^\top\right].$$

Moreover,

$$\left\|\mathbb{E}\!\left[\boldsymbol{x}_c\boldsymbol{x}_c^\top\right]\right\| = \sup_{\|\boldsymbol{v}\|\leq 1} \mathbb{E}\!\left[\langle \boldsymbol{x}_c, \boldsymbol{v}\rangle^2\right] \leq \sup_{\|\boldsymbol{v}\|\leq 1} \mathbb{E}\!\left[\langle \boldsymbol{x}, \boldsymbol{v}\rangle^2\right] = \left\|\mathbb{E}\!\left[\boldsymbol{x}\boldsymbol{x}^\top\right]\right\|.$$

Finally, by the covariance estimation bound of (Wainwright, 2019, Corollary 6.20) for centered subGaussian random vectors and the condition on $n$ given in the statement of the lemma,

$$\left\|\mathbb{E}_{S_n}\!\left[\boldsymbol{x}_c\boldsymbol{x}_c^\top\right]\right\| - \left\|\mathbb{E}\!\left[\boldsymbol{x}_c\boldsymbol{x}_c^\top\right]\right\| \lesssim \left\|\mathbb{E}\!\left[\boldsymbol{x}\boldsymbol{x}^\top\right]\right\|$$

with probability at least $1 - \mathcal{O}(n^{-q})$. Consequently, we have $\left\|\mathbb{E}_{S_n}\!\left[\boldsymbol{x}\boldsymbol{x}^\top\right]\right\| \lesssim \|\boldsymbol{\Sigma}\|$ with probability at least $1 - \mathcal{O}(n^{-q})$.

For the second part of the lemma, let $E$ denote the event on which the above $\left\|\mathbb{E}_{S_n}\!\left[\boldsymbol{x}\boldsymbol{x}^\top\right]\right\| \lesssim \|\boldsymbol{\Sigma}\|$ holds. Then,

$$\mathbb{E}\!\left[\left\|\mathbb{E}_{S_n}\!\left[\boldsymbol{x}\boldsymbol{x}^\top\right]\right\|^{1/2}\right] = \mathbb{E}\!\left[\mathbb{1}(E)\left\|\mathbb{E}_{S_n}\!\left[\boldsymbol{x}\boldsymbol{x}^\top\right]\right\|^{1/2}\right] + \mathbb{E}\!\left[\mathbb{1}(E^C)\left\|\mathbb{E}_{S_n}\!\left[\boldsymbol{x}\boldsymbol{x}^\top\right]\right\|^{1/2}\right]$$

$$\lesssim \|\boldsymbol{\Sigma}\|^{1/2} + \mathbb{P}\!\left(E^C\right)^{1/2} \mathbb{E}\!\left[\left\|\mathbb{E}_{S_n}\!\left[\boldsymbol{x}\boldsymbol{x}^\top\right]\right\|\right]^{1/2}$$

$$\lesssim \|\boldsymbol{\Sigma}\|^{1/2} + \mathcal{O}(n^{-q/2})c_x.$$

Suppose $q \geq 1$. Then for $n \gtrsim c_x^2/\|\boldsymbol{\Sigma}\|$, we have $\mathbb{E}\!\left[\left\|\mathbb{E}_{S_n}\!\left[\boldsymbol{x}\boldsymbol{x}^\top\right]\right\|^{1/2}\right] \lesssim \|\boldsymbol{\Sigma}\|^{1/2}$, which completes the proof. $\qquad\square$

We summarize the above results into a single event.

**Lemma 16.** *Suppose $n \gtrsim \frac{c_x^2}{\|\boldsymbol{\Sigma}\|}(1 + \sigma_n^2(q+1)\ln(n))\ln(dn^q)$. There exists an event $\mathcal{E}$ such that $\mathbb{P}(\mathcal{E}) \geq 1 - \mathcal{O}(n^{-q})$, and on $\mathcal{E}$:*

1. *$\left\|\boldsymbol{U}\boldsymbol{x}^{(i)}\right\| \leq \tilde{r}_x$ for all $1 \leq i \leq n$.*

2. *$\mathbb{E}_{S_n}\!\left[\|\boldsymbol{x}\|^2\right] \lesssim c_x^2$.*

3. *$\left\|\mathbb{E}_{S_n}\!\left[\boldsymbol{x}\boldsymbol{x}^\top\right]\right\| \lesssim \|\boldsymbol{\Sigma}\|$.*

4. *$\mathbb{E}\!\left[\left\|\mathbb{E}_{S_n}\!\left[\boldsymbol{x}\boldsymbol{x}^\top\right]\right\|^{1/2}\right] \lesssim \|\boldsymbol{\Sigma}\|^{1/2}$.*

5. *$\mathbb{E}_{S_n}\!\left[y^2\right] \lesssim 1$.*

We recall the variational lower bound for the KL divergence, which will be used at various stages of different proofs to relate certain expectations to the KL divergence.

**Lemma 17** (Donsker-Varadhan Variational Formula for KL Divergence (Donsker & Varadhan, 1983))**.** *Let $\mu$ and $\nu$ be probability measures on $\mathcal{W}$. Then,*

$$\mathcal{H}(\mu \,|\, \nu) = \sup_{f:\mathcal{W}\to\mathbb{R}} \int f\,\mathrm{d}\mu - \ln\!\left(\int e^f\,\mathrm{d}\nu\right).$$

Finally, we state the following lemma which will be useful in estimating smoothness constants in the convergence analysis.

**Lemma 18.** *Suppose $(z, \boldsymbol{x}) \in \mathbb{R} \times \mathbb{R}^d$ are drawn from a probability distribution $\mathcal{D}$. Then,*

$$\|\mathbb{E}_{\mathcal{D}}[z\boldsymbol{x}]\| \leq \sqrt{\mathbb{E}_{\mathcal{D}}[z^2]\|\mathbb{E}_{\mathcal{D}}[\boldsymbol{x}\boldsymbol{x}^\top]\|}.$$

**Proof.** We have

$$
\begin{aligned}
\|\mathbb{E}_{\mathcal{D}}[z\boldsymbol{x}]\| = \sup_{\|\boldsymbol{v}\| \leq 1} \langle \boldsymbol{v}, \mathbb{E}_{\mathcal{D}}[z\boldsymbol{x}]\rangle &= \sup_{\|\boldsymbol{v}\| \leq 1} \mathbb{E}_{\mathcal{D}}[z\langle \boldsymbol{v}, \boldsymbol{x}\rangle] \\
&\leq \sup_{\|\boldsymbol{v}\| \leq 1} \sqrt{\mathbb{E}_{\mathcal{D}}[z^2] \, \mathbb{E}_{\mathcal{D}}\Big[\langle \boldsymbol{v}, \boldsymbol{x}\rangle^2\Big]} \quad \text{(Cauchy-Schwartz)} \\
&\leq \sqrt{\mathbb{E}_{\mathcal{D}}[z^2] \sup_{\|\boldsymbol{v}\| \leq 1} \langle \boldsymbol{v}, \mathbb{E}_{\mathcal{D}}[\boldsymbol{x}\boldsymbol{x}^\top]\boldsymbol{v}\rangle} \\
&= \sqrt{\mathbb{E}_{\mathcal{D}}[z^2]\|\mathbb{E}_{\mathcal{D}}[\boldsymbol{x}\boldsymbol{x}^\top]\|}.
\end{aligned}
$$

$\square$

Notice that the distribution $\mathcal{D}$ can be both the empirical as well as the population distribution.

## A.2 Approximating the Target Function

We begin by stating the following approximation lemma which is the result of (Bach, 2017, Proposition 6) adapted to our setting.

**Proposition 19.** *Suppose $g : \mathbb{R}^k \to \mathbb{R}$ is $L$-Lipschitz and $|g(0)| = \mathcal{O}(L\tilde{r}_x)$. On the event of Lemma 16, there exists a measure $\mu \in \mathcal{P}_2(\mathbb{R}^{2d+2})$ with $\mathcal{R}(\mu) \leq \Delta^2/\tilde{r}_x^2$ such that*

$$\max_i \Big|g(\boldsymbol{U}\boldsymbol{x}^{(i)}) - \hat{y}(\boldsymbol{x}^{(i)}; \mu)\Big| \leq C_k L\tilde{r}_x \Big(\frac{\Delta}{L\tilde{r}_x}\Big)^{\frac{-2}{k+1}} \ln\Big(\frac{\Delta}{L\tilde{r}_x}\Big) + \frac{\ln 4}{\kappa},$$

*for all $\Delta \geq C_k$, where $C_k$ is a constant depending only on $k$, provided that the hyperparameter $\iota$ satisfies $\iota \geq C_k L\tilde{r}_x \Big(\frac{\Delta}{L\tilde{r}_x}\Big)^{2k/(k+1)}$.*

**Proof.** Throughout the proof, we will use $C_k$ to denote a constant that only depends on $k$, whose value may change across instantiations. Let $\boldsymbol{z} := \boldsymbol{U}\boldsymbol{x} \in \mathbb{R}^k$ and $\tilde{\boldsymbol{z}} := (\boldsymbol{z}^\top, \tilde{r}_x)^\top \in \mathbb{R}^{k+1}$. Recall that on the event of Lemma 12 we have $\|\boldsymbol{z}^{(i)}\| \leq \tilde{r}_x$ and $|g(\boldsymbol{z}^{(i)})| \lesssim L\tilde{r}_x$ for all $1 \leq i \leq n$. Let $\tau$ denote the uniform probability measure on $\mathbb{S}^k$. By (Bach, 2017, Proposition 6), for all $\Delta \geq C_k$, there exists $p \in L^2(\tau)$ with $\|p\|_{L^2(\tau)} \leq \Delta$ such that

$$\max_i \Big|g(\boldsymbol{z}^{(i)}) - \int_{\mathbb{S}^k} p(\boldsymbol{v})\phi_\infty\Big(\frac{1}{\tilde{r}_x}\big\langle \boldsymbol{v}, \tilde{\boldsymbol{z}}^{(i)}\big\rangle\Big)\mathrm{d}\tau(\boldsymbol{v})\Big| \leq C_k L\tilde{r}_x \Big(\frac{\Delta}{L\tilde{r}_x}\Big)^{\frac{-2}{k+1}} \ln\Big(\frac{\Delta}{L\tilde{r}_x}\Big).$$

In fact, we have a stronger guarantee on $p$. Specifically, $p(\boldsymbol{v})$ is given by

$$p(\boldsymbol{v}) = \sum_{j \geq 1} \lambda_j^{-1} r^j h_j(\boldsymbol{v}),$$

where $r \in (0, 1), \lambda_j, h_j : \mathbb{S}^k \to \mathbb{R}$ are introduced by (Bach, 2017, Appendix D). In particular,

$$h(\boldsymbol{v}) = g\Big(\frac{\tilde{r}_x \boldsymbol{v}_{1:k}}{\boldsymbol{v}_{k+1}}\Big)\boldsymbol{v}_{k+1},$$

with the spheircal harmonics decomposition $h(\boldsymbol{v}) = \sum_{j \geq 0} h_j(\boldsymbol{v})$. It is shown in (Bach, 2017, Appendix D.2) that $\lambda_j \leq C_k j^{(k+1)/2}$, and one can prove through spherical harmonics calculations (omitted here for brevity) that $|h_j(\boldsymbol{v})| \leq C_k \sup_{\boldsymbol{v} \in \mathbb{S}^k} h(\boldsymbol{v}) j^{(k-1)/2} \leq C_k L\tilde{r}_x j^{(k-1)/2}$. As a result,

$$|p(\boldsymbol{v})| \leq \sum_{j \geq 0} \lambda_j^{-1} r^j |h_j(\boldsymbol{v})| \leq \sum_{j \geq 1} \lambda_j^{-1} r^j |h_j(\boldsymbol{v})| \leq C_k L\tilde{r}_x \sum_{j \geq 1} j^k r^j \leq \frac{C_k L\tilde{r}_x}{(1-r)^k}.$$

Using $1 - r = \left( C_k L \tilde{r}_x / \Delta \right)^{2/(k+1)}$ as in (Bach, 2017, Appendix D.4) yields

$$|p(\boldsymbol{v})| \le C_k L \tilde{r}_x \left( \frac{\Delta}{L \tilde{r}_x} \right)^{2k/(k+1)}.$$

Define $p_+(\boldsymbol{v}) \coloneqq p(\boldsymbol{v}) \vee 0$ and $p_-(\boldsymbol{v}) \coloneqq (-p(\boldsymbol{v})) \vee 0$. Then, by positive 1-homogeneity of ReLU,

$$
\begin{aligned}
\int_{\mathbb{S}^k} p(\boldsymbol{v}) \phi_\infty \Big( \frac{1}{\tilde{r}_x} \langle \boldsymbol{v}, \tilde{\boldsymbol{z}} \rangle \Big) \mathrm{d}\tau(\boldsymbol{v}) &= \int_{\mathbb{S}^k} p_+(\boldsymbol{v}) \phi_\infty \Big( \frac{1}{\tilde{r}_x} \langle \boldsymbol{v}, \tilde{\boldsymbol{z}} \rangle \Big) \mathrm{d}\tau(\boldsymbol{v}) - \int_{\mathbb{S}^k} p_-(\boldsymbol{v}) \phi_\infty \Big( \frac{1}{\tilde{r}_x} \langle \boldsymbol{v}, \tilde{\boldsymbol{z}} \rangle \Big) \mathrm{d}\tau(\boldsymbol{v}) \\
&= \int_{\mathbb{S}^k} \phi_\infty \Big( \frac{p_+(\boldsymbol{v})}{\tilde{r}_x} \langle \boldsymbol{v}, \tilde{\boldsymbol{z}} \rangle \Big) \mathrm{d}\tau(\boldsymbol{v}) - \int_{\mathbb{S}^k} \phi_\infty \Big( \frac{p_-(\boldsymbol{v})}{\tilde{r}_x} \langle \boldsymbol{v}, \tilde{\boldsymbol{z}}^{(i)} \rangle \Big) \mathrm{d}\tau(\boldsymbol{v}) \\
&= \int_{\mathbb{R}^{k+1}} \phi_\infty (\langle \boldsymbol{v}, \tilde{\boldsymbol{z}} \rangle) \mathrm{d}\tilde{\mu}_1(\boldsymbol{v}) - \int_{\mathbb{R}^{k+1}} \phi_\infty (\langle \boldsymbol{v}, \tilde{\boldsymbol{z}} \rangle) \mathrm{d}\tilde{\mu}_2(\boldsymbol{v}) \\
&= \int_{\mathbb{R}^{d+1}} \phi_\infty (\langle \boldsymbol{w}, \tilde{\boldsymbol{x}} \rangle) \mathrm{d}\mu_1(\boldsymbol{w}) - \int_{\mathbb{R}^{d+1}} \phi_\infty (\langle \boldsymbol{w}, \tilde{\boldsymbol{x}} \rangle) \mathrm{d}\mu_2(\boldsymbol{w}),
\end{aligned}
$$

where $\tilde{\mu}_1 \coloneqq \frac{(\cdot) p_+(\cdot)}{\tilde{r}_x} \# \tau$ and $\tilde{\mu}_2 \coloneqq \frac{(\cdot) p_-(\cdot)}{\tilde{r}_x} \# \tau$ are the corresponding pushforward measures, $\mu_1 = T_{\boldsymbol{U}} \# \tilde{\mu}_1$ and $\mu_2 = T_{\boldsymbol{U}} \# \tilde{\mu}_2$, where $T_{\boldsymbol{U}}(\boldsymbol{v}) = (\boldsymbol{U}^\top \boldsymbol{v}_k, v_{k+1})^\top \in \mathbb{R}^{d+1}$ for $\boldsymbol{v} = (\boldsymbol{v}_k^\top, v_{k+1})^\top \in \mathbb{R}^{k+1}$. In other words, $\boldsymbol{w} \sim \mu_1$ is generated by sampling $\boldsymbol{v} \sim \tilde{\mu}_1$ and letting $\boldsymbol{w} = (\boldsymbol{U}^\top \boldsymbol{v}_k, v_{k+1})^\top$, with a similar procedure for $\boldsymbol{w} \sim \mu_2$. Furthermore,

$$
\begin{aligned}
\mathcal{R}(\mu) = \int_{\mathbb{R}^{d+1}} \|\boldsymbol{w}\|^2 \mathrm{d}\mu_1(\boldsymbol{w}) + \int_{\mathbb{R}^{d+1}} \|\boldsymbol{w}\|^2 \mathrm{d}\mu_2(\boldsymbol{w}) &= \int_{\mathbb{R}^{k+1}} \|\boldsymbol{v}\|^2 \mathrm{d}\tilde{\mu}_1(\boldsymbol{v}) + \int_{\mathbb{R}^{k+1}} \|\boldsymbol{v}\|^2 \mathrm{d}\tilde{\mu}_2(\boldsymbol{v}) \\
&= \int_{\mathbb{S}^k} \frac{p(\boldsymbol{v})^2}{\tilde{r}_x^2} \mathrm{d}\tau(\boldsymbol{v}) \le \frac{\Delta^2}{\tilde{r}_x^2}.
\end{aligned}
$$

The last step is to replace $\phi_\infty$ with $\phi_{\kappa,\iota}$. Note that for all $i$, and almost surely over $\boldsymbol{w} \sim \mu_1$, we have $\left| \langle \boldsymbol{w}, \tilde{\boldsymbol{x}}^{(i)} \rangle \right| \le p_+(\boldsymbol{v}) \le C_k L \tilde{r}_x \left( \frac{\Delta}{L \tilde{r}_x} \right)^{2k/(k+1)}$, with a similar bound holding for $\boldsymbol{w} \sim \mu_2$. As a result, by choosing $\iota \ge C_k L \tilde{r}_x \left( \frac{\Delta}{L \tilde{r}_x} \right)^{2k/(k+1)}$, we have $\phi_{\kappa,\iota} \left( \langle \boldsymbol{w}, \tilde{\boldsymbol{x}}^{(i)} \rangle \right) = \phi_\kappa \left( \langle \boldsymbol{w}, \tilde{\boldsymbol{x}}^{(i)} \rangle \right)$ for all $i$ and almost surely over $\boldsymbol{w} \sim \mu_1$ and $\boldsymbol{w} \sim \mu_2$. By the triangle inequality, we have

$$
\begin{aligned}
\left| g(\boldsymbol{U} \boldsymbol{x}^{(i)}) - \hat{y}(\boldsymbol{x}^{(i)}; \mu) \right| \le & \left| \left\{ \int \phi_{\kappa,\iota} \left( \langle \boldsymbol{w}, \tilde{\boldsymbol{x}}^{(i)} \rangle \right) - \phi_\infty \left( \langle \boldsymbol{w}, \tilde{\boldsymbol{x}}^{(i)} \rangle \right) \right\} \mathrm{d}\mu_1(\boldsymbol{w}) \right| \\
& + \left| \left\{ \int \phi_{\kappa,\iota} \left( \langle \boldsymbol{w}, \tilde{\boldsymbol{x}}^{(i)} \rangle \right) - \phi_\infty \left( \langle \boldsymbol{w}, \tilde{\boldsymbol{x}} \rangle^{(i)} \right) \right\} \mathrm{d}\mu_2(\boldsymbol{w}) \right| \\
& + \left| g(\boldsymbol{U} \boldsymbol{x}^{(i)}) - \int \phi_\infty \left( \langle \boldsymbol{w}, \tilde{\boldsymbol{x}}^{(i)} \rangle \right) (\mathrm{d}\mu_1(\boldsymbol{w}) - \mathrm{d}\mu_2(\boldsymbol{w})) \right| \\
\le & \frac{2 \ln 2}{\kappa} + C_k L \tilde{r}_x \left( \frac{\Delta}{L \tilde{r}_x} \right)^{\frac{-2}{k+1}} \ln \left( \frac{\Delta}{L \tilde{r}_x} \right),
\end{aligned}
$$

which completes the proof. $\qquad \square$

Next, we control the effect of entropic regularization on the minimum of $\tilde{\mathcal{F}}_{\beta,\lambda}$ via the following lemma.

**Lemma 20.** *Suppose $\rho$ is $C_\rho$ Lipschitz. For every $\mu^* \in \mathcal{P}(\mathbb{R}^{2d+2})$, we have*

$$\min_{\mu \in \mathcal{P}^{\mathrm{ac}}(\mathbb{R}^{2d+2})} \tilde{\mathcal{F}}_{\beta,\lambda}(\mu) \le \hat{\mathcal{J}}_0(\mu^*) + \frac{\lambda}{2} \mathcal{R}(\mu^*) + \frac{2\sqrt{2} C_\rho}{\sqrt{\pi \lambda \beta}} \mathbb{E}_{S_n}[\|\tilde{\boldsymbol{x}}\|].$$

**Proof.** We will smooth $\mu^*$ by convoluting it with $\gamma$, i.e. we consider $\mu = \mu^* * \gamma$. Let $\boldsymbol{u} \sim \gamma$ independent of $\boldsymbol{w} \sim \mu^*$ and denote $\boldsymbol{u} = (\boldsymbol{u}_1^\top, \boldsymbol{u}_2^\top)^\top$ with $\boldsymbol{u}_1, \boldsymbol{u}_2 \in \mathbb{R}^{d+1}$. We first bound $\hat{\mathcal{J}}_0(\mu^* * \gamma)$.

Using the Lipschitzness of the loss and of $\phi_{\kappa,\iota}$, we have

$$
\begin{aligned}
\hat{\mathcal{J}}_0(\mu^* * \gamma) - \hat{\mathcal{J}}_0(\mu^*) =& \mathbb{E}_{S_n}\left[\ell\Big(\int \Psi(\boldsymbol{x};\boldsymbol{w})\mathrm{d}(\mu^* * \gamma)(\boldsymbol{w}) - y\Big) - \ell\Big(\int \Psi(\boldsymbol{x};\boldsymbol{w})\mathrm{d}\mu^*(\boldsymbol{w}) - y\Big)\right] \\
\leq& C_\rho \mathbb{E}_{S_n}\left[\Big|\Big|\int \Psi(\boldsymbol{x};\boldsymbol{w})\mathrm{d}(\mu^* * \gamma)(\boldsymbol{w}) - \int \Psi(\boldsymbol{x};\boldsymbol{w})\mathrm{d}\mu^*(\boldsymbol{w})\Big|\Big|\right] \\
=& C_\rho \mathbb{E}_{S_n}\left[\Big|\Big|\int (\mathbb{E}_{\boldsymbol{u}}[\Psi(\boldsymbol{x};\boldsymbol{w}+\boldsymbol{u})] - \Psi(\boldsymbol{x};\boldsymbol{w}))\mathrm{d}\mu^*(\boldsymbol{w})\Big|\Big|\right] \\
\leq& C_\rho \mathbb{E}_{S_n}\left[\int \mathbb{E}_{\boldsymbol{u}}[|\phi_{\kappa,\iota}(\langle\boldsymbol{\omega}_1+\boldsymbol{u}_1,\tilde{\boldsymbol{x}}\rangle) - \phi_{\kappa,\iota}(\langle\boldsymbol{\omega}_1,\tilde{\boldsymbol{x}}\rangle)|]\mathrm{d}\mu^*(\boldsymbol{w})\right] \\
&+ C_\rho \mathbb{E}_{S_n}\left[\int \mathbb{E}_{\boldsymbol{u}}[|\phi_{\kappa,\iota}(\langle\boldsymbol{\omega}_2+\boldsymbol{u}_2,\tilde{\boldsymbol{x}}\rangle) - \phi_{\kappa,\iota}(\langle\boldsymbol{\omega}_2,\tilde{\boldsymbol{x}}\rangle)|]\mathrm{d}\mu^*(\boldsymbol{w})\right] \\
\leq& C_\rho \mathbb{E}_{S_n}\left[\int \{\mathbb{E}_{\boldsymbol{u}_1}[|\langle\boldsymbol{u}_1,\tilde{\boldsymbol{x}}\rangle|] + \mathbb{E}_{\boldsymbol{u}_2}[|\langle\boldsymbol{u}_2,\tilde{\boldsymbol{x}}\rangle|]\}\mathrm{d}\mu^*(\boldsymbol{\omega})\right] \\
=& \frac{2\sqrt{2}C_\rho}{\sqrt{\pi\lambda\beta}}\mathbb{E}_{S_n}[\|\tilde{\boldsymbol{x}}\|].
\end{aligned}
$$

Next, we bound the KL divergence via its convexity in the first argument,

$$
\mathcal{H}(\mu^* * \gamma \,|\, \gamma) = \mathcal{H}\left(\int \gamma(\cdot - \boldsymbol{w}')\mathrm{d}\mu^*(\boldsymbol{w}') \,|\, \gamma\right) \leq \int \mathcal{H}(\gamma(\cdot - \boldsymbol{w}') \,|\, \gamma(\cdot))\mathrm{d}\mu^*(\boldsymbol{w}').
$$

Furthermore,

$$
\mathcal{H}(\gamma(\cdot - \boldsymbol{w}') \,|\, \gamma(\cdot)) = \int \frac{\lambda\beta}{2}\big(-\|\boldsymbol{w}-\boldsymbol{w}'\|^2 + \|\boldsymbol{w}\|^2\big)\gamma(\mathrm{d}\boldsymbol{w}-\boldsymbol{w}') = \frac{\lambda\beta\|\boldsymbol{w}'\|^2}{2}.
$$

Consequently,

$$
\mathcal{H}(\mu^* * \gamma \,|\, \gamma) \leq \frac{\lambda\beta}{2}\mathcal{R}(\mu^*),
$$

which finishes the proof. $\qquad\square$

Combining above results, we have the following statement.

**Corollary 21.** *Suppose the event of Lemma 16 holds, $\rho$ is $C_\rho$ Lipschitz, and $\lambda \lesssim 1$. Then,*

$$
\min_{\mu \in \mathcal{P}^{\mathrm{ac}}(\mathbb{R}^{2d+2})} \tilde{\mathcal{F}}_{\beta,\lambda}(\mu) - \mathbb{E}_{S_n}[\rho(\xi)] \lesssim C_\rho \frac{\tilde{r}_x}{r_x}\left(\frac{r_x\Delta}{\tilde{r}_x}\right)^{\frac{-2}{k+1}}\ln\left(\frac{r_x\Delta}{\tilde{r}_x}\right) + \frac{C_\rho}{\kappa} + \frac{\lambda\Delta^2}{\tilde{r}_x^2} + \frac{C_\rho(c_x+\tilde{r}_x)}{\sqrt{\lambda\beta}},
$$

*for all $\Delta \geq C_k$, provided that $\iota \geq C_k\Delta^{2k/(k+1)}(r_x/\tilde{r}_x)^{(k-1)/(k+1)}$.*

**Proof.** We will use Lemma 20 with $\mu^* \in \mathcal{P}(\mathbb{R}^{2d+2})$ constructed in Proposition 19. Then, for all $\Delta \geq C_k$,

$$
\begin{aligned}
\hat{\mathcal{J}}_0(\mu^*) &= \mathbb{E}_{S_n}[\rho(\hat{y}(\boldsymbol{x};\mu^*) - y)] \\
&= \mathbb{E}_{S_n}[\rho(\hat{y}(\boldsymbol{x};\mu^*) - g(\boldsymbol{U}\boldsymbol{x}) - \xi)] \\
&\leq \mathbb{E}_{S_n}[\rho(\xi)] + C_\rho \mathbb{E}_{S_n}[|\hat{y}(\boldsymbol{x};\mu^*) - g(\boldsymbol{U}\boldsymbol{x})|] \\
&\leq \mathbb{E}_{S_n}[\rho(\xi)] + C_k C_\rho \frac{\tilde{r}_x}{r_x}\left(\frac{r_x\Delta}{\tilde{r}_x}\right)^{-\frac{2}{k+1}}\ln\left(\frac{r_x\Delta}{\tilde{r}_x}\right) + \frac{C_\rho \ln 4}{\kappa}.
\end{aligned}
$$

Furthermore, Proposition 19 guarantees $\mathcal{R}(\mu^*) \leq \Delta^2/\tilde{r}_x^2$. Combining these bounds with Lemma 20 completes the proof. $\qquad\square$

### A.3 GENERALIZATION ANALYSIS

Let

$$
\mu_\beta^* := \underset{\mu \in \mathcal{P}_2^{\mathrm{ac}}(\mathbb{R}^{2d+2})}{\arg\min} \mathcal{F}_{\beta,\lambda}(\mu) = \underset{\mu \in \mathcal{P}_2^{\mathrm{ac}}(\mathbb{R}^{2d+2})}{\arg\min} \tilde{\mathcal{F}}_{\beta,\lambda}(\mu).
$$

Corollary 21 gives an upper bound on $\hat{\mathcal{J}}_0(\mu^*)$. In this section, we transfer the bound to $\mathcal{J}_0(\mu^*)$ via a Rademacher complexity analysis. Since Corollary 21 implies a bound on $\mathcal{H}(\mu \mid \gamma)$, we will control the following quantity,

$$\sup_{\mu:\mathcal{H}(\mu \mid \gamma)\leq\Delta^2} \mathcal{J}_0(\mu) - \hat{\mathcal{J}}_0(\mu).$$

To be able to provide guarantees with high probability, we will prove uniform convergence over a truncated version of the risk instead, given by

$$\sup_{\mu:\mathcal{H}(\mu \mid \gamma)\leq\Delta^2} \mathcal{J}_0^{\varkappa}(\mu) - \hat{\mathcal{J}}_0^{\varkappa}(\mu),$$

where

$$\mathcal{J}_0^{\varkappa}(\mu) := \mathbb{E}[\rho_{\varkappa}(\hat{y}(\boldsymbol{x};\mu) - y)], \quad \hat{\mathcal{J}}_0^{\varkappa}(\mu) := \mathbb{E}_{S_n}[\rho_{\varkappa}(\hat{y}(\boldsymbol{x};\mu) - y)],$$

and $\rho_{\varkappa}(\cdot) := \rho(\cdot) \wedge \varkappa$. We will later specify the choice of $\varkappa$.

We are now ready to present the Rademacher complexity bound.

**Lemma 22** ((Chen et al., 2020, Lemma 5.5), (Suzuki et al., 2023b, Lemma 1)). *Suppose $\rho$ is either a $C_\rho$-Lipschitz loss or the squared error loss. Let $\vartheta := \sqrt{2\varkappa}$ for the squared error loss and $C_\rho$ for the Lipschitz loss. Recall $\gamma = \mathcal{N}(0, \frac{\mathbf{I}_{d+1}}{\lambda\beta})$. Then,*

$$\mathbb{E}\left[\sup_{\{\mu\in\mathcal{P}^{\mathrm{ac}}(\mathbb{R}^{2d+2}):\mathcal{H}(\mu \mid \gamma)\leq M\}} \mathcal{J}_0^{\varkappa}(\mu) - \hat{\mathcal{J}}_0^{\varkappa}(\mu)\right] \leq 4\vartheta\iota\sqrt{\frac{2M}{n}}.$$

**Proof.** We repeat the proof here for the reader's convenience. Let $(\xi_i)_{i=1}^n$ denote i.i.d. Rademacher random variables. Notice that for the squared error loss, $\rho_{\varkappa}$ is $\sqrt{2\varkappa}$ Lipschitz. Then, by a standard symmetrization argument and Talagrand's contraction lemma, we have

$$\mathbb{E}\left[\sup_{\mu:\mathcal{H}(\mu \mid \gamma)\leq M} \mathcal{J}_0(\mu) - \hat{\mathcal{J}}_0(\mu)\right] \leq 2\mathbb{E}\left[\sup_{\mu:H(\mu \mid \gamma)\leq M} \frac{1}{n}\sum_{i=1}^n \xi_i\rho(\hat{y}(\boldsymbol{x}^{(i)};\mu) - y)\right]$$

$$\leq 2\vartheta\mathbb{E}\left[\sup_{\mu:H(\mu \mid \gamma)\leq M} \frac{1}{n}\sum_{i=1}^n \xi_i\hat{y}(\boldsymbol{x}^{(i)};\mu)\right]$$

Next, we proceed to bound the Rademacher complexity. Specifically,

$$\mathbb{E}_{\boldsymbol{\xi}}\left[\sup_{\mu:H(\mu \mid \gamma)\leq M} \frac{1}{n}\sum_{i=1}^n \xi_i \int \Psi(\boldsymbol{x}^{(i)};\boldsymbol{w})\mathrm{d}\mu(\boldsymbol{w})\right] = \mathbb{E}_{\boldsymbol{\xi}}\left[\frac{1}{\alpha}\sup_{\mu:H(\mu \mid \gamma)\leq M} \int \frac{\alpha}{n}\sum_{i=1}^n \xi_i\Psi(\boldsymbol{x}^{(i)};\boldsymbol{w})\mathrm{d}\mu(\boldsymbol{w})\right]$$

$$\leq \frac{M}{\alpha} + \frac{1}{\alpha}\mathbb{E}_{\boldsymbol{\xi}}\left[\ln\int\exp\left(\frac{\alpha}{n}\sum_{i=1}^n \xi_i\Psi(\boldsymbol{x}^{(i)};\boldsymbol{w})\right)\mathrm{d}\gamma(\boldsymbol{w})\right]$$

$$\leq \frac{M}{\alpha} + \frac{1}{\alpha}\ln\int\mathbb{E}_{\xi}\left[\exp\left(\frac{\alpha}{n}\sum_{i=1}^n \xi_i\Psi(\boldsymbol{x}^{(i)};\boldsymbol{w})\right)\right]\mathrm{d}\gamma(\boldsymbol{w}),$$

where the first inequality follows from the KL divergence lower bound of Lemma 17. Additionally, by sub-Gaussianity and independence of $(\xi_i)$ and Lipschitzness of $\phi_{\kappa,\iota}$, we have

$$\mathbb{E}_\xi\left[\exp\left(\frac{\alpha}{n}\sum_{i=1}^n \xi_i\Psi(\boldsymbol{x}^{(i)};\boldsymbol{w})\right)\right] \leq \exp\left(\frac{\alpha^2}{2n^2}\sum_{i=1}^n \Psi(\boldsymbol{x}^{(i)};\boldsymbol{w})^2\right)$$

$$\leq \exp\left(\frac{2\alpha^2\iota^2}{n}\right)$$

Plugging this back into our original bound, we obtain

$$\mathbb{E}_{\boldsymbol{\xi}}\left[\sup_{\mu:H(\mu \mid \gamma)\leq M} \frac{1}{n}\sum_{i=1}^n \xi_i\hat{y}(\boldsymbol{x};\mu)\right] \leq \frac{M}{\alpha} + \frac{2\alpha\iota^2}{n}.$$

Choosing $\alpha = \sqrt{\frac{Mn}{2\iota^2}}$, we obtain

$$\mathbb{E}_{\boldsymbol{\xi}}\left[\sup_{\mu:\mathcal{H}(\mu\,|\,\gamma)\leq M}\frac{1}{n}\sum_{i=1}^{n}\xi_i\hat{y}(\boldsymbol{x};\mu)\right] \leq 2\iota\sqrt{\frac{2M}{n}},$$

which completes the proof. $\qquad\square$

We can convert the above bound in expectation to a high-probability bound as follows.

**Lemma 23.** *In the setting of Lemma 22, for any $\delta > 0$, we have*

$$\sup_{\mu\in\mathcal{P}^{ac}(\mathbb{R}^{2d+2}):\mathcal{H}(\mu\,|\,\gamma)\leq M}\mathcal{J}_0^{\varkappa}(\mu) - \hat{\mathcal{J}}_0^{\varkappa}(\mu) \lesssim \vartheta\iota\sqrt{\frac{M}{n}} + \varkappa\sqrt{\frac{\ln(1/\delta)}{n}},$$

*with probability at least $1 - \delta$.*

**Proof.** As the truncated loss is bounded by $\varkappa$, the result is an immediate consequence of McDiarmid's inequality. $\qquad\square$

Next, we control the effect of truncation by bounding $\mathcal{J}_0(\mu)$ via $\mathcal{J}_0^{\varkappa}(\mu)$, which is achieved by the following lemma.

**Lemma 24.** *Suppose $\mathcal{H}(\mu\,|\,\gamma) \leq M$. Then,*

$$\mathcal{J}_0(\mu) - \mathcal{J}_0^{\varkappa}(\mu) \lesssim \left(\iota + \mathbb{E}\big[y^2\big]^{1/2}\right)\left(e^{-\Omega(\varkappa^2)} + n^{-q-1}\right).$$

**Proof.** Notice that since the loss is $C_\rho$-Lipschitz and $\rho(0) = 0$, we have $|\rho(\hat{y} - y)| \leq C_\rho|\hat{y} - y|$. Recall that we use $L$ for the Lipschitz constant of $g$, and $|\hat{y}(\boldsymbol{x};\mu)| \leq 2\iota$. Then,

$$\mathcal{J}_0(\mu) - \mathcal{J}_0^{\varkappa}(\mu) \leq \mathbb{E}[\mathbb{1}(\rho(\hat{y}(\boldsymbol{x};\mu) - y) \geq \varkappa)\rho(\hat{y}(\boldsymbol{x};\mu) - y)]$$

$$\leq C_\rho\mathbb{P}(\rho(\hat{y}(\boldsymbol{x};\mu) - y) \geq \varkappa)^{1/2}\,\mathbb{E}\big[(\hat{y}(\boldsymbol{x};\mu) - y)^2\big]^{1/2}$$

$$\leq C_\rho\mathbb{P}(2\iota + |y| \geq \varkappa/C_\rho)^{1/2}\left(\mathbb{E}\big[\hat{y}(\boldsymbol{x};\mu)^2\big]^{1/2} + \mathbb{E}\big[y^2\big]^{1/2}\right).$$

Additionally, by local Lipschitzness of $g$,

$$\mathbb{P}(2\iota + |y| \geq \varkappa/C_\rho) \leq \mathbb{P}\big(\{\{2\iota + |y| \geq \varkappa/C_\rho\} \cap \{\|\boldsymbol{U}\boldsymbol{x}\| \leq \tilde{r}_x\}\} \cup \{\|\boldsymbol{U}\boldsymbol{x}\| \geq \tilde{r}_x\}\big)$$

$$\leq \mathbb{P}(2\iota + |g(0)| + L\|\boldsymbol{U}\boldsymbol{x}\| + |\xi| \geq \varkappa/C_\rho) + \mathbb{P}(\|\boldsymbol{U}\boldsymbol{x}\| \geq \tilde{r}_x)$$

$$\leq \mathbb{P}(2\iota + |g(0)| + L\|\boldsymbol{U}\boldsymbol{x}\| + |\xi| \geq \varkappa/C_\rho) + n^{-(q+1)}.$$

Furthermore, Let $\varkappa/C_\rho \geq 4\iota + 2|g(0)| + 2Lr_x + 2\,\mathbb{E}[|\xi|]$, and recall that $L = \mathcal{O}(1/r_x)$. Then, by a subGaussian concentration bound, we have

$$\mathbb{P}(2\iota + |g(0)| + L\|\boldsymbol{U}\boldsymbol{x}\| + \xi \geq \varkappa/C_\rho)^{1/2} \leq e^{-\Omega\left(\frac{\varkappa^2}{\sigma_u^2 C_\rho^2}\right)}.$$

We conclude the proof by remarking that by our assumptions, $\sigma_u$ and $C_\rho$ are absolute constants. $\qquad\square$

Finally, we combine the steps above to give an upper bound on $\mathcal{J}_0(\mu_\beta^*)$, stated in the following lemma.

**Lemma 25.** *Suppose $\lambda = \tilde{\lambda}r_x^2$ and $\beta = \frac{d_{\text{eff}} + \tilde{r}_x^2/r_x^2}{\varepsilon^2\tilde{\lambda}}$ for $\varepsilon, \tilde{\lambda} \lesssim 1$. Let $\tilde{\varepsilon} := \tilde{\mathcal{O}}(\tilde{\lambda}^{\frac{1}{k+2}}) + \varepsilon + \kappa^{-1}$. Suppose $n \gtrsim \frac{(d_{\text{eff}} + \tilde{r}_x^2/r_x^2)\iota^2}{\tilde{\lambda}\varepsilon^4}$ and $\iota \gtrsim \tilde{\lambda}^{-\frac{k}{k+2}}(\tilde{r}_x/r_x)^{\frac{2(k+1)}{k+2}}$. Then,*

$$\mathcal{J}_0(\mu_\beta^*) - \mathbb{E}[\rho(\xi)] \lesssim \tilde{\varepsilon}, \quad \text{and} \quad \beta^{-1}\mathcal{H}(\mu_\beta^*\,|\,\gamma) \lesssim \mathbb{E}[\rho(\xi)] + \tilde{\varepsilon} \lesssim 1.$$

**Proof.** By Corollary 21 and a standard concentration bound on $\mathbb{E}_{S_n}[\rho(\xi)]$ with sufficiently large $n$ to induce neglibile error in comaprison with the rest of the terms in the corollary, we have

$$\hat{\mathcal{J}}_0(\mu_\beta^*) + \beta^{-1}\mathcal{H}(\mu_\beta^*\,|\,\gamma) - \mathbb{E}[\rho(\xi)] \lesssim \frac{\tilde{r}_x}{r_x}\left(\frac{r_x\Delta}{\tilde{r}_x}\right)^{\frac{-2}{k+1}}\ln\left(\frac{r_x\Delta}{\tilde{r}_x}\right) + \frac{\lambda\Delta^2}{\tilde{r}_x^2} + \frac{(c_x + \tilde{r}_x)}{\sqrt{\lambda\beta}} + \frac{1}{\kappa}.$$

By choosing

$$\Delta = \left(\frac{r_x^2}{\lambda}\right)^{\frac{1}{2} \cdot \frac{k+1}{k+2}} \left(\frac{\tilde{r}_x}{r_x}\right)^{\frac{1}{2} \cdot \frac{3k+5}{k+2}},$$

and assuming $c_x \gtrsim \tilde{r}_x$,

$$\beta^{-1}\mathcal{H}(\mu_\beta^* \,|\, \gamma) \lesssim \mathbb{E}[\rho(\xi)] + \left(\frac{\lambda}{r_x^2}\right)^{\frac{1}{k+2}} \left(\frac{\tilde{r}_x}{r_x}\right)^{\frac{k+1}{k+2}} \ln\left(\frac{\tilde{r}_x r_x}{\lambda}\right) + \frac{c_x}{\sqrt{\lambda\beta}} + \frac{1}{\kappa}.$$

Note that the above choice on $\Delta$ translates to a lower bound on $\iota$ in Corollary 21, given by

$$\iota \gtrsim \tilde{\lambda}^{-\frac{k}{k+2}} \left(\frac{\tilde{r}_x}{r_x}\right)^{\frac{2(k+1)}{k+2}}.$$

By choosing $\lambda = \tilde{\lambda}r_x^2$ and using the fact that $\tilde{r}_x \leq \tilde{\mathcal{O}}(r_x)$ and $\beta = \frac{c_x^2}{r_x^2 \tilde{\lambda}\varepsilon^2}$, we have the simpification,

$$\beta^{-1}\mathcal{H}(\mu_\beta^* \,|\, \gamma) \lesssim \mathbb{E}[\rho(\xi)] + \tilde{\mathcal{O}}(\tilde{\lambda}^{\frac{1}{k+2}}) + \varepsilon + \frac{1}{\kappa} \lesssim 1,$$

and,

$$\hat{\mathcal{J}}_0(\mu_\beta^*) - \mathbb{E}[\rho(\xi)] \lesssim \tilde{\mathcal{O}}(\tilde{\lambda}^{\frac{1}{k+2}}) + \varepsilon + \frac{1}{\kappa} =: \tilde{\varepsilon}.$$

Note that $\hat{\mathcal{J}}_0^\varkappa(\mu_\beta^*) \leq \hat{\mathcal{J}}_0(\mu_\beta^*)$. Using the generalization bound of Lemma 23 with the choice of $\delta = n^{-q}$ for some constant $q > 0$, we have with probability $1 - \mathcal{O}(n^{-q})$,

$$\begin{aligned}
\mathcal{J}_0^\varkappa(\mu_\beta^*) - \hat{\mathcal{J}}_0^\varkappa(\mu_\beta^*) &\lesssim \iota\sqrt{\frac{\beta}{n}} + \varkappa\sqrt{\frac{\ln n}{n}} \\
&\lesssim \iota\sqrt{\frac{d_{\text{eff}}}{n\tilde{\lambda}\varepsilon^2}} + \varkappa\sqrt{\frac{\ln n}{n}}.
\end{aligned} \tag{A.5}$$

Furthermore, by Lemma 24 we have

$$\mathcal{J}_0(\mu_\beta^*) - \mathcal{J}_0^\varkappa(\mu_\beta^*) \lesssim \iota e^{-\Omega(\varkappa^2)}.$$

Combining the above with (A.5) and choosing on $\varkappa \asymp \sqrt{\ln n}$, we have

$$\mathcal{J}_0(\mu_\beta^*) - \mathbb{E}[\rho(\xi)] \lesssim \tilde{\varepsilon} + \iota\sqrt{\frac{d_{\text{eff}}}{n\tilde{\lambda}\varepsilon^2}} + \sqrt{\frac{\ln^2 n}{n}},$$

which holds with probability at least $1 - \mathcal{O}(n^{-q})$ over the randomness of $S_n$. $\qquad\square$

## A.4 CONVERGENCE ANALYSIS

So far, our analysis has only proved properties of $\mu_\beta^*$. In this section, we relate these properties to $\mu_l^m$ via propagation of chaos. In particular, Suzuki et al. (2023a) showed that for $\boldsymbol{W} \sim \mu_l^m$, $\hat{y}(\boldsymbol{x}; \mu_l^m)$ converges to $\hat{y}(\boldsymbol{x}; \mu_\beta^*)$ in a suitable sense characterized shortly, as long as the objective over $\mu_l^m$ converges to $\mathcal{F}_{\beta,\lambda}(\mu_\beta^*)$. Notice that $\mu_\ell^m$ is a measure on $\mathcal{P}(\mathbb{R}^{(2d+2)m})$ instead of $\mathcal{P}(\mathbb{R}^{2d+2})$. Thus, we need to adjust the definition of objective by defining the following

$$\mathcal{F}_{\beta,\lambda}^m(\mu^m) := \mathbb{E}_{\boldsymbol{W} \sim \mu^m}\left[\hat{J}_0(\boldsymbol{W}) + \frac{\lambda}{2}R(\boldsymbol{W})\right] + \frac{1}{m\beta}\mathcal{H}(\mu^m). \tag{A.6}$$

We can use the same reformulation introduced earlier in (A.1) to define

$$\tilde{\mathcal{F}}_{\beta,\lambda}^m(\mu^m) := \mathbb{E}_{\boldsymbol{W} \sim \mu^m}\left[\hat{J}_0(\boldsymbol{W})\right] + \frac{1}{m\beta}\mathcal{H}(\mu^m \,|\, \gamma^{\otimes m}), \tag{A.7}$$

which is equivalent to $\mathcal{F}_{\beta,\lambda}^m$ up to an additive constant. With these definitions, we can now control $\mathbb{E}_{\boldsymbol{W} \sim \mu_l^m}[J_0(\mu_l^m)]$ via $\mathcal{J}_0(\mu_\beta^*)$. The following lemma is based on (Suzuki et al., 2023a, Lemma 4), with a more careful analysis to obtain sharper constants.

**Lemma 26.** *Let $\bar{r}_x \coloneqq \|\mathbf{\Sigma}\|^{1/2} \vee \tilde{r}_x$, and suppose $\rho$ is $C_\rho \lesssim 1$-Lipschitz. Then,*

$$\mathbb{E}_{\boldsymbol{W} \sim \mu_l^m}[J_0(\boldsymbol{W})] - \mathcal{J}_0(\mu_\beta^*) \lesssim \sqrt{\frac{\bar{r}_x^2 W_2^2(\mu_l^m, \mu_\beta^{*\otimes m}) + \iota^2}{m}}. \tag{A.8}$$

*In particular, combined with ([Suzuki et al., 2023a](), Lemma 3), the above implies*

$$\mathbb{E}_{\boldsymbol{W} \sim \mu_l^m}[J_0(\boldsymbol{W})] - \mathcal{J}_0(\mu_\beta^*) \lesssim \sqrt{\frac{\bar{r}_x^2 \beta C_{\text{LSI}}}{m} \left(\tilde{\mathcal{F}}_{\beta,\lambda}^m(\mu_l^m) - \tilde{F}_{\beta,\lambda}(\mu_\beta^*)\right) + \frac{\iota^2}{m}}. \tag{A.9}$$

**Proof.** Notice that

$$
\begin{aligned}
\mathbb{E}_{\boldsymbol{W} \sim \mu_l^m}[J_0(\boldsymbol{W})] &= \mathbb{E}_{\boldsymbol{W}}\left[\mathbb{E}_{\boldsymbol{x}}\left[\rho(\hat{y}(\boldsymbol{x}; \mu_{\boldsymbol{W}}) - \hat{y}(\boldsymbol{x}; \mu_\beta^*) + \hat{y}(\boldsymbol{x}; \mu_\beta^*) - y)\right]\right] \\
&\leq \mathbb{E}_{\boldsymbol{x}}\left[\rho(\hat{y}(\boldsymbol{x}; \mu_\beta^*) - y)\right] + C_\rho \, \mathbb{E}_{\boldsymbol{W}}\left[\mathbb{E}_{\boldsymbol{x}}\left[|\hat{y}(\boldsymbol{x}; \mu_{\boldsymbol{W}}) - \hat{y}(\boldsymbol{x}; \mu_\beta^*)|\right]\right] \\
&\leq \mathcal{J}_0(\mu_\beta^*) + C_\rho \sqrt{\mathbb{E}_{\boldsymbol{x}}\left[\mathbb{E}_{\boldsymbol{W}}\left[(\hat{y}(\boldsymbol{x}; \mu_{\boldsymbol{W}}) - \hat{y}(\boldsymbol{x}; \mu_\beta^*))^2\right]\right]}
\end{aligned}
$$

Suppose $\boldsymbol{W} = (\boldsymbol{w}_1, \ldots, \boldsymbol{w}_m) \sim \mu_l^m$ and $\boldsymbol{W}' = (\boldsymbol{w}_1', \ldots, \boldsymbol{w}_m') \sim \mu_\beta^{*\otimes m}$. Let $\Gamma$ denote the optimal $W_2$ coupling between $\boldsymbol{W}$ and $\boldsymbol{W}'$, and assume $\boldsymbol{W}, \boldsymbol{W}' \sim \Gamma$. Then,

$$
\begin{aligned}
\mathbb{E}_{\boldsymbol{W}}\left[(\hat{y}(\boldsymbol{x}; \mu_{\boldsymbol{W}}) - \hat{y}(\boldsymbol{x}; \mu^*))^2\right] &= \mathbb{E}_{\boldsymbol{W}, \boldsymbol{W}'}\left[(\hat{y}(\boldsymbol{x}; \mu_{\boldsymbol{W}}) - \hat{y}(\boldsymbol{x}; \mu_{\boldsymbol{W}'}) + \hat{y}(\boldsymbol{x}; \mu_{\boldsymbol{W}'}) - \hat{y}(\boldsymbol{x}; \mu_\beta^*))^2\right] \\
&\leq 2 \, \mathbb{E}_{\boldsymbol{W}, \boldsymbol{W}'}\left[(\hat{y}(\boldsymbol{x}; \mu_{\boldsymbol{W}}) - \hat{y}(\boldsymbol{x}; \mu_{\boldsymbol{W}'}))^2\right] + 2 \, \mathbb{E}_{\boldsymbol{W}'}\left[(\hat{y}(\boldsymbol{x}; \mu_{\boldsymbol{W}'}) - \hat{y}(\boldsymbol{x}; \mu_\beta^*))^2\right]
\end{aligned}
$$

Moreover, by Jensen's inequality,

$$
\begin{aligned}
\mathbb{E}_{\boldsymbol{W}, \boldsymbol{W}'}\left[(\hat{y}(\boldsymbol{x}; \mu_{\boldsymbol{W}}) - \hat{y}(\boldsymbol{x}; \mu_{\boldsymbol{W}'}))^2\right] &\leq \frac{1}{m} \sum_{i=1}^m \mathbb{E}_{\boldsymbol{W}, \boldsymbol{W}'}\left[(\Psi(\boldsymbol{x}; \boldsymbol{w}_i) - \Psi(\boldsymbol{x}; \boldsymbol{w}_i'))^2\right] \\
&\leq \frac{2}{m} \sum_{i=1}^m \mathbb{E}_{\boldsymbol{W}, \boldsymbol{W}'}\left[\langle \boldsymbol{\omega}_{i1} - \boldsymbol{\omega}_{i1}', \tilde{\boldsymbol{x}} \rangle^2\right] + \frac{2}{m} \sum_{i=1}^m \mathbb{E}_{\boldsymbol{W}, \boldsymbol{W}'}\left[\langle \boldsymbol{\omega}_{i2} - \boldsymbol{\omega}_{i2}', \tilde{\boldsymbol{x}} \rangle^2\right].
\end{aligned}
$$

Hence,

$$
\begin{aligned}
\mathbb{E}_{\boldsymbol{x}}\left[\mathbb{E}_{\boldsymbol{W}, \boldsymbol{W}'}\left[(\hat{y}(\boldsymbol{x}; \mu_{\boldsymbol{W}}) - \hat{y}(\boldsymbol{x}; \mu_{\boldsymbol{W}'}))^2\right]\right] &\leq \frac{2\left\|\tilde{\mathbf{\Sigma}}\right\|}{m} \mathbb{E}_{\boldsymbol{W}, \boldsymbol{W}'}\left[\|\boldsymbol{W} - \boldsymbol{W}'\|_{\text{F}}^2\right] \\
&= \frac{2\left\|\tilde{\mathbf{\Sigma}}\right\|}{m} W_2^2(\mu_t^m, \mu_\beta^{*\otimes m}).
\end{aligned}
$$

For the second term, notice that $\hat{y}(\boldsymbol{x}; \mu_\beta^*) = \mathbb{E}_{\boldsymbol{W}'}[\hat{y}(\boldsymbol{x}; \mu_{\boldsymbol{W}'})] = \mathbb{E}_{\boldsymbol{w}_i'}[\Psi(\boldsymbol{x}; \boldsymbol{w}_i')]$ for all $1 \leq i \leq m$. By independence of $(\boldsymbol{w}_i')$ and Jensen's inequality, we have

$$
\begin{aligned}
\mathbb{E}_{\boldsymbol{W}'}\left[(\hat{y}(\boldsymbol{x}; \mu_{\boldsymbol{W}'}) - \hat{y}(\boldsymbol{x}; \mu_\beta^*))^2\right] &= \frac{1}{m} \mathbb{E}_{\boldsymbol{w}'}\left[(\Psi(\boldsymbol{x}; \boldsymbol{w}') - \hat{y}(\boldsymbol{x}; \mu^*))^2\right] \\
&= \frac{1}{m} \mathbb{E}_{\boldsymbol{w}'}\left[\left(\int (\Psi(\boldsymbol{x}; \boldsymbol{w}') - \Psi(\boldsymbol{x}; \boldsymbol{w})) \mathrm{d}\mu_\beta^*(\boldsymbol{w})\right)^2\right] \\
&\lesssim \frac{\iota^2}{m}.
\end{aligned}
$$

$\square$

Thus, the rest of this section deals with establishing convergence rates for $\mathcal{F}_{\beta,\lambda}^m(\mu_l^m) \to \mathcal{F}_{\beta,\lambda}(\mu_\beta^*)$. To use the one-step decay of optimality gap provided by [Suzuki et al. (2023a)](), we depend on the following assumption.

**Assumption 5.** *Suppose there exist constants $L$, $C_L$, and $R$, such that*

1. **(Lipschitz gradients of the Gibbs potential)** *For all $\mu, \mu' \in \mathcal{P}_2(\mathbb{R}^{2d+2})$ and $\boldsymbol{w}, \boldsymbol{w}' \in \mathbb{R}^{2d+2}$,*

$$\left\|\nabla \hat{\mathcal{J}}_0'[\mu](\boldsymbol{w}) - \nabla \hat{\mathcal{J}}_0'[\mu'](\boldsymbol{w}')\right\| \leq L(W_2(\mu, \mu') + \|\boldsymbol{w} - \boldsymbol{w}'\|), \tag{A.10}$$

   *where $W_2$ is the 2-Wasserstein distance.*

2. **(Bounded gradients of the Gibbs potential)** *For all $\mu \in \mathcal{P}_2(\mathbb{R}^{2d+2})$ and $\boldsymbol{w} \in \mathbb{R}^{2d+2}$, we have $\left\|\nabla \hat{\mathcal{J}}_0'[\mu](\boldsymbol{w})\right\| \leq R$.*

3. **(Bounded second variation)** *Denote the second variation of $\hat{\mathcal{J}}_0(\mu)$ at $\boldsymbol{w}$ via $\hat{\mathcal{J}}_0''[\mu](\boldsymbol{w}, \boldsymbol{w}')$, which is defined as the first variation of $\mu \mapsto \hat{\mathcal{J}}_0'[\mu](\boldsymbol{w})$ (see (2.6) for the definition of first variation). Then, for all $\mu \in \mathcal{P}_2(\mathbb{R}^{2d+2})$ and $\boldsymbol{w}, \boldsymbol{w}' \in \mathbb{R}^{2d+2}$,*

$$\left|\hat{\mathcal{J}}_0''[\mu](\boldsymbol{w}, \boldsymbol{w}')\right| \leq L(1 + C_L(\|\boldsymbol{w}\|^2 + \|\boldsymbol{w}'\|^2)). \tag{A.11}$$

We can now state the one-step bound.

**Theorem 27.** (Suzuki et al., 2023a, Theorem 2) *Suppose $\hat{\mathcal{J}}_0$ satisfies Assumption 5. Assume $\lambda \lesssim 1$, $\beta, L, R \gtrsim 1$, and the initialization satisfies $\mathbb{E}\left[\|\boldsymbol{w}_0^i\|^2\right] \lesssim R^2$ for all $1 \leq i \leq m$. Then, for all $\eta \leq 1/4$,*

$$\mathcal{F}_{\beta,\lambda}^m(\mu_{l+1}^m) - \mathcal{F}_{\beta,\lambda}(\mu_\beta^*) \leq \exp\left(\frac{-\eta}{2\beta C_{\text{LSI}}}\right)\left(\mathcal{F}_{\beta,\lambda}^m(\mu_l^m) - \mathcal{F}_{\beta,\lambda}(\mu_\beta^*)\right) + \eta A_{m,\beta,\lambda,\eta}, \tag{A.12}$$

*where*

$$A_{m,\beta,\lambda,\eta} := C\left(L^2\left(d + \frac{R^2}{\lambda}\right)\left(\eta^2 + \frac{\eta}{\beta}\right) + \frac{L}{m\beta}\left(\frac{1}{C_{\text{LSI}}} + \left(\frac{R^2}{\lambda^2} + \frac{d}{\lambda\beta}\right)\left(\frac{C_L}{C_{\text{LSI}}} + \frac{L}{\beta}\right)\right)\right) \tag{A.13}$$

*for some absolute constant $C > 0$.*

We now focus on bounding the constants that appear in Assumption 5.

**Lemma 28** (Lipschitzness of $\nabla \hat{\mathcal{J}}_0'$). *Suppose $\rho$ is either the squared error loss or is $C_\rho$ Lipschitz and has a $C_\rho'$ Lipschitz derivative. Assume $\kappa \gtrsim 1$. Notice that for the squared error loss, $C_\rho' = 1$. Then, for all $\mu, \nu \in \mathcal{P}_2(\mathbb{R}^{2d+2})$ and $\boldsymbol{w}, \boldsymbol{w}' \in \mathbb{R}^{2d+2}$, we have*

$$\left\|\nabla \hat{\mathcal{J}}_0'[\mu](\boldsymbol{w}) - \nabla \hat{\mathcal{J}}_0'[\mu'](\boldsymbol{w}')\right\| \lesssim \kappa C_\rho \left\|\mathbb{E}_{S_n}\left[\tilde{\boldsymbol{x}}\tilde{\boldsymbol{x}}^\top\right]\right\| \|\boldsymbol{w} - \boldsymbol{w}'\| + C_\rho' \left\|\mathbb{E}_{S_n}\left[\tilde{\boldsymbol{x}}\tilde{\boldsymbol{x}}^\top\right]\right\| W_2(\mu, \mu'),$$

*for the Lipschitz loss, and*

$$\left\|\nabla \hat{\mathcal{J}}_0'[\mu](\boldsymbol{w}) - \nabla \hat{\mathcal{J}}_0'[\mu'](\boldsymbol{w}')\right\| \lesssim \kappa \sqrt{\hat{\mathcal{J}}_0(\mu)}\left\|\mathbb{E}_{S_n}\left[\tilde{\boldsymbol{x}}^{\otimes 4}\right]\right\|_{2\to 2} \|\boldsymbol{w} - \boldsymbol{w}'\| + \left\|\mathbb{E}_{S_n}\left[\tilde{\boldsymbol{x}}\tilde{\boldsymbol{x}}^\top\right]\right\| W_2(\mu, \mu'),$$

*for the squared error loss, where $\left\|\mathbb{E}_{S_n}\left[\tilde{\boldsymbol{x}}^{\otimes 4}\right]\right\|_{2\to 2} := \sup_{\|\boldsymbol{v}\|\leq 1}\left\|\mathbb{E}_{S_n}\left[\langle\tilde{\boldsymbol{x}}, \boldsymbol{v}\rangle^2 \tilde{\boldsymbol{x}}\tilde{\boldsymbol{x}}^\top\right]\right\|$.*

**Proof.** Recall that $\hat{\mathcal{J}}_0'[\mu](\boldsymbol{w}) = \mathbb{E}_{S_n}[\rho'(\hat{y}(\boldsymbol{x}; \mu) - y)\Psi(\boldsymbol{x}; \boldsymbol{w})]$, where $\Psi(\boldsymbol{x}; \boldsymbol{w}) = \phi_{\kappa,\iota}(\langle\boldsymbol{\omega}_1, \tilde{\boldsymbol{x}}\rangle) - \phi_{\kappa,\iota}(\langle\boldsymbol{\omega}_2, \tilde{\boldsymbol{x}}\rangle)$. We start with the triangle inequality,

$$\left\|\nabla \hat{\mathcal{J}}_0'[\mu](\boldsymbol{w}) - \nabla \hat{\mathcal{J}}_0'[\mu'](\boldsymbol{w}')\right\| \leq \left\|\nabla \hat{\mathcal{J}}_0'[\mu](\boldsymbol{w}) - \nabla \hat{\mathcal{J}}_0'[\mu](\boldsymbol{w}')\right\| + \left\|\nabla \hat{\mathcal{J}}_0'[\mu](\boldsymbol{w}') - \nabla \hat{\mathcal{J}}_0'[\mu'](\boldsymbol{w}')\right\|.$$

We now focus on the first term. For the Lipschitz loss,

$$\left\|\nabla_{\boldsymbol{\omega}_1}\hat{\mathcal{J}}_0'[\mu](\boldsymbol{w}) - \nabla_{\boldsymbol{\omega}_1}\hat{\mathcal{J}}_0'[\mu](\boldsymbol{w}')\right\| = \left\|\mathbb{E}_{S_n}\left[\rho'(\hat{y}(\boldsymbol{x}; \mu) - y)(\phi_{\kappa,\iota}'(\langle\boldsymbol{\omega}_1, \tilde{\boldsymbol{x}}\rangle) - \phi_{\kappa,\iota}'(\langle\boldsymbol{\omega}_1', \tilde{\boldsymbol{x}}\rangle)\tilde{\boldsymbol{x}}\right]\right\|$$

$$\leq C_\rho \mathbb{E}_{S_n}\left[(\phi_{\kappa,\iota}'(\langle\boldsymbol{\omega}_1, \tilde{\boldsymbol{x}}\rangle) - \phi_{\kappa,\iota}'(\langle\boldsymbol{\omega}_1', \tilde{\boldsymbol{x}}\rangle))^2\right]^{1/2}\left\|\mathbb{E}_{S_n}\left[\tilde{\boldsymbol{x}}\tilde{\boldsymbol{x}}^\top\right]\right\|^{1/2}$$

$$\leq C_\rho \kappa \mathbb{E}_{S_n}\left[\langle\boldsymbol{\omega}_1 - \boldsymbol{\omega}_1', \tilde{\boldsymbol{x}}\rangle^2\right]^{1/2}\left\|\mathbb{E}_{S_n}\left[\tilde{\boldsymbol{x}}\tilde{\boldsymbol{x}}^\top\right]\right\|^{1/2}$$

$$\leq C_\rho \kappa \left\|\mathbb{E}_{S_n}\left[\tilde{\boldsymbol{x}}\tilde{\boldsymbol{x}}^\top\right]\right\|\|\boldsymbol{\omega}_1 - \boldsymbol{\omega}_1'\|,$$

where the first inequality follows from Lemma 18, and the second inequality follows from the fact that $|\phi''_\kappa| \le \kappa$. For the squared error loss, we have

$$
\begin{aligned}
\left\|\nabla_{\boldsymbol{\omega}_1}\hat{\mathcal{J}}'_0[\mu](\boldsymbol{w}) - \nabla_{\boldsymbol{\omega}_1}\hat{\mathcal{J}}'_0[\mu](\boldsymbol{w}')\right\| &= \left\|\mathbb{E}_{S_n}\left[(\hat{y}(\boldsymbol{x};\mu) - y)(\phi'_{\kappa,\iota}(\langle\boldsymbol{w},\tilde{\boldsymbol{x}}\rangle) - \phi'_{\kappa,\iota}(\langle\boldsymbol{w}',\tilde{\boldsymbol{x}}\rangle)\tilde{\boldsymbol{x}}\right]\right\| \\
&= \sup_{\|\boldsymbol{v}\|\le 1}\mathbb{E}_{S_n}\left[(\hat{y}(\boldsymbol{x};\mu) - y)(\phi'_{\kappa,\iota}(\langle\boldsymbol{\omega}_1,\tilde{\boldsymbol{x}}\rangle) - \phi'_{\kappa,\iota}(\langle\boldsymbol{\omega}'_1,\tilde{\boldsymbol{x}}\rangle)\langle\boldsymbol{v},\tilde{\boldsymbol{x}}\rangle\right] \\
&\le \sup_{\|\boldsymbol{v}\|\le 1}\sqrt{\mathbb{E}_{S_n}[(\hat{y}(\boldsymbol{x};\mu) - y)^2]\mathbb{E}_{S_n}\left[(\phi'_{\kappa,\iota}(\langle\boldsymbol{\omega}_1,\tilde{\boldsymbol{x}}\rangle) - \phi'_{\kappa,\iota}(\langle\boldsymbol{\omega}'_1,\tilde{\boldsymbol{x}}\rangle))^2\langle\boldsymbol{v},\tilde{\boldsymbol{x}}\rangle^2\right]} \\
&\le \kappa\sqrt{\hat{\mathcal{J}}_0(\mu)\sup_{\|\boldsymbol{v}\|\le 1}\left\langle\boldsymbol{v},\mathbb{E}_{S_n}\left[\langle\boldsymbol{\omega}_1 - \boldsymbol{\omega}'_1,\tilde{\boldsymbol{x}}\rangle^2\tilde{\boldsymbol{x}}\tilde{\boldsymbol{x}}^\top\right]\boldsymbol{v}\right\rangle} \\
&\le \kappa\sqrt{\hat{\mathcal{J}}_0(\mu)\left\|\mathbb{E}_{S_n}\left[\langle\boldsymbol{\omega}_1 - \boldsymbol{\omega}'_1,\tilde{\boldsymbol{x}}\rangle^2\tilde{\boldsymbol{x}}\tilde{\boldsymbol{x}}^\top\right]\right\|} \\
&\le \kappa\sqrt{\hat{\mathcal{J}}_0(\mu)\left\|\mathbb{E}_{S_n}\left[\tilde{\boldsymbol{x}}^{\otimes 4}\right]\right\|_{2\to 2}}\|\boldsymbol{\omega}_1 - \boldsymbol{\omega}'_1\|.
\end{aligned}
$$

Similar bounds apply to the gradient with respect to $\boldsymbol{\omega}_2$, which completes the bound on the first term of the triangle inequality.

We now consider the second term of the triangle inequality. Here we consider Lipschitz losses and the squared error loss at the same time since both have a Lipschitz derivative.

$$
\begin{aligned}
\left\|\nabla_{\boldsymbol{\omega}_1}\hat{\mathcal{J}}'_0[\mu](\boldsymbol{\omega}') - \nabla_{\boldsymbol{\omega}_1}\hat{\mathcal{J}}'_0[\mu](\boldsymbol{\omega}')\right\| &= \left\|\left(\rho'(\hat{y}(\boldsymbol{x};\mu) - y) - \rho'(\hat{y}(\boldsymbol{x};\mu') - y)\right)\phi'_{\kappa,\iota}(\langle\boldsymbol{\omega}'_1,\tilde{\boldsymbol{x}}\rangle)\tilde{\boldsymbol{x}}\right\| \\
&\le \mathbb{E}_{S_n}\left[\left(\rho'(\hat{y}(\boldsymbol{x};\mu) - y) - \rho'(\hat{y}(\boldsymbol{x};\mu') - y)\right)^2\right]^{1/2}\left\|\mathbb{E}_{S_n}\left[\tilde{\boldsymbol{x}}\tilde{\boldsymbol{x}}^\top\right]\right\|^{1/2} \\
&\le C'_\rho\mathbb{E}_{S_n}\left[(\hat{y}(\boldsymbol{x};\mu) - \hat{y}(\boldsymbol{x};\mu'))^2\right]^{1/2}\left\|\mathbb{E}_{S_n}\left[\tilde{\boldsymbol{x}}\tilde{\boldsymbol{x}}^\top\right]\right\|^{1/2},
\end{aligned}
$$
(A.14)

where the first inequality follows from Lemma 18. Let $\gamma \in \mathcal{P}_2(\mathbb{R}^{2d+2}\times\mathbb{R}^{2d+2})$ be a coupling of $\mu$ and $\mu'$ (i.e. the first and second marginals of $\gamma$ are equal to $\mu$ and $\mu'$ respectively). Recall that,

$$
\hat{y}(\boldsymbol{x};\mu) - \hat{y}(\boldsymbol{x};\mu') = \int\left(\phi_{\kappa,\iota}(\langle\boldsymbol{\omega}_1,\tilde{\boldsymbol{x}}\rangle) - \phi_{\kappa,\iota}(\langle\boldsymbol{\omega}_2,\tilde{\boldsymbol{x}}\rangle) - \phi_{\kappa,\iota}(\langle\boldsymbol{\omega}'_1,\tilde{\boldsymbol{x}}\rangle) + \phi_{\kappa,\iota}(\langle\boldsymbol{\omega}'_2,\tilde{\boldsymbol{x}}\rangle)\right)\mathrm{d}\gamma(\boldsymbol{w},\boldsymbol{w}').
$$

Therefore by the triangle inequality for the $L_2$ norm $\mathbb{E}_{S_n}\left[(\cdot)^2\right]^{1/2}$ and Jensen's inequality,

$$
\begin{aligned}
\mathbb{E}_{S_n}\left[(\hat{y}(\boldsymbol{x};\mu) - \hat{y}(\boldsymbol{x};\mu'))^2\right]^{1/2} &\le \mathbb{E}_{S_n}\left[\int\left(\phi_{\kappa,\iota}(\langle\boldsymbol{\omega}_1,\tilde{\boldsymbol{x}}\rangle) - \phi_{\kappa,\iota}(\langle\boldsymbol{\omega}'_1,\tilde{\boldsymbol{x}}\rangle)\right)^2\mathrm{d}\gamma\right]^{1/2} \\
&\quad + \mathbb{E}_{S_n}\left[\int\left(\phi_{\kappa,\iota}(\langle\boldsymbol{\omega}_2,\tilde{\boldsymbol{x}}\rangle) - \phi_{\kappa,\iota}(\langle\boldsymbol{\omega}'_2,\tilde{\boldsymbol{x}}\rangle)\right)^2\mathrm{d}\gamma\right]^{1/2} \\
&\le \int\mathbb{E}_{S_n}\left[\langle\boldsymbol{\omega}_1 - \boldsymbol{\omega}'_1,\tilde{\boldsymbol{x}}\rangle^2\right]^{1/2}\mathrm{d}\gamma + \int\mathbb{E}_{S_n}\left[\langle\boldsymbol{\omega}_2 - \boldsymbol{\omega}'_2,\tilde{\boldsymbol{x}}\rangle^2\right]^{1/2}\mathrm{d}\gamma \\
&\le \left\|\mathbb{E}_{S_n}\left[\tilde{\boldsymbol{x}}\tilde{\boldsymbol{x}}^\top\right]\right\|^{1/2}\int(\|\boldsymbol{\omega}_1 - \boldsymbol{\omega}'_1\| + \|\boldsymbol{\omega}_2 - \boldsymbol{\omega}'_2\|)\mathrm{d}\gamma(\boldsymbol{w}_1,\boldsymbol{w}_2) \\
&\le \sqrt{2\left\|\mathbb{E}_{S_n}\left[\tilde{\boldsymbol{x}}\tilde{\boldsymbol{x}}^\top\right]\right\|}\int\|\boldsymbol{w} - \boldsymbol{w}'\|^2\mathrm{d}\gamma(\boldsymbol{w},\boldsymbol{w}').
\end{aligned}
$$

By choosing $\gamma$ whose transport cost attains (or converges to) the optimal cost, we have

$$
\mathbb{E}_{S_n}\left[(\hat{y}(\boldsymbol{x};\mu) - \hat{y}(\boldsymbol{x};\mu'))^2\right]^{1/2} \le \sqrt{2\left\|\mathbb{E}_{S_n}\left[\tilde{\boldsymbol{x}}\tilde{\boldsymbol{x}}^\top\right]\right\|}W_2(\mu,\mu').
$$

Plugging the above result into (A.14), we have

$$
\left\|\nabla_{\boldsymbol{\omega}_1}\hat{\mathcal{J}}'_0[\mu](\boldsymbol{\omega}') - \nabla_{\boldsymbol{\omega}_2}\hat{\mathcal{J}}'_0[\mu'](\boldsymbol{\omega}')\right\| \le \sqrt{2}C'_\rho\left\|\mathbb{E}_{S_n}\left[\tilde{\boldsymbol{x}}\tilde{\boldsymbol{x}}^\top\right]\right\|W_2(\mu,\mu').
$$

Notice that the same bound holds for gradients with respect to $\boldsymbol{\omega}_2$. Thus the bound of the second term in the triangle inequality and the proof is complete. □

**Lemma 29** (Boundedness of $\nabla\hat{\mathcal{J}}_0'$). *In the same setting as Lemma 28, for all $\mu \in \mathcal{P}_2(\mathbb{R}^{2d+2})$ and $\boldsymbol{w} \in \mathbb{R}^{2d+2}$, we have*

$$\left\|\nabla\hat{\mathcal{J}}_0'[\mu](\boldsymbol{w})\right\| \leq \sqrt{2}\tilde{C}_\rho\left\|\mathbb{E}_{S_n}\left[\tilde{\boldsymbol{x}}\tilde{\boldsymbol{x}}^\top\right]\right\|^{1/2},$$

*where $\tilde{C}_\rho = C_\rho$ when $\rho$ is Lipschitz and $\tilde{C}_\rho = \sqrt{2\hat{\mathcal{J}}_0(\mu)}$ when $\rho$ is the squared error loss.*

**Proof.** Notice that $|\phi_{\kappa,\iota}'| \leq 1$. Therefore,

$$\left\|\nabla_{\boldsymbol{\omega}_1}\hat{\mathcal{J}}_0'[\mu](\boldsymbol{w})\right\| = \left\|\mathbb{E}_{S_n}\left[\rho'(\hat{y} - y)\phi_{\kappa,\iota}'(\langle\boldsymbol{\omega}_1, \tilde{\boldsymbol{x}}\rangle)\tilde{\boldsymbol{x}}\right]\right\|$$

$$\leq \sqrt{\mathbb{E}_{S_n}[\rho'(\hat{y} - y)^2]\left\|\mathbb{E}_{S_n}\left[\tilde{\boldsymbol{x}}\tilde{\boldsymbol{x}}^\top\right]\right\|}$$

$$\leq \tilde{C}_\rho\left\|\mathbb{E}_{S_n}\left[\tilde{\boldsymbol{x}}\tilde{\boldsymbol{x}}^\top\right]\right\|^{1/2},$$

where the first inequality follows from Lemma 18. $\qquad\square$

**Lemma 30** (Boundedness of $\hat{\mathcal{J}}_0''$). *In the same setting as Lemma 28, for all $\mu \in \mathcal{P}_2(\mathbb{R}^{2d+2})$ and $\boldsymbol{w}, \boldsymbol{w}' \in \mathbb{R}^{2d+2}$, we have*

$$\left|\hat{\mathcal{J}}_0''[\mu](\boldsymbol{w}, \boldsymbol{w}')\right| \leq C_\rho'\left\|\mathbb{E}_{S_n}\left[\tilde{\boldsymbol{x}}\tilde{\boldsymbol{x}}^\top\right]\right\|\left(\|\boldsymbol{w}\|^2 + \|\boldsymbol{w}'\|^2\right),$$

*where we recall that $C_\rho' = 1$ for the squared error loss.*

**Proof.** Via the definition given by (2.6), it is straightforward to show that

$$\hat{\mathcal{J}}_0''[\mu](\boldsymbol{w}, \boldsymbol{w}') = \mathbb{E}_{S_n}[\rho''(\hat{y}(\boldsymbol{x}; \mu) - y)\Psi(\boldsymbol{x}; \boldsymbol{w})\Psi(\boldsymbol{x}; \boldsymbol{w}')].$$

Then, by the Cauchy-Schwartz inequality,

$$\hat{\mathcal{J}}_0''[\mu](\boldsymbol{w}, \boldsymbol{w}') \leq C_\rho'\mathbb{E}_{S_n}\left[\Psi(\boldsymbol{x}; \boldsymbol{w})^2\right]^{1/2}\mathbb{E}_{S_n}\left[\Psi(\boldsymbol{x}; \boldsymbol{w}')^2\right]^{1/2}.$$

Moreover, by the Lipschitzness of $\phi_{\kappa,\iota}$,

$$\mathbb{E}_{S_n}\left[\Psi(\boldsymbol{x}; \boldsymbol{w})^2\right]^{1/2} \leq \mathbb{E}_{S_n}\left[\langle\boldsymbol{\omega}_1, \tilde{\boldsymbol{x}}\rangle^2\right]^{1/2} + \mathbb{E}_{S_n}\left[\langle\boldsymbol{\omega}_2, \tilde{\boldsymbol{x}}\rangle^2\right]^{1/2}$$

$$\leq \sqrt{2}\left\|\mathbb{E}_{S_n}\left[\tilde{\boldsymbol{x}}\tilde{\boldsymbol{x}}^\top\right]\right\|^{1/2}\|\boldsymbol{w}\|$$

We can similarly bound the expression for $\boldsymbol{w}'$, and arrive at the statement of the lemma via Young's inequality,

$$\hat{\mathcal{J}}_0''[\mu](w, w') \leq 2\left\|\mathbb{E}_{S_n}\left[\tilde{\boldsymbol{x}}\tilde{\boldsymbol{x}}^\top\right]\right\|C_\rho'\|\boldsymbol{w}\|\|\boldsymbol{w}'\| \leq \left\|\mathbb{E}_{S_n}\left[\tilde{\boldsymbol{x}}\tilde{\boldsymbol{x}}^\top\right]\right\|\left(\|\boldsymbol{w}\|^2 + \|\boldsymbol{w}'\|^2\right).$$

$\qquad\square$

We collect the smoothness estimates and simplify them under the event of Lemma 16 in the following Corollary.

**Corollary 31.** *Suppose $\rho$ and $\rho'$ are $C_\rho$ and $C_\rho'$ Lipschitz respectively, with $C_\rho, C_\rho' \lesssim 1$. Recall that $\boldsymbol{\Sigma} := \mathbb{E}\left[\boldsymbol{x}\boldsymbol{x}^\top\right]$. On the event of Lemma 16, we have $\left\|\mathbb{E}_{S_n}\left[\tilde{\boldsymbol{x}}\tilde{\boldsymbol{x}}^\top\right]\right\| \lesssim \|\boldsymbol{\Sigma}\| \vee \tilde{r}_x^2$, and consequently, $\hat{\mathcal{J}}_0'$ satisfies Assumption 5 with constants $L \lesssim \kappa(\|\boldsymbol{\Sigma}\| \vee \tilde{r}_x^2)$, $R \lesssim \|\boldsymbol{\Sigma}\|^{1/2} \vee \tilde{r}_x$, and $C_L = \kappa^{-1}$.*

Using the estimates above, we can present the following convergence bound $\mathcal{F}_{\beta,\lambda}^m(\mu_\beta^*) - \mathcal{F}_{\beta,\lambda}(\mu_\beta^*)$.

**Proposition 32.** *Let $\bar{r}_x := \|\boldsymbol{\Sigma}\| \vee \tilde{r}_x$, and for simplcity assume $C_{\mathrm{LSI}} \geq \beta$. For any $\varepsilon \lesssim 1$, suppose the step size satisfies*

$$\eta \lesssim \frac{\varepsilon}{C_{\mathrm{LSI}}\kappa^2\bar{r}_x^4(d + \bar{r}_x^2/\lambda)},$$

*the width of the network satisfies,*

$$m \gtrsim \frac{\kappa \bar{r}_x^2 \Big(1 + \big(\frac{\bar{r}_x^2}{\lambda^2} + \frac{d}{\lambda\beta}\big)\big(\frac{1}{\kappa} + \frac{\kappa \bar{r}_x^2 C_{\mathrm{LSI}}}{\beta}\big)\Big)}{\varepsilon},$$

*and the number of iterations satisfies*

$$l \gtrsim \frac{\beta C_{\mathrm{LSI}}}{\eta} \ln \Big(\frac{\mathcal{F}_{\beta,\lambda}^m(\mu_0^m) - \mathcal{F}_\beta^*}{\varepsilon}\Big).$$

*Then, we have $\mathcal{F}_{\beta,\lambda}^m(\mu_l^m) - \mathcal{F}_{\beta,\lambda}(\mu_\beta^*) \leq \varepsilon$.*

**Proof.** Throughout the proof, we will assume the event of Lemma 16 holds. Let $\mathcal{F}_{\beta,\lambda}^* := \mathcal{F}_{\beta,\lambda}(\mu_\beta^*)$. Notice that by iterating the bound of Theorem 27, we have

$$\mathcal{F}_{\beta,\lambda}^m(\mu_l^m) - \mathcal{F}_{\beta,\lambda}^* \leq \exp\Big(\frac{-l\eta}{2\beta C_{\mathrm{LSI}}}\Big)(\mathcal{F}_{\beta,\lambda}^m(\mu_0^m) - \mathcal{F}_{\beta,\lambda}^*) + \frac{\eta A_{m,\beta,\lambda,\eta}}{1 - \exp\big(\frac{-\eta}{2\beta C_{\mathrm{LSI}}}\big)}$$

$$\leq \exp\Big(\frac{-l\eta}{2\beta C_{\mathrm{LSI}}}\Big)(\mathcal{F}_{\beta,\lambda}^m(\mu_0^m) - \mathcal{F}_{\beta,\lambda}^*) + 4\beta C_{\mathrm{LSI}} A_{m,\beta,\lambda,\eta},$$

where the second inequality holds for $\eta \leq 2\beta C_{\mathrm{LSI}}$ since $1 - e^{-x} \geq x/2$ for $x \in [0,1]$. We now bound $A_{m,\beta,\lambda,\eta}$ so that the RHS of the above is less than $\mathcal{O}(\varepsilon)$ by choosing a sufficiently large $m$ and a sufficiently small $\eta$. Recall that given constants $L$ and $R$ from Assumption 5,

$$A_{m,\beta,\lambda,\eta} \asymp L^2\big(d + \frac{R^2}{\lambda}\big)\big(\eta^2 + \frac{\eta}{\beta}\big) + \frac{L}{m\beta}\Big(\frac{1}{C_{\mathrm{LSI}}} + \big(\frac{R^2}{\lambda^2} + \frac{d}{\lambda\beta}\big)\big(\frac{C_L}{C_{\mathrm{LSI}}} + \frac{L}{\beta}\big)\Big).$$

From Corollary 31, $L \asymp \kappa(\|\mathbf{\Sigma}\| \vee \tilde{r}_x^2)$, $R \asymp \|\mathbf{\Sigma}\|^{1/2} \vee \tilde{r}_x$, and $C_L = \kappa^{-1}$. To avoid notational clutter, let $\bar{r}_x^2 := \|\mathbf{\Sigma}\| \vee \tilde{r}_x^2$. Then, to control the terms containing $\eta$, it suffices to choose

$$\eta \lesssim \sqrt{\frac{\varepsilon}{\beta C_{\mathrm{LSI}} \kappa^2 \bar{r}_x^4 (d + \bar{r}_x^2/\lambda)}} \wedge \frac{\varepsilon}{C_{\mathrm{LSI}} \kappa^2 \bar{r}_x^4 (d + \bar{r}_x^2/\lambda)},$$

for which we can simply choose

$$\eta \lesssim \frac{\varepsilon}{C_{\mathrm{LSI}} \kappa^2 \bar{r}_x^4 (d + \bar{r}_x^2/\lambda)}.$$

Further, to control the term containing the number of particles $m$, we need

$$m \gtrsim \frac{\kappa \bar{r}_x^2 \Big(1 + \big(\frac{\bar{r}_x^2}{\lambda^2} + \frac{d}{\lambda\beta}\big)\big(\frac{1}{\kappa} + \frac{\kappa \bar{r}_x^2 C_{\mathrm{LSI}}}{\beta}\big)\Big)}{\varepsilon}.$$

To drive the suboptimality bound below $\varepsilon$, we also need to let the number of iterations $l$ satisfy

$$l \gtrsim \frac{\beta C_{\mathrm{LSI}}}{\eta} \ln \Big(\frac{\mathcal{F}_{\beta,\lambda}^m(\mu_0^m) - \mathcal{F}_{\beta,\lambda}^*}{\varepsilon}\Big).$$

With the above conditions, we can guarantee

$$\mathcal{F}_{\beta,\lambda}^m(\mu_l^m) - \mathcal{F}_{\beta,\lambda}^* \lesssim \varepsilon,$$

which finishes the proof. $\qquad\square$

Further, we now present the proof of the LSI estimate given by Proposition 2.

**Proof.** [Proof of Proposition 2] Recall that

$$\hat{\mathcal{J}}_\lambda'[\mu_{\mathbf{W}^l}](\mathbf{w}) = \hat{\mathcal{J}}_0'[\mu_{\mathbf{W}^l}](\mathbf{w}) + \frac{\lambda}{2}\|\mathbf{w}\|^2.$$

Thus we have $\nu_{\mu_{\mathbf{W}^l}}(\mathbf{w}) \propto \gamma(\mathbf{w}) \exp(-\beta \hat{\mathcal{J}}_0'[\mu_{\mathbf{W}^l}](\mathbf{w}))$. Since $\gamma$ satisfies the LSI with constant $1/(\beta\lambda)$, by the Holley-Stroock perturbation argument (Holley & Stroock, 1986), $\nu_{\mu_{\mathbf{W}^l}}$ satifies the LSI with constant

$$C_{\mathrm{LSI}} \leq \frac{\exp(\beta \operatorname{osc}(\hat{\mathcal{J}}_0'[\mu_{\mathbf{W}^l}]))}{\beta\lambda}.$$

Additionally,

$$\left| \hat{\mathcal{J}}_0'[\mu_{\boldsymbol{W}^l}](\boldsymbol{w}) \right| = \left| \frac{1}{n} \sum_{i=1}^{n} \rho'(\hat{y}(\boldsymbol{x}; \mu_{\boldsymbol{W}^l}) - y)\Psi(\boldsymbol{x}^{(i)}; \boldsymbol{w}) \right| \leq 2C_\rho \iota,$$

which completes the proof. $\qquad\square$

Finally, we are ready to present the proof of Theorem.

### A.5 Proof of Theorem 3 and Corollary 4

Recall that $\lambda = \tilde{\lambda} r_x^2$, and let $\beta = \frac{d_{\text{eff}} + \tilde{r}_x^2/r_x^2}{\varepsilon^2 \tilde{\lambda}}$ and $n \geq \frac{(d_{\text{eff}} + \tilde{r}_x^2/r_x^2)\iota^2}{\tilde{\lambda}\varepsilon^4}$ for some $\varepsilon \lesssim 1$, where $\tilde{\varepsilon} := \tilde{\mathcal{O}}(\tilde{\lambda}^{\frac{1}{k+2}} + \varepsilon + \kappa^{-1})$. Then, as long as $\iota \gtrsim \frac{\tilde{r}_x^2}{\tilde{\lambda}r_x^2}$, from Lemma 25, we have $\mathcal{J}_0(\mu_\beta^*) - \mathbb{E}[\rho(\xi)] \lesssim \tilde{\varepsilon}$. Note that while Lemma 25 only asks for $\iota \gtrsim \tilde{\lambda}^{-\frac{k}{k+2}}(\tilde{r}_x/r_x)^{\frac{2(k+1)}{k+2}}$, we simplify this expression in the statement of Theorem 3 so that the choice of $\iota$ does not depend on $k$.

On the other hand, given the step size $\eta$, width $m$, and number of iterations $l$ by Proposition 32, we have $\mathcal{F}_{\beta,\lambda}^m(\mu_l^m) - \mathcal{F}_{\beta,\lambda}(\mu_\beta^*) \leq \varepsilon$. Therefore,

$$\mathbb{E}_{\boldsymbol{W}\sim\mu_l^m}[J_0(\boldsymbol{W})] - \mathcal{J}_0(\mu_\beta^*) \lesssim \sqrt{\frac{\bar{r}_x^2 \beta C_{\text{LSI}}\varepsilon}{m} + \frac{\iota^2}{m}}.$$

Additionally, from Lemma 25, we have

$$\beta^{-1}\mathcal{H}(\mu_\beta^* \mid \gamma) \lesssim \mathbb{E}[\rho(\xi)] + \tilde{\mathcal{O}}(\tilde{\lambda}^{\frac{1}{k+2}}) + \varepsilon + \kappa^{-1} \lesssim 1.$$

Consequently, for $m \geq \frac{\bar{r}_x^2(d_{\text{eff}} + \tilde{r}_x^2/r_x^2)C_{\text{LSI}}}{\tilde{\lambda}\varepsilon^3} \vee \frac{\iota}{\varepsilon^2}$, we have $\mathbb{E}_{\boldsymbol{W}\sim\mu_l^m}[J_0(\boldsymbol{W})] - \mathcal{J}_0(\mu_\beta^*) \leq \varepsilon$. Therefore, combining the bounds above, we have

$$\mathbb{E}_{\boldsymbol{W}\sim\mu_l^m}[J_0(\boldsymbol{W})] - \mathbb{E}[\rho(\xi)] \lesssim \tilde{\mathcal{O}}(\tilde{\lambda}^{\frac{1}{k+2}}) + \varepsilon + \kappa^{-1}.$$

Consequently, we can take $\tilde{\lambda} = o_n(1)$, $\varepsilon = o_n(1)$, $\kappa^{-1} = o_n(1)$, which finishes the proof of Theorem 3.

We finally remark that under the LSI estimate of Proposition 2 and the choice of hyperparameters in Theorem 3, the sufficient number of neurons and iterations can be bounded by

$$m \leq \tilde{\mathcal{O}}\left(\frac{\bar{r}_x^4}{c_x^4}\left(\frac{d}{d_{\text{eff}}} + \frac{\bar{r}_x^2}{r_x^2}\right)e^{\tilde{\mathcal{O}}(d_{\text{eff}})}\right) \leq \tilde{\mathcal{O}}\left(\frac{\bar{r}_x^4 d_{\text{eff}}}{c_x^4}\left(\frac{d}{d_{\text{eff}}^2} + \frac{\bar{r}_x^2}{c_x^2}\right)e^{\tilde{\mathcal{O}}(d_{\text{eff}})}\right) \leq \tilde{\mathcal{O}}\left(de^{\tilde{\mathcal{O}}(d_{\text{eff}})}\right),$$

and

$$l \leq \tilde{\mathcal{O}}\left(\frac{\bar{r}_x^4 d}{c_x^4}e^{\tilde{\mathcal{O}}(d_{\text{eff}})}\right) \leq \tilde{\mathcal{O}}\left(de^{\tilde{\mathcal{O}}(d_{\text{eff}})}\right),$$

which completes the proof of Corollary 4. $\qquad\square$

## B Proofs of Section 4

We begin with the proof of Proposition 8.

**Proof.** [Proof of Proposition 8] Note that $\hat{\mathcal{J}}_0(\mu^*) = 0$ by definition. Moreover, the bound on $\mathcal{H}(\mu^* \mid \tau)$ is a simple application of Jensen's inequality, namely,

$$\mathcal{H}(\mu^* \mid \tau) = \int \ln \frac{e^f}{\int e^f \mathrm{d}\tau} \mathrm{d}\mu^* = \int f \mathrm{d}\mu^* - \ln \int e^f \mathrm{d}\tau \leq \int f(\mathrm{d}\mu^* - \mathrm{d}\tau).$$

$\qquad\square$

Next, using the Bakry-Émery curvature-dimension condition (Bakry & Émery, 1985), we prove the following dimension-free LSI bound.

**Proof.** [Proof of Proposition 9] By the curvature-dimension condition ([Bakry et al., 2014](#), Section 5.7), the Gibbs measure $\nu_\mu \propto \exp(-\beta\hat{\mathcal{J}}_0'[\mu])$ satisfies the LSI with constant $C_{\mathrm{LSI}} \leq \alpha^{-1}$ as long as

$$\mathrm{Ric}_{\mathfrak{g}} + \beta\nabla^2\hat{\mathcal{J}}_0'[\eta](\boldsymbol{w}) \geq \alpha\mathfrak{g},$$

for all $\boldsymbol{w} \in \mathcal{W}$ and some $\alpha > 0$. By the bound on the Ricci curvature from Assumption 4, it suffices to show

$$\varrho d\mathfrak{g} + \beta\nabla^2\hat{\mathcal{J}}_0'[\eta](\boldsymbol{w}) \succcurlyeq \alpha\mathfrak{g}.$$

Recall that

$$\hat{\mathcal{J}}_0(\mu) = \mathbb{E}_{S_n}\left[\rho\left(\int \Psi(\boldsymbol{x};\boldsymbol{w})\mathrm{d}\mu(\boldsymbol{w}) - y\right)\right].$$

Therefore,

$$\hat{\mathcal{J}}_0'[\mu](\boldsymbol{w}) = \mathbb{E}_{S_n}[\rho'(\hat{y}(\boldsymbol{x};\mu) - y)\Psi(\boldsymbol{x};\boldsymbol{w})],$$

and

$$\nabla_{\boldsymbol{w}}^2\hat{\mathcal{J}}_0'[\mu](\boldsymbol{w}) = \mathbb{E}_{S_n}\left[\rho'(\hat{y}(\boldsymbol{x};\mu) - y)\nabla_{\boldsymbol{w}}^2\Psi(\boldsymbol{x};\boldsymbol{w})\right].$$

Consider the case where $\rho$ is $C_\rho$ Lipschitz. Then,

$$\begin{aligned}
\lambda_{\min}(\nabla_{\boldsymbol{w}}^2\hat{\mathcal{J}}_0'[\mu](\boldsymbol{w})) &= \inf_{\|\boldsymbol{v}\|_{\mathfrak{g}}\leq 1}\mathbb{E}_{S_n}\left[\rho'(\hat{y}(\boldsymbol{x};\mu) - y)\langle\boldsymbol{v},\nabla_{\boldsymbol{w}}^2\Psi(\boldsymbol{x};\boldsymbol{w})\boldsymbol{w}\rangle\right] \\
&\geq -C_\rho\sup_{\|\boldsymbol{v}\|_{\mathfrak{g}}\leq 1}\mathbb{E}_{S_n}\left[|\langle\boldsymbol{v},\nabla_{\boldsymbol{w}}^2\Psi(\boldsymbol{x};\boldsymbol{w})\boldsymbol{v}\rangle|\right] \\
&= -C_\rho K.
\end{aligned}$$

$\square$

Before stating the proof of Theorem 10, we adapt the generalization analysis of Appendix A.3 to the Riemannian setting of this section. Recall the truncated risk functions $\mathcal{J}_0^\varkappa(\mu) = \mathbb{E}[\rho(\hat{y}(\boldsymbol{x};\mu) - y) \wedge \varkappa]$ and $\hat{\mathcal{J}}_0^\varkappa(\mu) = \mathbb{E}_{S_n}[\rho(\hat{y}(\boldsymbol{x};\mu) - y) \wedge \varkappa]$. Then, we have the following uniform convergence bound.

**Lemma 33.** *Under the setting of Example 7, where we recall $|\varphi(0)| \lesssim 1$ and $|\varphi'(z)| \lesssim 1$ for all $z$, we have*

$$\mathbb{E}\left[\sup_{\mu\in\mathcal{P}^{\mathrm{ac}}(\mathcal{W}):\mathcal{H}(\mu\,|\,\tau)\leq M}\mathcal{J}_0^\varkappa(\mu) - \hat{\mathcal{J}}_0^\varkappa(\mu)\right] \lesssim C_\rho\sqrt{\frac{M}{n}}\left(1 + \mathbb{E}\left[\left\|\hat{\boldsymbol{\Sigma}}\right\|^{1/2}\right]\right),$$

*where $\hat{\boldsymbol{\Sigma}} := \frac{1}{n}\sum_{i=1}^n\boldsymbol{x}^{(i)}\boldsymbol{x}^{(i)\top}$. Combined with McDiarmid's inequality, the above bound implies*

$$\sup_{\mathcal{H}(\mu\,|\,\tau)\leq M}\mathcal{J}_0^\varkappa(\mu) - \hat{\mathcal{J}}_0^\varkappa(\mu) \lesssim C_\rho\sqrt{\frac{M}{n}}\left(1 + \mathbb{E}\left[\left\|\hat{\boldsymbol{\Sigma}}\right\|^{1/2}\right]\right) + \varkappa\sqrt{\frac{\ln(1/\delta)}{n}},$$

*with probability at least $1 - \delta$.*

**Proof.** Based on the same argument as Lemma 22, for any $\alpha > 0$, we have

$$\mathbb{E}\left[\sup_{\mathcal{H}(\mu\,|\,\tau)\leq M}\mathcal{J}_0^\varkappa(\mu) - \hat{\mathcal{J}}_0^\varkappa(\mu)\right] \leq 2C_\rho\,\mathbb{E}\left[\sup_{\mathcal{H}(\mu\,|\,\tau)\leq M}\frac{1}{n}\sum_{i=1}^n\xi_i\hat{y}(\boldsymbol{x}^{(i)};\mu)\right]$$

where $(\xi_i)$ are i.i.d. Rademacher random variables. Once again following Lemma 22, we have,

$$\mathbb{E}_\xi\left[\sup_{\mathcal{H}(\mu\,|\,\tau)\leq M}\frac{1}{n}\sum_{i=1}^n\xi_i\hat{y}(\boldsymbol{x}^{(i)};\mu)\right] \leq \frac{M}{\alpha} + \frac{1}{\alpha}\mathbb{E}\left[\ln\int\exp\left(\frac{\alpha^2}{2n^2}\sum_{i=1}^n\Psi(\boldsymbol{x}^{(i)};\boldsymbol{w})^2\right)\mathrm{d}\tau(\boldsymbol{w})\right].$$

Furthermore,

$$\begin{aligned}
\int\exp\left(\frac{\alpha^2}{2n^2}\sum_{i=1}^n\Psi(\boldsymbol{x}^{(i)};\boldsymbol{w})^2\right)\mathrm{d}\tau(\boldsymbol{w}) &\leq \exp\left(\frac{\alpha^2\varphi(0)^2}{n^2}\right)\int\exp\left(\frac{\alpha^2\|\varphi'\|_\infty^2}{n^2}\sum_{i=1}^n\langle\boldsymbol{w},\boldsymbol{x}^{(i)}\rangle^2\right)\mathrm{d}\tau(\boldsymbol{w}) \\
&\leq \exp\left(\frac{\alpha^2\varphi(0)^2}{n}\right)\int\exp\left(\frac{\alpha^2\|\varphi'\|_\infty^2}{n}\langle\boldsymbol{w},\hat{\boldsymbol{\Sigma}}\boldsymbol{w}\rangle\right)\mathrm{d}\tau(\boldsymbol{w}) \\
&\leq \exp\left(\frac{\alpha^2\left[\varphi(0)^2 + \|\varphi'\|_\infty^2\left\|\hat{\boldsymbol{\Sigma}}\right\|\right]}{n}\right).
\end{aligned}$$

Therefore,

$$\mathbb{E}_\xi\left[\sup_{\mathcal{H}(\mu\,|\,\tau)\leq M}\frac{1}{n}\sum_{i=1}^n\xi_i\hat{y}(\boldsymbol{x}^{(i)};\mu)\right]\leq\frac{M}{\alpha}+\alpha\frac{\varphi(0)^2+\|\varphi'\|_\infty^2\left\|\hat{\boldsymbol{\Sigma}}\right\|}{n}.$$

Using $|\varphi(0)|,\|\varphi'\|_\infty\lesssim 1$ and optimizing over $\alpha$ yield

$$\mathbb{E}_\xi\left[\sup_{\mathcal{H}(\mu\,|\,\tau)\leq M}\frac{1}{n}\sum_{i=1}^n\xi_i\hat{y}(\boldsymbol{x}^{(i)};\mu)\right]\lesssim\sqrt{\frac{M}{n}}\left(1+\left\|\hat{\boldsymbol{\Sigma}}^{1/2}\right\|\right).$$

Taking expectation with respect to the training set concludes the proof. $\qquad\square$

We can also control the effect of truncating similar to that of Lemma 24.

**Lemma 34.** *Under the setting of Example 7, for any $\mu\in\mathcal{P}(\mathcal{W})$, we have*

$$\mathcal{J}_0(\mu)-\mathcal{J}_0^\varkappa(\mu)\leq 2C_\rho^2\cdot\frac{2\varphi(0)^2+2\|\varphi'\|_\infty^2\|\boldsymbol{\Sigma}\|+\mathbb{E}\big[y^2\big]}{\varkappa}.$$

**Proof.** Similarly to the arguments in Lemma 24, by using the Cauchy-Schwartz and Markov inequalities, we have

$$\begin{aligned}
\mathcal{J}_0(\mu)-\mathcal{J}_0^\varkappa(\mu)&\leq\mathbb{E}[\mathbb{1}(\rho(\hat{y}(\boldsymbol{x};\mu)-y)\geq\varkappa)\rho(\hat{y}(\boldsymbol{x};\mu-y))]\\
&\leq\mathbb{P}(\rho(\hat{y}(\boldsymbol{x};\mu)-y)\geq\varkappa)^{1/2}\,\mathbb{E}\big[\rho(\hat{y}(\boldsymbol{x};\mu)-y)^2\big]^{1/2}\\
&\leq C_\rho\mathbb{P}(|\hat{y}(\boldsymbol{x};\mu)-y|\geq\varkappa/C_\rho)^{1/2}\,\mathbb{E}\big[(\hat{y}(\boldsymbol{x};\mu)-y)^2\big]^{1/2}\\
&\leq C_\rho^2\frac{\mathbb{E}\big[(\hat{y}(\boldsymbol{x};\mu)-y)^2\big]}{\varkappa}\\
&\leq 2C_\rho^2\frac{\mathbb{E}\big[\hat{y}(\boldsymbol{x};\mu)^2\big]+\mathbb{E}\big[y^2\big]}{\varkappa}.
\end{aligned}$$

Moreover, we have

$$\begin{aligned}
\mathbb{E}\big[\hat{y}(\boldsymbol{x};\mu)^2\big]\leq\mathbb{E}\left[\int\Psi(\boldsymbol{x};\boldsymbol{w})^2\mathrm{d}\mu(\boldsymbol{w})\right]&\leq 2\varphi(0)^2+2\|\varphi'\|_\infty^2\,\mathbb{E}\left[\int\langle\boldsymbol{w},\boldsymbol{x}\rangle^2\mathrm{d}\mu(\boldsymbol{w})\right]\\
&\leq 2\varphi(0)^2+2\|\varphi'\|_\infty^2\|\boldsymbol{\Sigma}\|,
\end{aligned}$$

concluding the proof of the lemma. $\qquad\square$

Finally, we can state the proof of the main theorem of this section.

**Proof.** [Proof of Theorem 10] Note that given $\beta=\frac{\bar{\Delta}}{\bar{\varepsilon}}$ and $d\geq 2C_\rho K\bar{\Delta}/(\varrho\varepsilon)$, Proposition 9 guarantees that $C_{\mathrm{LSI}}\leq 2/(\varrho d)$ along the trajectory. Consequently, by the convergence guarantee of (2.10), we have

$$\mathcal{F}_\beta(\mu_T)\leq\mathcal{F}_\beta(\mu_\beta^*)+e^{-\frac{\varrho dT}{\beta}}(\mathcal{F}_\beta(\mu_0)-\mathcal{F}_\beta(\mu_\beta^*))\leq\mathcal{F}_\beta(\mu_\beta^*)+\varepsilon.$$

Further, by $\bar{\mu}$ of Assumption 4, we have

$$\mathcal{F}_\beta(\mu_\beta^*)\leq\mathcal{F}_\beta(\bar{\mu})\leq\bar{\varepsilon}+\beta^{-1}\bar{\Delta}\leq\bar{\varepsilon}+\varepsilon.$$

As a result, $\hat{\mathcal{J}}_0(\mu_T)\leq\mathcal{F}_\beta(\mu_T)\leq\bar{\varepsilon}+2\varepsilon$, and similarly $\mathcal{H}(\mu_T\,|\,\tau)\leq\beta(\bar{\varepsilon}+2\varepsilon)$.

Note that $\hat{\mathcal{J}}_0^\varkappa(\mu_T)\leq\hat{\mathcal{J}}_0(\mu_T)$. Using the fact that $C_\rho,\|\boldsymbol{\Sigma}\|,\mathbb{E}\big[|y|^2\big]\lesssim 1$, and combining the bounds of Lemma 33 and Lemma 34, with a porbability of failure $\delta=\mathcal{O}(n^{-q})$ for some constant $q>0$, we have

$$\mathcal{J}_0(\mu_T)-\hat{\mathcal{J}}_0(\mu_T)\lesssim\sqrt{\frac{\beta(\bar{\varepsilon}+\varepsilon)}{n}}+\varkappa\sqrt{\frac{\ln n}{n}}+\frac{1}{\varkappa}.$$

Optimizing over $\varkappa$ implies

$$\mathcal{J}_0(\mu_T) \lesssim \bar{\varepsilon} + \varepsilon + \sqrt{\frac{\beta(\bar{\varepsilon} + \varepsilon)}{n}} + \left(\frac{\ln n}{n}\right)^{1/4}$$

$$\lesssim \bar{\varepsilon} + \varepsilon + \sqrt{\frac{\bar{\Delta}(1 + \bar{\varepsilon}/\varepsilon)}{n}} + \left(\frac{\ln n}{n}\right)^{1/4}.$$

Choosing $n$ according to the statement of the theorem completes the proof. $\qquad\square$

## C    COMPARISONS WITH THE FORMULATION OF NITANDA ET AL. (2024)

Here, we provide a number of comparisons with results of Nitanda et al. (2024). In Section C.1, we show that the statistical model (2.1) is more general than their formulation, even for parity learning problems. In Section C.2, we provide an informal comparison of their effective dimension to our setting, exhibiting the improvement in our definition of effective dimension.

### C.1    GENERALITY OF THE FORMULATIONS

We begin by pointing out that the formulation of $k$-index model of (2.1) is strictly more general than that of Nitanda et al. (2024), even for learning $k$-sparse parities. Recall that in their setting, they consider inputs of the type $x = \Sigma^{1/2}z$ for some positive definite $\Sigma$, where $z \sim \mathrm{Unif}(\{\pm 1\}^d)$ (their original formulation uses $z \sim \mathrm{Unif}(\{\pm 1/\sqrt{d}\}^d)$, but we rescale the input to be consistent with the notation of this paper). The labels are given by

$$y = \mathrm{sign}\Big(\prod_{i=1}^k \langle \tilde{u}_i, z \rangle\Big) = \mathrm{sign}\Big(\prod_{i=1}^k \big\langle \Sigma^{-1/2}\tilde{u}_i, x \big\rangle\Big), \tag{C.1}$$

where $\{\tilde{u}_i\}_{i=1}^k$ are orthonormal vectors. Then, we can define an orthonormal set of vectors $\{u_i\}_{i=1}^k$ such that $\mathrm{span}(u_1, \dots, u_k) = \mathrm{span}(\Sigma^{-1/2}\tilde{u}_1, \dots, \Sigma^{-1/2}\tilde{u}_k)$, and define $g$ such that

$$g\left(\frac{\langle u_1, x\rangle}{\sqrt{k}}, \dots, \frac{\langle u_k, x\rangle}{\sqrt{k}}\right) = g\left(\frac{\big\langle \Sigma^{1/2}u_1, z\big\rangle}{\sqrt{k}}, \dots, \frac{\big\langle \Sigma^{1/2}u_k, z\big\rangle}{\sqrt{k}}\right) = \mathrm{sign}\Big(\prod_{i=1}^k \langle \tilde{u}_i, z\rangle\Big),$$

for all $z \in \{\pm 1\}^d$. Therefore, the parity formulation of (C.1) can be seen as a special case of the $k$-index model (2.1). Note that $g$ is only defined on $2^d$ points, and we can extend it to all of $\mathbb{R}^k$ such that $g : \mathbb{R}^k \to \mathbb{R}$ is Lipschitz continuous.

In contrast, the $k$-index model can represent parity problems that cannot be represented by (C.1). Starting from an orthonormal set of vectors $\{u_i\}_{i=1}^k$ in $\mathbb{R}^d$, let

$$y = g\left(\frac{\langle u_1, x\rangle}{\sqrt{k}}, \dots, \frac{\langle u_k, x\rangle}{\sqrt{k}}\right) = \mathrm{sign}\Big(\prod_{i=1}^k \langle u_i, x\rangle\Big). \tag{C.2}$$

Consider the case where $k = 2$, then $y = \mathrm{sign}\Big(\big\langle \Sigma^{1/2}u_1, z\big\rangle\big\langle \Sigma^{1/2}u_2, z\big\rangle\Big)$. To be able to reformulate this to (C.1), we need to be find orthonromal $\tilde{u}_1, \tilde{u}_2 \in \mathbb{R}^d$ such that

$$\mathrm{sign}\left(\big\langle \Sigma^{1/2}u_1, z\big\rangle\big\langle \Sigma^{1/2}u_2, z\big\rangle\right) = \mathrm{sign}(\langle \tilde{u}_1, z\rangle\langle \tilde{u}_2, z\rangle), \quad \forall z \in \{\pm 1\}^d.$$

If $\Sigma$ has rank less than $d$ such that $\Sigma^{1/2}u_1 = \Sigma^{1/2}u_2$, then the above implies $\mathrm{sign}(\langle \tilde{u}_1, z\rangle\langle \tilde{u}_2, z\rangle) \geq 0$ for all $z \in \{\pm 1\}^d$. In particular, we must have some $z$ where $\mathrm{sign}(\langle \tilde{u}_1, z\rangle\langle \tilde{u}_2, z\rangle) > 0$, which implies that

$$\sum_{i=1}^{2^d} \langle \tilde{u}_1, z_i\rangle\langle \tilde{u}_2, z_i\rangle = \left\langle \tilde{u}_1, \sum_{i=1}^{2^d} z_i z_i^\top \tilde{u}_2 \right\rangle = 2^d\langle \tilde{u}_1, \tilde{u}_2\rangle > 0, \tag{C.3}$$

which is in contradiction with $\langle \tilde{u}_1, \tilde{u}_2\rangle = 0$. Therefore, for such $\Sigma$, we cannot formulate (C.2) as a special case of (C.1). This argument is robust with respect to small perturbations of $\Sigma$ which make it

full-rank. Specifically, suppose $\boldsymbol{\Sigma}^{1/2}\boldsymbol{u}_2 = \boldsymbol{\Sigma}^{1/2}\boldsymbol{u}_1 + \boldsymbol{\delta}$. Notice that we can choose $\boldsymbol{\Sigma}^{1/2}\boldsymbol{u}_1$ such that $\left\langle \boldsymbol{\Sigma}^{1/2}\boldsymbol{u}_1, \boldsymbol{z} \right\rangle^2 \neq 0$ for all $\boldsymbol{z} \in \{\pm 1\}^d$, e.g. by choosing $\boldsymbol{\Sigma}^{1/2}\boldsymbol{u}_1 \propto \boldsymbol{e}_1$, i.e. the first standard basis vector. It is straightforward to construct full-rank $\boldsymbol{\Sigma}$, $\boldsymbol{u}_1$, and $\boldsymbol{u}_2$ such that $\|\boldsymbol{\delta}\|$ is arbitrarily small, in which case

$$\text{sign}\left(\left\langle \boldsymbol{\Sigma}^{1/2}\boldsymbol{u}_1, \boldsymbol{z}\right\rangle\left\langle \boldsymbol{\Sigma}^{1/2}\boldsymbol{u}_2, \boldsymbol{z}\right\rangle\right) = \text{sign}\left(\left\langle \boldsymbol{\Sigma}^{1/2}\boldsymbol{u}_1, \boldsymbol{z}\right\rangle^2 + \left\langle \boldsymbol{\Sigma}^{1/2}\boldsymbol{u}_1, \boldsymbol{z}\right\rangle\langle \boldsymbol{\delta}, \boldsymbol{z}\rangle\right) \geq 0.$$

Following (C.3), once again the above would be in contradiction with $\langle \tilde{\boldsymbol{u}}_1, \tilde{\boldsymbol{u}}_2 \rangle = 0$. This implies that the $k$-index model (2.1) is strictly more general than (C.1) when considering full-rank covariance matrices.

## C.2 COMPARISON WITH THE EFFECTIVE DIMENSION OF NITANDA ET AL. (2024)

A close inspection of the proofs in Nitanda et al. (2024) demonstrates that one can define their effective dimension in a scale invariant manner as $\tilde{d}_{\text{eff}} := \text{tr}(\boldsymbol{\Sigma})\left\|\sum_{i=1}^{k}\boldsymbol{\Sigma}^{-1/2}\tilde{\boldsymbol{u}}_i\right\|^2$. From the previous section, we observed that to reduce their setting to ours, we need to choose a set $\{\boldsymbol{u}_i\}_{i=1}^{k}$ of normalized vectors that spans the set of vectors $\{\boldsymbol{\Sigma}^{-1/2}\tilde{\boldsymbol{u}}_i\}_{i=1}^{k}$. In particular, we can choose $\boldsymbol{u}_i = \frac{\boldsymbol{\Sigma}^{-1/2}\tilde{\boldsymbol{u}}_i}{\|\boldsymbol{\Sigma}^{-1/2}\tilde{\boldsymbol{u}}_i\|}$, or equivalently write $\tilde{\boldsymbol{u}}_i = \frac{\boldsymbol{\Sigma}^{1/2}\boldsymbol{u}_i}{\|\boldsymbol{\Sigma}^{1/2}\boldsymbol{u}_i\|}$. While $\{\boldsymbol{u}_i\}_{i=1}^{k}$ are not orthogonal, our proofs do not strictly rely on the orthogonality assumption and it is only made for simplicity. Hence, we have

$$\tilde{d}_{\text{eff}} = \text{tr}(\boldsymbol{\Sigma})\left\|\sum_{i=1}^{k}\frac{\boldsymbol{u}_i}{\left\|\boldsymbol{\Sigma}^{1/2}\boldsymbol{u}_i\right\|}\right\|^2 \leq k\,\text{tr}(\boldsymbol{\Sigma})\sum_{i=1}^{k}\left\|\boldsymbol{\Sigma}^{1/2}\boldsymbol{u}_i\right\|^{-2}.$$

Note that the above upper bound is sharp when $k = 1$, and is lower bounded by our definition of effective dimension stated in Definition 1. Therefore, we use the above bound in Table 1.

## D   EXAMPLES OF TARGET FUNCTIONS IN PROPOSITION 8

In this section, we provide a natural example of a target function of the form given in Proposition 8. Suppose $\mathcal{W} = \mathbb{S}^{d-1}$, and $\Psi(\boldsymbol{x}; \boldsymbol{w}) = \langle \boldsymbol{w}, \boldsymbol{x}\rangle^2$. Define $\mu^* = \mathcal{T}\#\tau$, where $\mathcal{T}(\boldsymbol{v}) = \frac{\boldsymbol{A}^{1/2}\boldsymbol{v}}{\|\boldsymbol{A}^{1/2}\boldsymbol{v}\|}$, for some PSD matrix $\boldsymbol{A}$ to be defined later. Let

$$y = \hat{y}(\boldsymbol{x}; \mu^*) = \left\langle \boldsymbol{x}, \mathbb{E}_{\boldsymbol{w}\sim\mu^*}\big[\boldsymbol{w}\boldsymbol{w}^\top\big]\boldsymbol{x}\right\rangle = \left\langle \boldsymbol{x}, \mathbb{E}_{\boldsymbol{v}\sim\tau}\Big[\frac{\boldsymbol{A}^{1/2}\boldsymbol{v}\boldsymbol{v}^\top\boldsymbol{A}^{1/2}}{\|\boldsymbol{A}^{1/2}\boldsymbol{v}\|^2}\Big]\boldsymbol{x}\right\rangle.$$

By defining $\boldsymbol{B} = d \cdot \mathbb{E}_{\boldsymbol{v}\sim\tau}\Big[\frac{\boldsymbol{A}^{1/2}\boldsymbol{v}\boldsymbol{v}^\top\boldsymbol{A}^{1/2}}{\|\boldsymbol{A}^{1/2}\boldsymbol{v}\|^2}\Big]$, we have $y = \frac{1}{d}\left\|\boldsymbol{B}^{1/2}\boldsymbol{x}\right\|^2$. Additionally, note that $\hat{y}(\boldsymbol{x}; \tau) = \frac{1}{d}\|\boldsymbol{x}\|^2$. Before proceeding further, we remark that the typical $\|\boldsymbol{x}\|$ should be of order $\sqrt{d}$ to get $\Theta(1)$ output, which is due to choosing the unit sphere as the weight space.

Next, we construct $\boldsymbol{A}$. Suppose distribution of $\boldsymbol{x}$ is such that with probability $1/2$ we have $\boldsymbol{x} = \boldsymbol{e}_1$, the first standard basis vector. Let $\boldsymbol{A} = \text{diag}(\lambda_1, 1, \ldots, 1)$. Then,

$$y = \hat{y}(\boldsymbol{e}_1; \mu^*) = \mathbb{E}_{\boldsymbol{v}\sim\tau}\left[\frac{\lambda_1 d v_1^2}{1 + (\lambda_1 - 1)v_1^2}\right] \leq c\lambda_1,$$

where $c$ is an absolute constant. On the other hand, $\hat{y}(\boldsymbol{e}_1; \tau) = 1$. Choosing $\lambda_1 = \frac{1}{d}$, we observe that for sufficiently large $d$, with probability at least $1/2$ the distance $|\hat{y}(\boldsymbol{x}; \mu^*) - \hat{y}(\boldsymbol{x}; \tau)| \geq C$ for some absolute constant $C > 0$. This means that $\mu^*$ and $\tau$ are meaningfully different, and from an initialization of $\tau$ one needs to train the network (e.g. with MFLD) to recover $\mu^*$. It remains to find an estimate on $\mathcal{H}(\mu^* \,|\, \tau)$.

Let $\mathfrak{T} : \mathbb{R}^d \to \mathbb{S}^{d-1}$ denote the normalization mapping, i.e. $\mathfrak{T}(\boldsymbol{v}) = \frac{\boldsymbol{v}}{\|\boldsymbol{v}\|}$. Then, note that $\tau = \mathfrak{T}\#\mathcal{N}(0, \mathbf{I}_d)$, and $\mu^* = \mathfrak{T}\#\mathcal{N}(0, \boldsymbol{A})$. Therefore, by the data processing inequality,

$$\mathcal{H}(\mu^* \,|\, \tau) \leq \mathcal{H}(\mathcal{N}(0, \boldsymbol{A}) \,|\, \mathcal{N}(0, \mathbf{I}_d)) = -\frac{d}{2} + \frac{\text{tr}(\boldsymbol{A})}{2} - \frac{1}{2}\ln\det(\boldsymbol{A}) \leq \frac{\ln d}{2},$$

where $\lambda_1 = \frac{1}{d}$. Therefore, $\bar{\Delta} = \frac{\ln d}{2} = o(d)$, and we can apply Theorem 10 to obtain polynomial convergence time, as desired.

## E NUMERICAL SIMULATION

In this section, we perform numerical simulations to verify the intuitions from Theorem 3. Specifically, we train a two-layer neural network with width $m = 50$ and ReLU activation, where the first layer weights are initialized uniformly on the sphere, and fix the first half of the second layer coordinates at $+1/m$, and the second half at $-1/m$. The input follows the distribution $\boldsymbol{x} \sim \mathcal{N}(0, \boldsymbol{\Sigma})$ (with an extra 1 appended for bias), where $\boldsymbol{\Sigma} = \text{diag}(\boldsymbol{\sigma}^2)$ with $\sigma_1^2 = 1$ and $\sigma_i^2 = \frac{d_{\text{eff}}-1}{d-1}$ for input dimension $d = 50$. The labels are generated by a single-index model of the following form

$$y = g(\langle \boldsymbol{e}_1, \boldsymbol{x} \rangle) = \frac{\langle \boldsymbol{e}_1, \boldsymbol{x} \rangle^2 - 1}{\sqrt{2}}.$$

Therefore, the effective dimension from Definition 1 is exactly equal to $d_{\text{eff}}$. We train the neural network using the squared loss with MFLA, with a stepsize of 0.1, weight decay parameter 0.01, temperature 0.001.

Figure 1b shows the test loss at the end of 200 iterations of MFLA for different numbers of training samples $n$ and effective dimension $d_{\text{eff}}$. For each value of $n$ and $d_{\text{eff}}$, we average the test loss over 5 independent runs with different realizations of data and initialization. In Figure 2 we measure the generalization gap, i.e. the average loss difference on the training set of $n$ samples, and a test set of 100000 samples, at the end of 3000 iterations of training with MFLA. For this experiment, we try $n = 100$, $n = 200$, and $n = 500$. As seen from both figures, $d_{\text{eff}}$ controls the generalization gap and test loss, both of which decay with larger $n$.

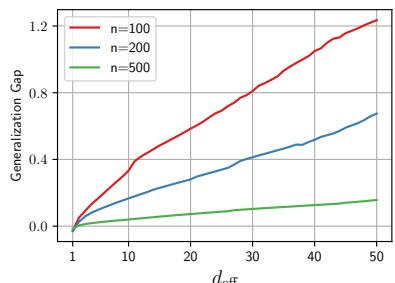

Figure 2: Generalization gap measured by variying the effective dimension.

The code to reproduce the experimental results is provided at: https://github.com/mousavih/MFLD-Learnability.

