# OpenReview forum: "Learning Multi-Index Models with Neural Networks via Mean-Field Langevin Dynamics"
_ICLR.cc/2025/Conference — ICLR 2025 Poster_

### Official Review · Reviewer_PdGr · 2024-10-21

**Soundness:** 4
**Presentation:** 3
**Contribution:** 3
**Rating:** 8
**Confidence:** 3

**Summary:**

This paper studies the sample and computational complexities of the mean field Langevin algorithm (MFLA). The analysis is centered aroung a newly introduced ``effective dimension'' which measures the complexity of the learning task (in terms of the data distribution and the teacher model). Under several regularity conditions and conditions on various constants, it is proven that the sample complexity and the computational complexity can both be controlled in terms of the effective dimension, which can be significantly smaller than the ambient dimension of the input space. The authors estimate (theoretically) this effective dimension on several examples. While the sample complexity bound is very satisfying and scales linearly in the effective dimension, the computational complexity may scale exponentially in the effective dimension and linearly in the ambient dimension. While this might be an issue in the general case, the authors show that this issue can be alleviate in a compact Riemannian setting. Improving the exponential dependence in the Euclidean setting remains open.

**Strengths:**

- Both the statistical and the computational complexity are bounded in terms of a newly introduced effective dimension, which is a measure of the complexity of the learning task. It is interesting that the same quantity can be used for both notions of complexity.

 - The study of Mean-Filed Langevin Dynamics may provide important insights on the sample complexity of modern deep learning models.

 - The limitations of the paper are well-discussed.

 - The paper is well-written. Even though the subject is quite technical, good technical background and references are provided, making it sufficiently clear. The assumptions and theorems are clearly stated.

**Weaknesses:**

- The computational complexity may scale exponentially with the effective dimension (in the worst case). While the authors show that this might not be an issue in a compact Riemannian setting, the issues persists in the Euclidean setting. Additional discussion on how to go beyond this Riemanian setting could enhance the paper. In particular, what are (maybe intuitively) the conditions that could lead to non-exponential complexity?

 - As discussed in the paper, the analysis is restricted to a particular class of 2 layers neural networks. Moreover, the analysis is restricted to ``simple'' learning tasks, ie, the teacher models uses $k \ll d$ neurons. It could be interesting to further discuss what the results become when $k$ Is large.

**Questions:**

- Can we derive a LSI like in Proposition 2 in a setting where there is no explicit regularization, but instead we make a dissipativity assumption on the model? Could it be of any interest to extend the analysis beyond regularization?

 - Could you explain further the meaning of the expectation $E_\xi [\rho(\xi)]$ is Equation (3.6)?

 - Could part of the analysis be extended to more complex function classes than 2 layers neural networks?

 - Is it possible to construct explicit examples where the computational complexity becomes exponential in the effective dimension?

---

> ### Author Response · Authors · 2024-11-23
>
> We thank the reviewer for their valuable evaluation and insightful feedback, and address their comments and questions below.
>
> ---
> **What conditions can lead to non-exponential complexity?**
>
> A main challenge in proving polynomial-time convergence here is to find settings in which the log-Sobolev constant scales polynomially with the inverse temperature. Unfortunately, as long as the Gibbs potential has disjoint local minima with large energy barriers, one cannot hope to establish polynomial log-Sobolev inequalities in general [1]. We believe that further analysis on the Gibbs potential can identify settings in which no such energy barrier exists, which would likely include simpler functions such as those with ***information exponent*** equal to 1, indeed partial progress towards this has been achieved in a two-timescales setting [2]. This is an interesting direction for future research, as it would provide a unified analysis for learning these ***simpler*** functions as opposed to the ad-hoc analyses in the current literature, and would cover a much broader class of input distributions.
>
> ---
> **The analysis is restricted to 2-layer neural networks and $k \ll d$**:
>
> We agree with the reviewer that the setting of our theoretical study is quite simple in comparison with deep neural networks used in practice. However, we remark that two-layer neural networks have been the focus of most recent theoretical work on feature learning with neural networks. There have been attempts at generalizing the mean-field formulation to more than 2 layers, e.g. [3], but much less is known in that setting, and it is an interesting direction to prove learnability guarantees with gradient-based algorithms for deeper networks.\
> Please note that since the minimax lower bounds in this setting suggest that learning the multi-index model without additional assumptions (such as ridge separability [5]) and with $\epsilon$ error requires $n = \Omega\big((1/\epsilon)^{\Omega(k)}\big)$ samples, a statistically efficient algorithm can only exist when $k \ll d$. However, we believe Proposition 8 provides one interesting example of a setting in which $k$ is large, i.e. the target depends on many neurons, yet both sample and time complexity scale mildly with problem parameters as shown in Theorem 10.
>
> ---
> **Proving LSI without regularization/with dissipativity**:
>
> There are two key challenges in removing the weight decay regularization in our theoretical analysis.\
> First, as pointed out by the reviewer, this regularization allows us to prove LSI for the Gibbs potential by considering it as a bounded perturbation of a strongly convex potential. We remark that dissipativity-type conditions are not likely to work here. In particular, the gradient of the Gibbs potential with respect to the weights will have a bounded norm (for activations with bounded first derivative), i.e. $\Vert \nabla_w \hat{\mathcal{J}}_0’ [\mu] (w) \Vert \leq C$ for all $w$, thus $\langle w, \nabla_w \hat{\mathcal{J}}_0’ [\mu] (w) \rangle \leq C\Vert w \Vert$, which rules out the possibility of a dissipativity condition.\
> Second, we also require weight decay to be able to apply our generalization bounds. While in practice it is believed that gradient descent will have an inductive bias towards minimum norm solutions that generalize well, this phenomenon is not well studied in the setting we consider here. We believe that removing regularization while still showing that gradient-based algorithms learn networks with good generalization properties is an important open problem.
>
> ---
> **The meaning of $\mathbb{E}_\zeta[\rho(\zeta)]$**:
>
> $\rho$ denotes the loss function, $\zeta$ is the additive noise in the multi-index model. This term can be thought of as an irreducible loss in regression.

---

> > ### Author Response · Authors · 2024-11-23
> >
> > ---
> > **Constructing explicit examples where computational complexity is exponential**:
> >
> > We believe an explicit example can be a single-index model of the form $g(x) = \sigma(\langle u, x\rangle)$, where $\sigma$ is a function with *generative exponent* larger than 2, such as $\sigma(z) = z^2e^{-z^2}$ which has *generative exponent* 4. In this case, the statistical query (SQ) lower bound [4] predicts that any SQ algorithm that can learn with linear sample complexity requires exponential computational complexity. We remark however that this is an informal argument since the SQ framework assumes the access to population expectations is corrupted by adversarial, rather than i.i.d. concentration noise.\
> > Alternatively, we believe it is possible to show that the uniform LSI constant provides a tight characterization of the convergence of MFLD. Then, one can use the approach in [1] to establish exponential lower bounds on the uniform LSI in settings where we have a “double-well” Gibbs potential, i.e. local minima separated by a large energy barrier. Proving such lower bounds rigorously would be an interesting direction for future research.
> >
> > ---
> > We would be happy to clarify any concerns or answer any questions that may come up during the discussion period.
> >
> > References:
> >
> > [1] G. Menz and A. Schichting. “Poincaré and logarithmic Sobolev inequalities by decomposition of the energy landscape.” Annals of Probability, 2014.
> >
> > [2] G. Wang et al. “Mean-Field Langevin Dynamics for Signed Measures via a Bilevel Approach.” arXiv 2024.
> >
> > [3] Z. Chen et al. “A Functional-Space Mean-Field Theory of Partially-Trained Three-Layer Neural Networks.” arXiv 2022.
> >
> > [4] A. Damian et al. "Computational-Statistical Gaps in Gaussian Single-Index Models." COLT 2024.
> >
> > [5] K. Oko et al. “Learning sum of diverse features: computational hardness and efficient gradient-based training for ridge combinations.” COLT 2024.

---

> > > ### Comment · Reviewer_PdGr · 2024-11-25
> > > **Thank you for your answer**
> > >
> > > I thank the authors for their answers which addressed some of my concerns and may help clarify the paper. I think this is a good paper and will maintain my score (8 - accept).

---

### Official Review · Reviewer_bvhi · 2024-11-02

**Soundness:** 3
**Presentation:** 3
**Contribution:** 2
**Rating:** 6
**Confidence:** 3

**Summary:**

In this work, the authors study the task of learning a general multi-index model using neural networks and the
mean-field Langevin Algorithm (MFLA). Specifically, the authors introduce a notion of effective dimension in this context,
which can potentially be much smaller than the ambient dimension when the input distribution is spiked the spikes
align with the relevant subspace of the target multi-index model. Then, the author derive bounds for the sample and
computational complexity of MFLA based the effective dimension in both Euclidean and Riemannian settings.

**Strengths:**

* Overall, the paper is well-written, and the presentation is clear.
* Proving polynomial convergence guarantees for MFLA is an important and challenging subject. This paper makes process in
  this direction by proving a bound that is exponential in the effective dimension. They also show that for certain
  models, the effective dimension can be as small as $O(\mathrm{poly}\log d)$, which leads to a quasi-polynomial
  convergence guarantee.

**Weaknesses:**

* My main concern with this paper is that the effective dimension, as defined in Definition 1, is essentially about
  how spiked the input distribution is (and whether it aligns with the target function) and does NOT capture the potential
  low-dimensional structure of the target function.
  In fact, if the input distribution is isotropic, the effective dimension (as defined in Definition 1) will be
  $d$ even when the target function is a single-index model.
  This limits the applicability of this paper's results, as it does not cover even the simplest Gaussian single-index case.
* The comparison with the information/generative exponent (Section 3.1) is misleading. The authors suggest that one
  advantage of the MFLA analysis over the information/generative exponent analysis is that the sample complexity is
  always $\tilde{O}(d)$ (in the isotropic case), while for SGD, (the bounds on) the sample complexity can be $\tilde{\omega}(d)$
  when the information/generative exponent is larger than $2$. However, this is not true if we are allowed to have
  exponentially many neurons, as assumed in the setting of Corollary 4. It has been shown in the information exponent paper [1]
  that after the neuron attains a constant correlation with the ground-truth direction, the number of samples needed
  in the final "descent" phase is $\tilde{O}(d)$ regardless the information exponent. Meanwhile, if we have
  exponentially many neurons, we can ensure that some neuron will have a constant correlation with the ground-truth
  direction at initialization with at least constant probability. In addition, the number of steps needed in the
  information exponent analysis is also $\tilde{O}(d)$, which is better than the exponential bound given in Corollary 4.
* In Sec. 4 (the Riemannian setting), the authors require the existence of a good distribution $\mu$ with a small loss
  and $H(\mu | \tau) = o(d)$ where $\tau$ is the uniform distribution on the manifold. They justify the second condition
  in Prop. 8 by showing that if the ground-truth distribution is given by $\mu^* \propto e^f \mathrm{d} \tau$, then we have
  $H(\mu^* | \tau) \le \mathrm{osc}(f)$. However, this bound is vacuous if $\mu^*$ is supported only on a few points as $f$ would then take $\infty$ (or a large value if
  we smooth $\mu^*$). This excludes models such as the single-index model and $k$-multi-index model with $k$ being small.
  To be fair, the authors do mention this limitation in the paper. I just think this assumption is too strong/unnatural.
  It would be great if the authors could give a natural example where this condition holds.


[1] Gerard Ben Arous, Reza Gheissari, and Aukosh Jagannath. Online stochastic gradient descent on non-convex losses from high-dimensional inference. 2021

**Questions:**

See the Weakness part of the review.

---

> ### Author Response · Authors · 2024-11-23
>
> We thank the reviewer for their valuable evaluation and insightful feedback, and address their comments and questions below.
>
> ---
> **The comparison with information/generative exponent analysis**:
>
> We thank the reviewer for raising their concern about this comparison. We do not claim that MFLA achieves a sample complexity beyond that suggested by computational lower bounds (dictated by the information/generative exponent). Indeed, with exponential width, one may conclude that a small fraction of neurons enter the descent phase upon random initialization. However, as this subset is exponentially small, it is not clear that standard end-to-end training can achieve sufficient amplification of these “warm” neurons after $O(d)$ steps to lower the loss – this will not be the case if we run the spherical gradient algorithm (which is the update studied in most information/generative exponent analyses), because the neural network output is still dominated by the majority of neurons that have not “escaped from mediocrity”. \
> Moreover, we highlight that the information/generative exponent analysis is only applicable to the ***Gaussian input*** distribution setting (with the exception of [1] which only considers rotationally invariant input distributions and known link function), and is further limited in either assuming a known single-index link function and choosing the same link function as the activation, or having nontrivial modifications of the standard training procedure for 2-layer neural networks. Moreover, most results so far capture learnability for ***single-index*** models, and characterizing learnability of general ***multi-index*** models with this type of analysis is much more challenging, and these analyses typically fall short of providing a concrete sample complexity for such general settings.\
> Part of our motivation here is to instead provide a general and unifying theory for ***all multi-index models*** in a ***distribution free*** sense, which comes at the cost of a larger worst-case computational complexity. Moreover, the adaptivity of such analysis to ***low effective dimension***, i.e. low-dimensional input embedded in a high-dimensional space, has not been established in prior works. We hope that this discussion helps provide additional context in comparing our work with the literature.
>
> ---
> **Examples of natural functions in Proposition 8**:
>
> Thank you for the suggestion. We have added an example in Appendix D that we believe will in fact provide further insights to the readers. In summary, we let $\mathcal{W} = \mathbb{S}^{d-1}$ and $\Psi(x;w) = \langle x, w\rangle^2$. We let $\mu^*$ be the distribution of $\frac{A^{1/2} u}{\Vert A^{1/2} u\Vert }$, where $u \sim \mathrm{Unif}(\mathbb{S}^{d-1})$. Then, $\hat{y}(x;\mu^*) = \frac{1}{d}\Vert B^{1/2} x\Vert^2$ where $B = A^{1/2}\mathbb{E}\left[\frac{uu^\top}{\Vert A^{1/2}u\Vert^2}\right]A^{1/2}$, whereas $\hat{y}(x;\tau) = \frac{1}{d}\Vert x \Vert^2$ (please note that we assume the typical norm of $x$ to be of order $\sqrt{d}$). If we simply let $A = \mathrm{diag}(\lambda_1,1,...,1)$ and $\lambda_1 = 1/d$, then $\Vert B - I_d\Vert_{\mathrm{op}} \geq \Omega(1)$. This means that for some distribution over $x$, the initialization $\tau$ has $\Omega(1)$ loss, and one needs to train with MFLD to recover $\mu^*$ and achieve small loss. Furthermore, $\bar{\Delta} \leq \mathcal{O}(\ln d)$ in this setting, thus we can achieve the polynomial convergence guarantee of Theorem 10.
>
> ---
> **The effective dimension does not capture the latent low dimension in the multi-index model**:
>
> We would like to highlight that there are two sources of potential low-dimensionality, and only having a small number of indices $k$ is not sufficient to learn with linear sample complexity while having polynomial time convergence, as evident by computational lower bounds [2,3]. While for isotropic input covariance we can achieve the optimal linear sample complexity *regardless of information/generative exponent*, we believe that the additional benefit in sample and computational complexity thanks to the *low effective dimension* is an interesting phenomenon which is mostly overlooked in the current literature on feature learning of neural networks, especially since in practice directions of high variation tend to be better predictors of the label, i.e. many problems tend to have a low effective dimension [4].
>
> ---
> We would be happy to clarify any concerns or answer any questions that may come up during the discussion period.
>
> References:
>
> [1] A. Zweig et al. “On single index models beyond gaussian data.” NeurIPS 2023.
>
> [2] A. Damian et al. "Smoothing the landscape boosts the signal for sgd: Optimal sample complexity for learning single index models." NeurIPS 2023.
>
> [3] A. Damian et al. "Computational-Statistical Gaps in Gaussian Single-Index Models." COLT 2024.
>
> [4] T. Hastie et al. "The elements of statistical learning: data mining, inference, and prediction." 2009.

---

> > ### Comment · Reviewer_bvhi · 2024-11-23
> >
> > Thank you for the clarification and the new example. While I still think certain results in the paper are not as strong as what the paper claims to be, they are decent enough and I will raise my score to 6.

---

### Official Review · Reviewer_aq6U · 2024-11-03

**Soundness:** 3
**Presentation:** 3
**Contribution:** 3
**Rating:** 6
**Confidence:** 4

**Summary:**

This paper examines the sample and computational complexity of learning general multi-index models in high-dimensional settings using two-layer neural networks and the mean-field Langevin algorithm. Under very mild assumptions, the authors successfully demonstrate that the sample complexity for such learning scales linearly with the effective input dimension, $d_{\text{eff}}$, which is information-theoretic optimal. However, the computational complexity typically scales exponentially with $d_{\text{eff}}$, which is also in general unavoidable. To address this, the authors explore specific conditions under which computational complexity might have a polynomial dependence on $d_{\text{eff}}$. They identify certain settings—such as particular target functions and constraints on weights to lie on the hypersphere—that allow for the desired polynomial-time convergence.

**Strengths:**

1. This paper is well-written, clear, and mathematically rigorous.

2. Theoretical guarantees and insights into feature learning in deep learning are notoriously challenging, so any solid progress in this area is highly valuable. In particular, I find the paper’s second contribution quite interesting. Identifying cases where polynomial-time complexity (and thus efficient learning) can be achieved under the mean-field regime is of great importance, and this work is among the first to address this question, even if the current assumptions are somewhat restrictive.

**Weaknesses:**

I feel that the result in the first part is somewhat expected (though I recognize that proving it rigorously is still a quite valuable accomplishment. For the second part, I wonder if it might be possible to establish a polynomial-time convergence guarantee for a broader class of target functions (beyond the teacher neuron setup) by furthermore just considering the case where we restrict the weights just to the sphere. In general, I am convinced that a more detailed analysis/discussion would be valuable regarding the setups in which polynomial-time convergence with optimal sample complexity can be achieved, even when the input distribution is inherently high-dimensional, such as standard normal distribution with mean zero and covariance $\Sigma = I_d$.

**Questions:**

See the weakness part.

---

> ### Author Response · Authors · 2024-11-23
>
> We thank the reviewer for their valuable evaluation and insightful feedback, and address their comments and questions below.
>
> ---
> * **A discussion on when polynomial-time convergence is achievable**:
>
> A main challenge in proving polynomial-time convergence here is to find settings in which the log-Sobolev constant scales polynomially with the inverse temperature. Unfortunately, as long as the Gibbs potential has disjoint local minima with large energy barriers, one cannot hope to establish polynomial log-Sobolev inequalities in general [1]. We believe that further analysis on the Gibbs potential can identify settings in which no such energy barrier exists, which would likely include simpler functions such as those with ***information exponent*** equal to 1, indeed partial progress towards this has been achieved in a two-timescales setting [2]. This is an interesting direction for future research, as it would provide a unified analysis for learning these ***simpler*** functions as opposed to the ad-hoc analyses in the current literature, and would cover a much broader class of input distributions.
>
> We would also like to highlight that we have added a natural example of a target function in the form of Proposition 8 in Appendix D.
>
> ---
> We would be happy to clarify any concerns or answer any questions that may come up during the discussion period.
>
> References:
>
> [1] G. Menz and A. Schichting. “Poincaré and logarithmic Sobolev inequalities by decomposition of the energy landscape.” Annals of Probability, 2014.
>
> [2] G. Wang et al. “Mean-Field Langevin Dynamics for Signed Measures via a Bilevel Approach.” arXiv 2024.

---

### Official Review · Reviewer_hWwK · 2024-11-03

**Soundness:** 3
**Presentation:** 3
**Contribution:** 3
**Rating:** 8
**Confidence:** 3

**Summary:**

This paper explores learning multi-index models in high dimensions using a two-layer neural network trained with the mean-field Langevin algorithm (MFLA). The authors identify an "effective dimension" that can significantly reduce sample complexity by adapting to low-dimensional structures in data. While worst-case computational complexity may still grow exponentially, they propose a solution to achieve polynomial-time convergence by constraining weights to a compact manifold under specific conditions.

**Strengths:**

* The presentation is good, which is easy to follow.
* Discussion on previous works is extensive and informative, and comparison with previous works is clear.
* This paper uses MFLA to learn multi-index model and consider the effective dimension, which is novel to me.
* This paper achieves optimal sample complexity, albeit leading to exponential dependence of $d_{eff}$ in $\mathbb{R}^d$. But the authors well explain this phenomenon and improve this dependence in Riemannian manifold under some curvature conditions.

**Weaknesses:**

* Discussion on Assumption 1 is weak.
* Some notations are not clearly stated.
* This paper is short of some simulation.

See more details in Questions below.

**Questions:**

* The authors claim "the Brownian noise can help escaping from spurious local minima and saddle points" after presenting (2.3). Can you provide some reference of this statement? This is different from SGD which uses inexact oracle (stochastic gradient) leading to different randomness at each iteration. Adding Brownian motion is equivalent to adding randomness with the same magnitude at each iteration.

* In the row between 230 and 231, should the weights be $W\in\mathbb{R}^{(2d+2)\times m}$ instead of $W\in\mathbb{R}^{2d+2}$?

* What is $\mathbb{E}_{\xi}[\rho(\xi)]$ in Thm 3?

* Assumption 1 is restrictive to me. Could you provide some references which also consider this assumption on dataset? Or could you verify this assumption on some dataset? Is this assumption only for the proof? If so, why you need this assumption in your proof should be clearly stated in the main paper. It would be helpful to provide some proof sketch of Thm 3 discussing that.

* Since you establish better sample complexity under subgaussian data assumption, it would be interesting to do more simulation even with artificial mean-zero and subgaussian data for empirical comparison with other results in Table 1.

---

> ### Author Response · Authors · 2024-11-23
>
> We thank the reviewer for their valuable evaluation and insightful feedback, and address their comments and questions below.
>
> ---
> **On Brownian noise helping to escape from local minima**:
>
> Indeed, here we add artificial noise different from noise arising from mini-batch sampling in SGD. This noise is crucial for convergence analysis, and intuitively allows the dynamics to escape from saddle points and local minima. Please see [1, Section 5.2] for an example of this phenomenon with mean-field Langevin dynamics, or [2] and references therein for using additional noise in GD/SGD to escape saddle points. We will further highlight this discussion in the final version.
>
> ---
> **The generality of Assumption 1**:
>
> We thank the reviewer for raising their concern here. Our assumption here is much broader than the Gaussian data $x \sim \mathcal{N}(0,I_d)$ assumption of the recent literature on feature learning of neural networks, e.g. [4, 5, 6]. Indeed, Assumption 1 is automatically verified when $x \sim \mathcal{N}(0,\Sigma)$, with $\sigma_n = \sigma_u = \mathcal{O}(1)$. We would like to highlight however that Assumption 1 is much broader than the Gaussian case. For example, if $x = \Sigma^{1/2} z$ where $z$ has i.i.d. mean-zero subGaussian coordinates, then Assumption 1 is verified with $\sigma_u = \sigma_n = \mathcal{O}(1)$ [3, Theorem 6.3.2]. We have clarified this point in the revised version.
>
> ---
> **What is $\mathbb{E}_\zeta[\rho(\zeta)]$ in Theorem 3?**
>
> $\rho$ denotes the loss function, $\zeta$ is the additive noise in the multi-index model. This term can be thought of as an irreducible loss in regression.
>
> ---
> **Numerical Simulation**:
>
> Thank you for this suggestion. We have added a simple numerical simulation to verify the intuitions from our theory, specifically that the effective dimension controls the generalization gap, in Appendix E. We will provide more extensive numerical simulations in the camera-ready version.
>
> Further, we thank the reviewer for mentioning the typo between lines 230 and 231, which we have fixed.
>
>
> ---
> We would be happy to clarify any concerns or answer any questions that may come up during the discussion period.
>
> References:
>
> [1] L. Chizat. “Mean-field langevin dynamics: Exponential convergence and annealing.” TMLR 2022.
>
> [2] H. Daneshmand et al. “Escaping Saddles with Stochastic Gradients.” ICML 2018.
>
> [3] R. Vershynin. “High-Dimensional Probability: An Introduction with Applications in Data Science.”
>
> [4] A. Damian et al. “Neural networks can learn representations with gradient descent.” COLT 2022.
>
> [5] J. Ba et al. “High-dimensional asymptotics of feature learning: how one gradient step improves the representation.” NeurIPS 2022.
>
> [6] A. Bietti et al. “Learning Single-Index Models with Shallow Neural Networks.” NeurIPS 2022.

---

### Meta-Review · Area_Chair_22FL · 2024-12-26

[review text omitted: it was posted to a different submission]

---

### Decision · Program_Chairs · 2025-01-22

Accept (Poster)